# RBFOX2 modulates a metastatic signature of alternative splicing in pancreatic cancer

Amina Jbara[1], Kuan-Ting Lin[2], Chani Stossel[3], Zahava Siegfried[1], Haya Shqerat[1], Adi Amar-Schwartz[1], Ela Elyada[1], Maxim Mogilevsky[1], Maria Raitses-Gurevich[3], Jared L. Johnson[4,5], Tomer M. Yaron[4,5,6,7], Ofek Ovadia[1], Gun Ho Jang[8], Miri Danan-Gotthold[9], Lewis C. Cantley[4,5], Erez Y. Levanon[9], Steven Gallinger[8], Adrian R. Krainer[2], Talia Golan[3] & Rotem Karni[1✉]

Pancreatic ductal adenocarcinoma (PDA) is characterized by aggressive local invasion and metastatic spread, leading to high lethality. Although driver gene mutations during PDA progression are conserved, no specific mutation is correlated with the dissemination of metastases[1–3]. Here we analysed RNA splicing data of a large cohort of primary and metastatic PDA tumours to identify differentially spliced events that correlate with PDA progression. De novo motif analysis of these events detected enrichment of motifs with high similarity to the RBFOX2 motif. Overexpression of RBFOX2 in a patient-derived xenograft (PDX) metastatic PDA cell line drastically reduced the metastatic potential of these cells in vitro and in vivo, whereas depletion of RBFOX2 in primary pancreatic tumour cell lines increased the metastatic potential of these cells. These findings support the role of RBFOX2 as a potent metastatic suppressor in PDA. RNA-sequencing and splicing analysis of RBFOX2 target genes revealed enrichment of genes in the RHO GTPase pathways, suggesting a role of RBFOX2 splicing activity in cytoskeletal organization and focal adhesion formation. Modulation of RBFOX2-regulated splicing events, such as via myosin phosphatase RHO-interacting protein (MPRIP), is associated with PDA metastases, altered cytoskeletal organization and the induction of focal adhesion formation. Our results implicate the splicing-regulatory function of RBFOX2 as a tumour suppressor in PDA and suggest a therapeutic approach for metastatic PDA.

Analyses of pancreatic cancer genomes have revealed four central PDA driver genes: *KRAS*, *SMAD4*, *CDKN2A* and *TP53*[4,5]. Studies attempting to analyse PDA according to gene expression profiles classify PDA into two[6], three[7] or four[8] subtypes and suggest that these may be prognostic for the outcome and predictive for therapy response[9]. Nevertheless, these driver gene mutations and their expression levels, are conserved during PDA progression, but no mutation specific for progression to metastases has been identified[1–3]. There have been few reports on alternative splicing and the role of splicing factors in PDA[10,11]. We hypothesized that alternative splicing has a role in metastatic PDA progression. To test this idea, we used published RNA-sequencing (RNA-seq) datasets from 395 PDA patient samples[1,12–14] and applied principal component analysis (PCA) based on profiling alternative splicing rather than on gene expression[15]. This analysis revealed two clusters that were not classified according to the mutation status of the driver genes *KRAS*, *SMAD4*, *CDKN2A* and *TP53* (Extended Data Fig. 1a–d and Supplementary Table 1).

## Alternative splicing landscape of PDA

Annotation of the clusters based on clinical and/or pathological information revealed that the two clusters correlated well with the tumour site; 61% of primary pancreatic tumours were in cluster 1, and 71% of metastatic tumour samples were in cluster 2 (Fig. 1a). Furthermore, 90% of the metastatic samples in cluster 2 (the metastatic group) were annotated as advanced stage IV, and 96% of the primary samples in cluster 1 (primary group) were annotated as stages IB (5.9%), IIA (13.9%), IIB (70.2%) or III (5.96%), supporting a stage-dependent classification of these samples (Extended Data Fig. 1e and Supplementary Table 1). A significant number of primary tumour samples that were clustered with the metastatic group were identified as advanced stage IV. Conversely, some metastasis samples were clustered with the primary tumour group (see Fig. 1a). One possible explanation for this could be contamination of metastatic samples with non-malignant cells from the microenvironment. Since most of the metastatic samples were liver metastases, we could not perform sub-clustering of the metastatic group on based on

[1]Department of Biochemistry and Molecular Biology, Institute for Medical Research Israel–Canada, Hebrew University–Hadassah Medical School, Jerusalem, Israel. [2]Cold Spring Harbor Laboratory, Cold Spring Harbor, NY, USA. [3]Division of Oncology, Sheba Medical Center Tel Hashomer, Ramat-Gan, Israel. [4]Meyer Cancer Center, Weill Cornell Medicine, New York, NY, USA. [5]Department of Medicine, Weill Cornell Medicine, New York, NY, USA. [6]Englander Institute for Precision Medicine, Institute for Computational Biomedicine, Weill Cornell Medicine, New York, NY, USA. [7]Department of Physiology and Biophysics, Weill Cornell Medicine, New York, NY, USA. [8]Department of Surgery, University of Toronto, Toronto, Ontario, Canada. [9]The Mina and Everard Goodman Faculty of Life Sciences, Bar-Ilan University, Ramat-Gan, Israel. ✉e-mail: rotemka@ekmd.huji.ac.il

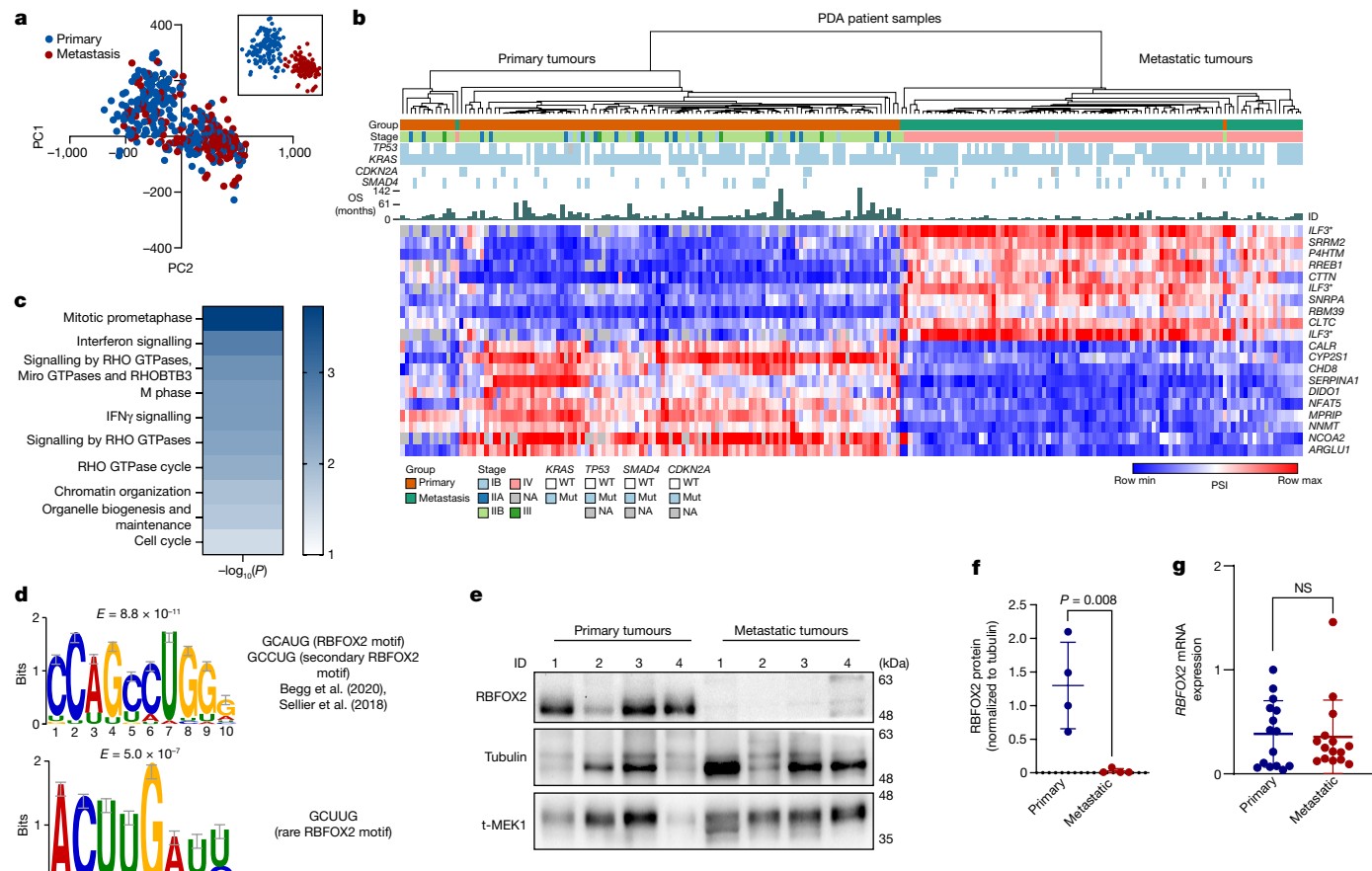

**Fig. 1 | The alternative splicing landscape of PDA. a**, PCA of alternative splicing events in 395 samples from patients with pancreatic cancer, computed by PSI scores. Blue dots represent primary tumour samples, and red dots represent metastatic tumour samples. The inset shows a PCA analysis of a subset of samples whose clinical annotation (either primary or metastatic) matched their respective cluster. **b**, Unsupervised clustering of 249 PDA patient samples based on the top 20 differentially spliced events (PSI). Asterisks indicate different splice events for the same gene. Mutation status, clinical stage and overall survival (OS) are also shown. Max, maximum; min, minimum; mut, mutant; WT, wild type; NA, not available. **c**, Reactome pathway analysis of the significant differentially spliced genes between primary tumours and metastatic tumours. Genes entered into the analysis had an imposed cut-off (|ΔPSI| > 10%) and a nominal P value (P < 0.05, over-representation analysis (hypergeometric distribution) test). Gene sets were limited to between 5 and 500 genes, and pathways were filtered for a statistical threshold of P < 0.05.

**d**, The top two enriched motifs identified in the differentially spliced genes involved in the RHO GTPase pathway[19,40]. **e**, Immunoblot analysis of RBFOX2 in representative primary and PDX metastatic pancreatic tumours. Tubulin and total MEK1 (t-MEK1) serve as loading controls (n = 4 patient samples for each group). Gel source data are provided in Supplementary Fig. 1. **f**, Quantification of RBFOX2 protein relative to tubulin (n = 4 patient samples per group; data are mean ± s.d.) in primary and metastatic tumours. **g**, Quantitative PCR with reverse transcription (RT–qPCR) analysis of *RBFOX2* expression in primary tumours and metastases from patients with PDA (n = 15 patient samples for each group; data are mean ± s.d.). **f**,**g**, Two-tailed Student's t-test. NS, not significant. Genetic alterations and clinical data of patients with PDA are shown in Supplementary Table 1; differentially spliced events in PDA patient samples are shown in Supplementary Table 2; sequence motif enrichment analysis is shown in Supplementary Table 3; reactome analysis is shown in Supplementary Table 4.

tumour location (liver, lung, ascites or heart) (Extended Data Fig. 1f). PCA based on gene expression did not detect the same pattern of clustering (Extended Data Fig. 1g). Therefore, this analysis supports the hypothesis that splicing alterations are associated with PDA progression.

Focusing on the most robust differentially spliced events between the primary and the metastatic groups, we used the samples that were correctly annotated as either primary or metastatic tumours for further analysis (Fig. 1a, inset). Approximately 8,000 significant differential splicing events were identified between the primary and the metastatic groups. Only events that were detected in at least 75% of the samples with |ΔPSI| greater than 10% (where PSI is per cent spliced in) and nominal P value below 0.01 were used for further analysis (Supplementary Table 2). The top 20 splicing events were sufficient to classify the PDA patient samples into two groups with high correlation to clinical stage (Fig. 1b). However, such clustering based on gene expression of these genes did not result in distinct groups (Extended Data Fig. 1h and Supplementary Table 5).

## Enrichment of RBFOX2 motif

Next, we applied an unbiased de novo motif analysis using the XSTREME tool on these events[16]. Two of the significant motifs identified showed similarity to the primary and secondary splicing factor RBFOX2 binding sequences GCAUG, GCCUG or GCUUG (Extended Data Fig. 2a, Supplementary Table 3). Reactome pathway enrichment analysis on the significant differentially spliced events showed enrichment for genes in the RHO GTPase pathway, which is involved in cellular cytoskeleton organization and migration[17] (Fig. 1c and Supplementary Table 4). We then performed an unbiased de novo motif analysis on the differentially spliced events in the RHO pathway genes. The most enriched motif upstream of the 5' splice sites of these events shows high homology to a RBFOX2 binding sequence (Fig. 1d and Supplementary Table 3). Comparison of differentially spliced genes from our analysis to known RBFOX2 target genes[18] detected 274 common genes, with enrichment of genes in

the RHO GTPase pathways (Extended Data Fig. 2b and Supplementary Tables 4 and 7). A recent study showed enrichment of the canonical RBFOX2 motif in genes that regulate cytoskeletal organization[19,20]. These results suggest a possible mechanism by which RBFOX2-mediated splicing may regulate the metastatic process through cytoskeletal changes.

## RBFOX2 acts as a metastatic suppressor

We developed a PDA patient-derived xenograft (PDX) model from primary and metastatic human PDA samples. These samples were subcutaneously injected into NOD-SCID mice and used to generate both PDX tumours and PDX-derived cell lines[21] (Supplementary Table 1). We observed lower protein levels of splicing factor RBFOX2 in PDX-derived metastatic patient samples compared to primary tumour samples, with no significant difference in mRNA levels (Fig. 1e–g). This is in contrast to other splicing factors, such as SRSF6 and SRSF1—known to act as oncogenes in many cancers[22,23]—which have higher protein levels in the metastatic samples compared with the primary tumour samples (Extended Data Fig. 3a,b). The half-life of *RBFOX2* mRNA is similar in the metastatic and in the primary tumour cell lines, whereas the half-life of RBFOX2 protein is shorter in the metastatic X50 cells, suggesting enhanced protein degradation (Extended Data Fig. 3c,d). Several studies have reported that RBFOX2 is upregulated, induces a mesenchymal splicing signature and promotes oncogenic splice-switching that drives an invasive phenotype in certain cancers[24–27]. Other studies have reported decreased expression of RBFOX2 in these cancers[28]. Our results suggest that RBFOX2 has a tumour-suppressive role in the metastatic process of PDA.

We next examined whether RBFOX2 directly influences the metastatic capacity of PDA cells. To this end, we transduced PDX-derived metastatic cells X50 and X139 with RBFOX2-expressing retrovirus (Fig. 2a and Extended Data Fig. 4a). RBFOX2 overexpression drastically decreased the metastatic potential of these cells, as observed by their reduced migration in a wound healing assay and the inhibition of colony formation compared with control cells, without affecting proliferation (Fig. 2b,c and Extended Data Fig. 4b–d). To test the effect of RBFOX2 on invasion in vivo, we injected mCherry-labelled X50 PDX cells transduced with RBFOX2 cDNA intravenously into NOD-SCID mice. We observed a substantial reduction in the number of lung metastases compared with mice injected with control cells (Fig. 2d). There was no difference in tumour volume following subcutaneous injection (Extended Data Fig. 4e). These results suggest that overexpression of RBFOX2 suppresses the metastatic potential of PDA cells. Knockout of RBFOX2 using two different CRISPR guide RNAs significantly increased colony formation and migration rates of BxPC3 primary PDA cells and a primary PDX-derived cell line (X252). Again, no effect on proliferation was detected (Fig. 2a,e,f and Extended Data Fig. 4f–i). Intravenous injection of GFP-labelled RBFOX2-depleted BxPC3 primary PDA cells into NOD-SCID mice revealed an increased number of lung metastases compared with control cells and a significant increase in tumour volume (Fig. 2g and Extended Data Fig. 4j). To confirm whether these phenotypes were attributable to RBFOX2 depletion, we designed a single guide RNA (sgRNA) that targets an exon–intron junction in *RBFOX2* (*RBFOX2* EIJ sgRNA), thereby silencing only endogenous *RBFOX2* (Fig. 2a). Cas9-expressing BxPC3 cells treated with *RBFOX2* EIJ sgRNA had a similar phenotype to RBFOX2-knockout cells, and introducing RBFOX2 cDNA into these cells rescued their phenotype (Fig. 2h,i and Extended Data Fig. 4k).

The RBFOX2 protein contains an RNA recognition motif (RRM), which recognizes the consensus sequence GCAUG. To determine whether the splicing-regulatory activity of RBFOX2 is responsible for its effect on metastatic PDA, we attempted to rescue the phenotype of X50 metastatic PDA cells and BxPC3 primary PDA cells that express *RBFOX2* EIJ sgRNA, using *RBFOX2* cDNA that is deleted for the RRM (*RBFOX2[ΔRRM]*). Expression of *RBFOX2[ΔRRM]* did not attenuate the phenotype in either cell line, suggesting that the RRM of RBFOX2 is essential for its tumour suppressor activity in PDA progression (Fig. 2j–n and Extended Data Fig. 4l,m).

## RBFOX2 mediates focal adhesion

We next performed deep RNA sequencing of both metastatic human PDX-derived X50 cells with and without *RBFOX2* cDNA overexpression and primary tumour BxPC3 cells with and without two *RBFOX2* sgRNAs (sgRNA-1 or sgRNA-2). Evaluation of the changes in alternative splicing identified 474 significant differential splicing events in X50 cells (overexpression versus control), 458 significant differential splicing events in BxPC3 cells (sgRNA-1 versus control) and 488 events in BxPC3 cells (sgRNA-2 versus control), using a *P* value of less than 0.05 and |ΔPSI| greater than 10% (Extended Data Fig. 5a–c and Supplementary Table 6). To determine the essential RBFOX2 splicing targets that regulate the metastatic process in PDA cells, we identified 114 RBFOX2-dependent alternatively spliced events that are regulated reciprocally in each comparison, with *P* values of less than 0.05 and |ΔPSI| greater than 10% (Fig. 3a,b, Extended Data Fig. 6 and Supplementary Table 6). Analysis of these 114 alternative splicing events revealed that the main alternative splicing event types in this group are single-exon skipping (45%), alternative 5′ splice site (14%), alternative 3′ splice site (16%), mutually exclusive exons (7%), multiple exon skipping (9%) and intron retention (9%) (Fig. 3b). The RBFOX2 binding motif GCAUG was significantly overrepresented upstream of the 5′ splice site of the 114 RBFOX2-dependent alternative splicing events (Extended Data Fig. 5f and Supplementary Table 3). Out of 87 genes identified in the RNA-seq analysis of the manipulated cells, 35 genes had previously been reported as RBFOX2 splicing targets[18], and 37 genes were detected as differentially spliced in the analysis of patients with PDA (Fig. 1b, Extended Data Fig. 5d,e and Supplementary Table 7). Reactome analysis of the 87 RBFOX2-regulated genes showed enrichment of genes in the RHO GTPase (RHOA, CDC42 and RAC1) pathways (Fig. 3c and Supplementary Table 6). Genes in the RHO GTPase pathways were also enriched in the overlap between the patient samples and the merged genes (37) and the overlap between the known RBFOX2 targets and the merged genes (35) (Extended Data Fig. 5d,e and Supplementary Table 4). These pathways are well-known regulators of the actin cytoskeleton, cell polarity, microtubule dynamics and vesicle trafficking, supporting a biological role for RBFOX2 as a metastatic tumour suppressor in PDA.

Nascent focal adhesions have been shown to form during migration, apparently acting as mechanical anchor points that promote the actin polymerization-dependent protrusion at the leading edge[29]. Enhanced focal adhesion formation assembly can contribute to migration and invasion[30–32]. We examined whether RBFOX2 regulates focal adhesion formation. BxPC3 primary tumour cells expressing either of the two *RBFOX2* sgRNAs demonstrated increased focal adhesion formation on the cell periphery, as indicated by paxillin immunofluorescence. By contrast, X50 cells overexpressing RBFOX2 demonstrated reduced focal adhesion formation (Extended Data Fig. 7a–d). Using Cas9-expressing BxPC3 cells infected with *RBFOX2* EIJ sgRNA (Fig. 2a,h,i), we observed an induction of focal adhesion formation compared with control cells, similar to BxPC3 cells with *RBFOX2* sgRNA. Expression of *RBFOX2* cDNA in these cells rescued the phenotype, increasing focal adhesion formation compared with control cells (Extended Data Fig. 7e). No changes in focal adhesion formation were detected in either X50 or BxPC3 cells expressing RBFOX2(ΔRRM) (Extended Data Fig. 7f–g). These results confirm the contribution of the splicing activity of RBFOX2 to cytoskeleton organization and focal adhesion formation.

## Inhibition of the RHO GTPase pathway

To examine whether the regulation of the RHO GTPase pathways by RBFOX2 is critical for pancreatic cancer progression, we took advantage of two inhibitors of the RHO GTPase pathways: MBQ167, a dual RAC/CDC42 inhibitor[33], and azathioprine, a blocker of RAC1 activity and other small GTPases of the RAS superfamily[34,35]. MBQ167 administration to RBFOX2-depleted BxPC3 cells inhibited the migration rates

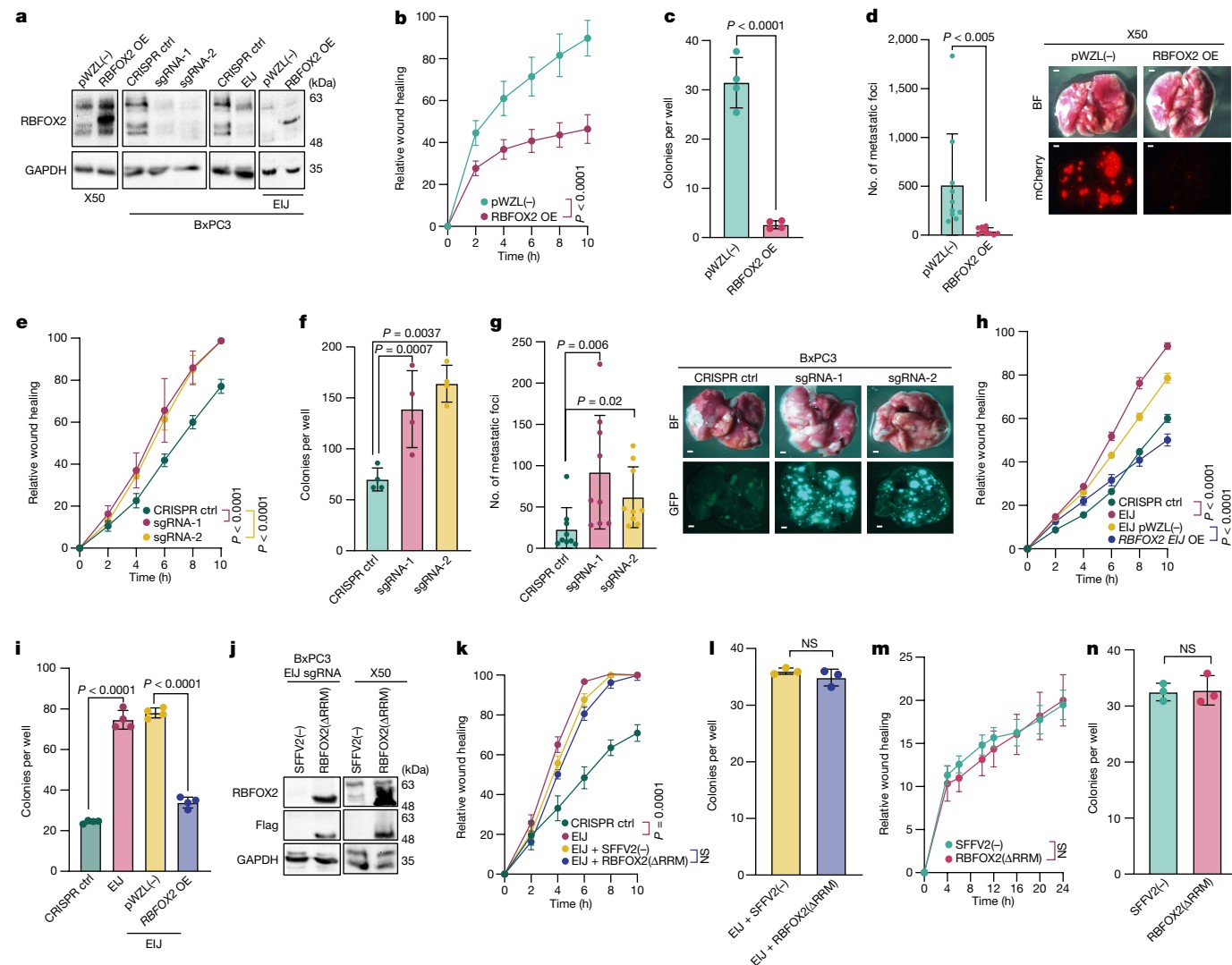

**Fig. 2 | RBFOX2 acts as a metastatic tumour suppressor in pancreatic cancer progression. a**, Immunoblot analysis of X50 cells transduced with pWZL(−) (empty vector control) or *RBFOX2* cDNA (RBFOX2 overexpression (OE)) (left), BxPC3 cells transduced with CRISPR control (ctrl), *RBFOX2* sgRNA-1 or *RBFOX2* sgRNA-2 (second from left), CRISPR control or *RBFOX2*-specific sgRNA targeting the *RBFOX2* exon−intron junction (EIJ) (third from left), and these cells transduced again with pWZL(−) or *RBFOX2* cDNA (right). **b,c**, Wound healing assay (**b**) and colony formation in soft agar assay (**c**) of X50 cells in **a**, left. **d**, Left, quantification of lung metastases in NOD-SCID mice injected intravenously with mCherry-labelled X50 cells expressing either pWZL(−) or RBFOX2 OE (*n* = 10 mice per group). Right, representative images of the lungs visualized by bright-field (BF) (top) and fluorescence (bottom) microscopy. Scale bars, 1,000 µm. **e,f**, Wound healing assay (**e**) and colony formation in soft agar assay (**f**) of cells described in **a**, second from left. **g**, Left, quantification of lung metastases in NOD-SCID mice injected intravenously with GFP-labelled BxPC3 cells expressing CRISPR control, *RBFOX2* sgRNA-1 or *RBFOX2* sgRNA-2 (*n* = 9 mice per group). Right, representative images of the lungs visualized by bright-field (top) and fluorescence (bottom) microscopy. Scale bars, 1,000 µm. **h,i**, Wound healing assay (**h**) and colony formation in soft agar assay (**i**) of cells described in **a**, third from left and **a**, right. **j**, Immunoblot analysis of BxPC3 cells expressing *RBFOX2* EIJ sgRNA transduced with SFFV2(−) (empty vector control) or RBFOX2(ΔRRM) (left) and X50 cells transduced with lentiviruses encoding either SFFV2(−) or RBFOX2(ΔRRM) (right). **k,l**, Wound healing assay (**k**) and colony formation in soft agar assay (**l**) of cells in **j**, left. **m,n**, Wound healing assay (**m**) and colony formation in soft agar assay (**n**) of cells in **j**, right. Gel source data in **a,j** are provided in Supplementary Fig. 2. Data are mean ± s.d. In all panels, *n* ≥ 3 independent experiments, exact *P* values are shown. Two-way ANOVA (**b,e,f,h,i,k,m**) or unpaired two-tailed Student's *t*-test (**c,d,g,l,n**).

without a change in survival or proliferation rates in vitro (Fig. 3d–f). In vivo administration of MBQ167 to mice injected intravenously with either GFP-labelled RBFOX2-depleted BxPC3 cells or GFP-labelled PDX-derived metastatic X50 cells inhibited lung metastases (Fig. 3g,h and Extended Data Fig. 8a,b), without an effect on subcutaneous primary tumour growth (Extended Data Fig. 8c–f). Similar results were obtained using azathioprine on the metastatic X50 cell line (Extended Data Fig. 8g–j). However, azathioprine treatment inhibited primary tumour growth of the primary tumour cell line BxPC3 (Extended Data Fig. 8k,l). To mimic the effect of these inhibitors, we genetically knocked out *RAC1* in RBFOX2-depleted BxPC3 cells (Extended Data Fig. 8m). Similar to treatment with the Rac1 inhibitors, these cells had slower

migration rates in vitro, without any change in proliferation, and formed a significantly reduced number of lung metastases in vivo (Extended Data Fig. 8n–p). Thus, azathioprine and MBQ167 are good candidates for potential therapeutic intervention in metastatic PDA. These results demonstrate that RBFOX2-depleted BxPC3 cells are more sensitive to the effect of RAC1 and RHO inhibitors or *RAC1* knockout.

## RBFOX2 splicing targets in PDA

One of the RBFOX2-regulated splicing events identified in our analysis is splicing of *MPRIP*. MPRIP was initially identified as a RHOA-binding protein and was found to bind F-actin and myosin phosphatase-targeting

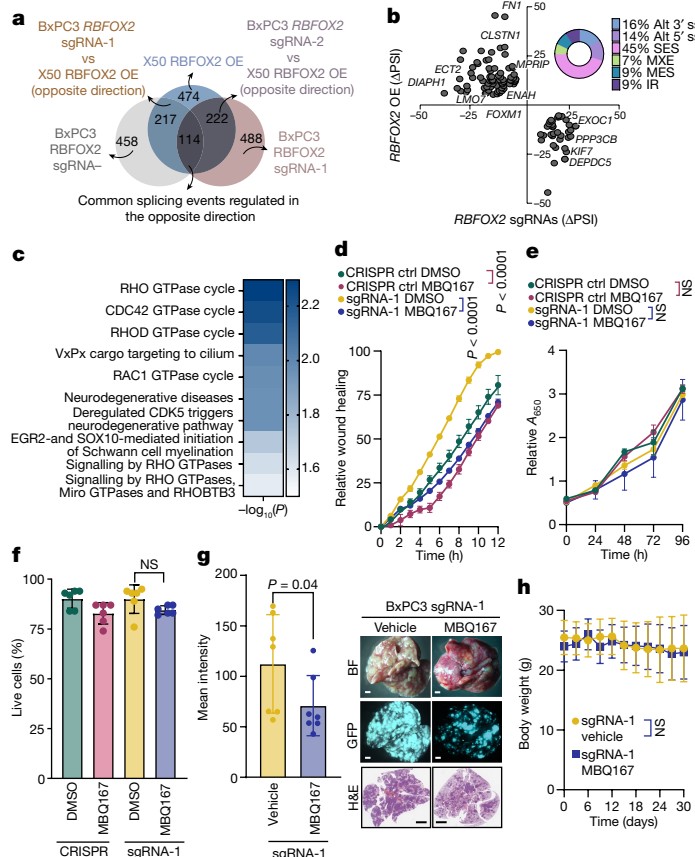

**Fig. 3 | RBFOX2 regulates splicing events in pancreatic cancer progression.**
**a**, Venn diagram showing the overlap of differentially spliced events, with
reciprocal splicing. Two comparisons are shown: X50 cells with RBFOX2 OE
versus BxPC3 cells expressing *RBFOX2* sgRNA-1, and X50 cells with RBFOX2
OE versus BxPC3 cells expressing *RBFOX2* sgRNA-2. **b**, ΔPSI of the reciprocal
splicing events described in **a**. The doughnut chart shows the distribution of
the types of alternatively spliced events: single-exon skipping (SES), alternative
5′ splice site (alt 5′ ss), alternative 3′ splice site (alt 3′ ss), mutually exclusive exon
(MXE), multiple exon skipping (MES) and intron retention (IR). *P* values were
calculated using PSI-sigma bioinformatic analysis (nominal *P* value < 0.05 and
|ΔPSI| > 10%) (details in Methods). **c**, Reactome pathway analysis for 114 shared
events identified in **a**. Genes included in the analysis had imposed cut-offs,
as described in Methods. Pathways were filtered for a statistical threshold of
*P* < 0.05 using an over-representation analysis (hypergeometric distribution)
test. **d**, Wound healing assay of BxPC3 RBFOX2 sgRNA-1 cells treated with
either MBQ167 (0.05 μM) or DMSO. **e**, Proliferation assay of cells in **d**. **f**, Trypan
blue cell count of cells described in **d** 24 h after treatment. **g**, Left, mean GFP
intensity in lungs from NOD-SCID mice injected intravenously with GFP-labelled
BxPC3 *RBFOX2* sgRNA-1 cells treated with either vehicle or MBQ167 (3 mg per kg)
(*n* = 7 mice per group). Right, representative images of the lungs visualized by
bright-field (top) and fluorescence (middle) microscopy, and haematoxylin
and eosin (H&E) staining (bottom). Scale bars, 1,000 μm. Full data are shown in
Supplementary Fig. 12. **h**, Weight of mice during the experiment in **g**. Data are
mean ± s.d. In all panels, *n* ≥ 3 independent experiments. Exact *P* values are
shown. Unpaired two-tailed Student's *t*-test (**f**), two-way ANOVA (**d**,**e**,**h**) or
unpaired one-tailed Student's *t*-test (**g**). Reactome analysis is provided in
Supplementary Table 4 and differentially spliced events in RBFOX2 manipulated
cell lines are listed in Supplementary Table 6.

---

subunit 1 (MYPT1). In addition, MPRIP has been reported to inhibit
RHOA via its RHO GTPase-activating protein activity, indicating an
inhibitory regulatory activity on the RHOA–ROCK pathway[36,37]. How-
ever, to our knowledge, there are no reports regarding the splicing iso-
forms of MPRIP and their role in metastasis. Overexpression of RBFOX2
induced the inclusion of exon 23 of *MPRIP*, whereas RBFOX2 knockout
induced skipping of this exon (Fig. 4a,b). The amount of skipped *MPRIP*
isoform was higher in metastatic PDA samples (lower PSI) compared
with primary tumours (Fig. 4c). Patients in the MPRIP-low group, based
on the median PSI value, had a worse survival outcome (52.34 months
shorter than patients in the high-PSI group) (Fig. 4d). These results
support the clinical relevance of this *MPRIP* splicing event in PDA pro-
gression. Thus, we predict that the skipped *MPRIP* isoform would be
associated with oncogenic activity in PDA progression.

We next tested whether modulation of *MPRIP* splicing could affect
the metastatic potential of PDA tumour cell lines. To induce skipping
of exon 23 of *MPRIP* we designed CRISPR sgRNAs against either the 3′
or 5′ splice sites of exon 23. Both of these sgRNAs triggered efficient
skipping of exon 23 (Fig. 4e). BxPC3 primary tumour cells expressing

these sgRNAs demonstrated increased migration capacity and colony
formation in soft agar compared with control cells, with no effect on
the proliferation rate (Fig. 4f,g and Extended Data Fig. 9a). GFP-labelled
BxPC3 primary tumour cells with either 3′ or 5′ splice site sgRNAs
injected intravenously into NOD-SCID mice resulted in a significant
increase in the number of lung metastatic foci compared with control
cells (Fig. 4h).

To force the inclusion of exon 23 of *MPRIP* in metastatic cells, we
designed a sgRNA that disrupts the RBFOX2 motif (GCAUG) located in
the 3′ untranslated region in exon 24. Interruption of the RBFOX2 motif
by CRISPR-based mutagenesis increased exon 23 inclusion (Fig. 4i). X50
metastatic cells harbouring sgRNA targeting the RBFOX2 motif (DS-24
*MPRIP* sgRNA) showed markedly slower migration ability and decreased
colony formation in soft agar without an effect on proliferation rates
(Fig. 4j,k and Extended Data Fig. 9b). To examine the effects of *MPRIP*
splicing modulation in vivo, we injected GFP-labelled X50 metastatic
cells expressing either CRISPR control or DS-24 *MPRIP* sgRNA intrave-
nously into NOD-SCID mice. Notably, the inclusion of *MPRIP* exon 23
in metastatic X50 cells inhibited metastatic potential in vivo (Fig. 4l).

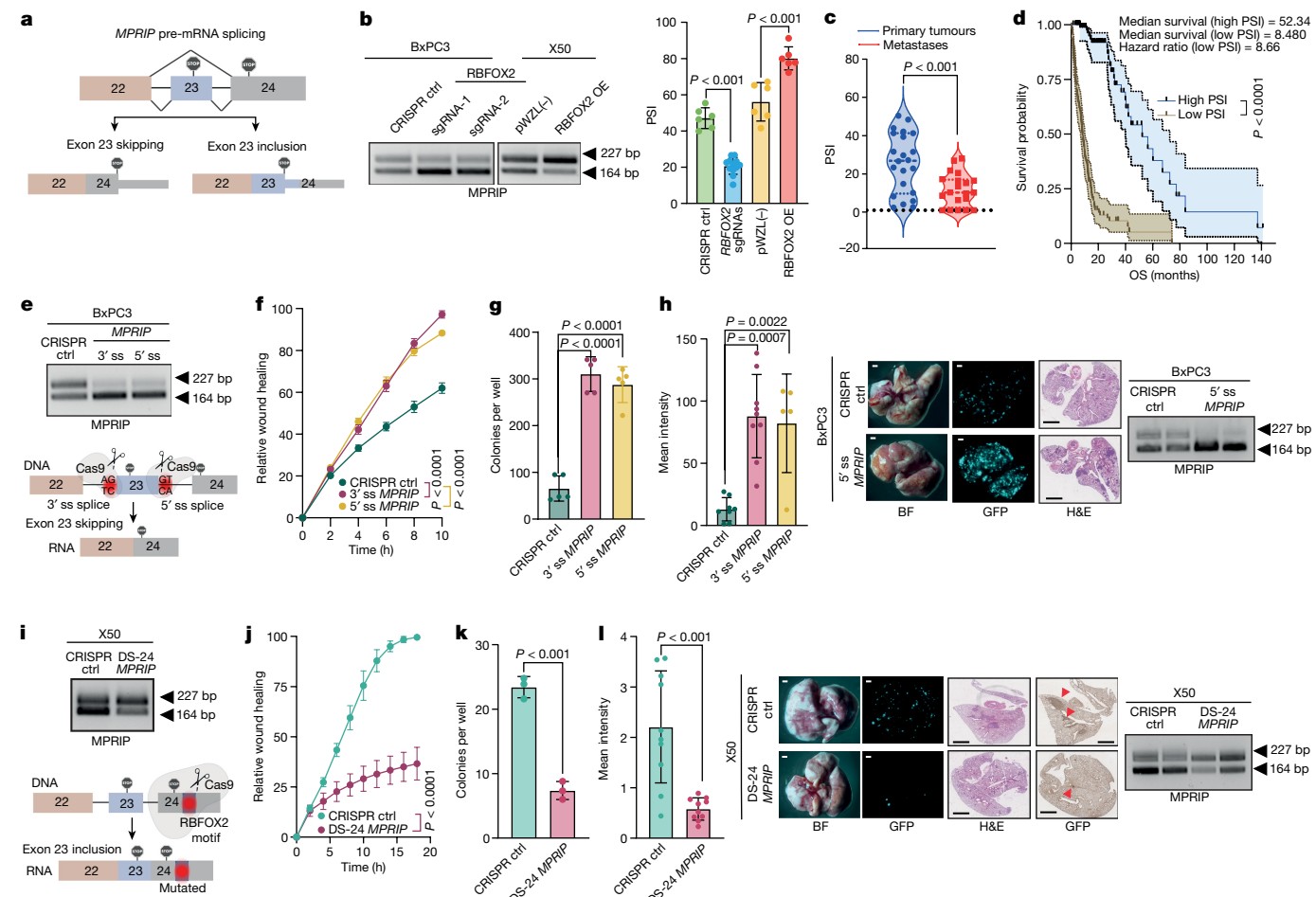

**Fig. 4 | Modulation of *MPRIP* splicing regulates the metastatic potential of pancreatic tumour cells. a**, Scheme of RBFOX2-mediated *MPRIP* splicing. **b**, PCR with reverse transcription (RT–PCR) and quantification of *MPRIP* PSI values in BxPC3 cells expressing CRISPR control, *RBFOX2* sgRNA-1 or *RBFOX2* sgRNA-2, and X50 cells expressing pWZL(−) or RBFOX2 OE. **c**, Violin plot of *MPRIP* PSI values in samples from patients with pancreatic cancer (*n* = 22 primary tumours and *n* = 20 metastases), analysed using LabChip GX. **d**, Kaplan–Meier curves for overall survival and *MPRIP* PSI values from the PanCurX dataset. **e**, Bottom, scheme showing modulation of MPRIP splicing (skipping). Top, RT–PCR analysis of *MPRIP* splicing in BxPC3 cells transduced with *MPRIP* sgRNA. **f,g**, Wound healing assay (**f**) and colony formation in soft agar assay (**g**) of cells in **e**. **h**, Left, mean GFP intensity in lungs from NOD-SCID mice injected intravenously with GFP-labelled BxPC3 cells expressing *MPRIP* sgRNA (*n* = 8 mice per group for CRISPR control and 3′ splice site *MPRIP* sgRNA, *n* = 5 mice per group for 5′ splice site *MPRIP* sgRNA). Centre, representative images of lungs visualized by visualized by bright-field (left) and fluorescence

(middle) microscopy, and haematoxylin and eosin staining (right) (scale bars, 1,000 μm). Right, RT–PCR analysis of RNA from two lungs from each group (right) from a repeated in vivo experiment (*n* = 4 mice for CRISPR control and *n* = 5 mice for 5′ splice site *MPRIP* sgRNA). **i**, Bottom, scheme of *MPRIP* splicing (inclusion) (bottom). Top, RT–PCR showing *MPRIP* splicing in X50 cells transduced with DS-24 *MPRIP* sgRNA. **j,k**, Wound healing assay (**j**) and colony formation in soft agar assay (**k**) of cells in **i**. **l**, Left, mean GFP intensity in lungs from NOD-SCID mice injected intravenously with GFP-labelled X50 metastatic cells expressing either CRISPR control or DS-24 *MPRIP* sgRNA (*n* = 10 mice per group). Centre, representative images of lungs. Scale bars, 1,000 μm. Right, RT–PCR analysis of RNA from two lungs from each group. Gel source data, images of the lungs and H&E staining are provided in Supplementary Figs. 3, 13 and 14. Data are mean ± s.d. In all panels, *n* ≥ 3 independent experiments. Exact *P* values are shown. Unpaired two-tailed Student's *t*-test (**b,c,h,k,l**), log-rank (Mantel–Cox) test (**d**) and two-way ANOVA (**f,g,j**). The schemes in **a,e,i** were created with Biorender.com.

We detected fewer focal adhesions in X50 metastatic cells expressing RBFOX2 motif sgRNA compared with cells with CRISPR control, and more focal adhesions in BxPC3 cells expressing either 3′ splice site or 5′ splice site sgRNA, compared with control cells (Extended Data Fig. 9c,d).

To identify the mechanism of action of the spliced isoform of MPRIP, we analysed the potential kinase repertoire predicted to phosphorylate each serine or threonine in the unique sequence of each MPRIP isoform using a serine-threonine kinome analysis prediction tool[38]. This analysis identified differing predicted phosphorylation sites in each of the MPRIP splicing isoforms (Extended Data Fig. 10a and Supplementary Table 8). Using the AlphaFold structure prediction tool[39], we established that the C terminus of the MPRIP isoform with the skipped exon 23 has an α-helical structure. This region is predicted to project outwards from the protein, in contrast to the shorter C terminus of the

isoform that includes exon 23 (Extended Data Fig. 10b). This α-helical structure might enable the interaction of the oncogenic MPRIP skipped isoform with different binding partners. Immunoprecipitation followed by mass spectrometry revealed enrichment for proteins in different pathways for each isoform. The exon 23-skipped isoform pulled-down proteins enriched in MAPK family signalling cascades, the RAF–MAP kinase cascade and cell cycle pathways. Both isoforms pulled down RHO GTPase effector pathway proteins, with the skipped isoform binding more proteins from this pathway (Extended Data Fig. 10c–g and Supplementary Table 9). Co-immunoprecipitation experiments recapitulated the binding of the MPRIP exon 23-skipped isoform to A-RAF (Extended Data Fig. 10h). This establishes a role for MPRIP splicing in altering cytoskeletal organization, signalling pathways and inducing metastases in a PDA metastasis model.

We investigated two additional RBFOX2 target genes that exhibited reciprocal splicing in the *RBFOX2*-knockout BxPC3 cells compared with RBFOX2-overexpressing X50 cells. Myosin light chain 6 (*MYL6*) exhibited increased exon 6 skipping in RBFOX2-depleted cells (Extended Data Fig. 11a). Calsyntenin (*CLSTN1*) showed increased exon 10 skipping in RBFOX2-depleted cells (Extended Data Fig. 12a). Patient-derived metastatic PDA samples had larger amounts of *MYL6* and *CLSTN1* skipped isoforms (lower PSI) compared with primary PDA samples, suggesting that *MYL6* exon 6 and *CLSTN1* exon 10 skipping contribute to the oncogenic characteristics of metastatic PDA cells (Extended Data Figs. 11b and 12b). We designed CRISPR sgRNAs against the 3′ and 5′ splice sites of these exons to induce their skipping in BxPC3 primary PDA cells. We observed that sgRNAs targeting either of these target exons increased the migration rate compared with control cells (Extended Data Figs. 11c,d and 12c,d). Intravenous injection of GFP-labelled BxPC3 primary tumour cells containing sgRNAs targeting the 3′ splice sites of *MYL6* or *CLSTN1* into NOD-SCID mice revealed a significant increase in the number of lung metastatic foci, compared with cells with a control sgRNA targeting *MYL6* (Extended Data Fig. 11g), with a slight, nonsignificant effect for *CLSTN1* (Extended Data Fig. 12g). Moreover, more focal adhesions were observed in BxPC3 cells expressing sgRNAs targeting the 3′ splice sites of *MYL6* or *CLSTN1* compared with control cells (Extended Data Figs. 11h and 12h). The migration ability of RBFOX2-overexpressing metastatic cells modulated for each target individually was similar to that of control metastatic cells. Epistasis experiments modulating all three targets simultaneously showed no additive effect (Extended Data Figs. 9e,f, 11e,f and 12e,f).

In summary, we report here a role for the splicing factor RBFOX2 as a metastatic suppressor in PDA and identified an RBFOX2-regulated alternative splicing signature in metastatic PDA. We detected enrichment for RHO GTPase pathway genes among the RBFOX2-regulated splicing targets and demonstrated their functional role in invasion by pancreatic cancer cells. Pharmacological manipulation of the RHO–RAC pathway or precise modulation of alternative splicing events in these pathways can alter the balance of oncogenic versus tumour suppressor isoforms and may have potential as therapeutic targets for PDA.

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

## Methods

### Patient samples and RNA-seq analysis

RNA-seq data from 395 patients with PDA was obtained from the University Health Network (Toronto), Sunnybrook Health Sciences Centre (Toronto), Kingston General Hospital (Kingston), McGill University (Montreal), Mayo Clinic (Rochester), Massachusetts General Hospital (Boston) and Sheba Medical Center (Tel Aviv) and has been described previously[1,12–14]. The samples were provided with informed patient consent, and approval for the study was obtained from the Institutional Review or Research Ethics Board of each site. Genetic alterations and clinical data of patients with PDA are provided in Supplementary Table 1.

RNA sequencing was performed as described previously[13]. In brief, RNA was isolated using PicoPure RNA Isolation Kit, treated with RNase-free DNase, and quantified using Qubit dsRNA High Sensitivity kit. The RNA quality was determined using both RNA Screen Tape Assay and the 2200 TapeStation Nucleic Acid System. RNA libraries were prepared using TruSeq RNA Access. Reads were aligned to the human reference genome (hg38) and transcriptome (Ensembl v84) using STAR v.2.5.3a[41].

### Principal component analysis

PCA was applied to the RNA-seq data of 395 PDA patient samples to inspect the alternative splicing and gene expression changes with the prcomp function in R. The samples were manually filtered to include only samples that were clustered according to their clinical annotation to reduce background noise. 63% of the samples (136 primary tumour samples and 113 metastatic samples) were utilized for further analysis.

### Alternative splicing and gene expression analysis of patient samples

Alternative splicing analysis was performed with PSI-Sigma (version 1.9c)[15] on the filtered samples (249 samples). Ensembl gene annotation (version 87) was used as the reference transcriptome. ΔPSI and $P$ value based on exon coordinates were used to identify significant splicing changes between the two clusters. Significant events were identified with imposed cut-offs |ΔPSI| > 10% and nominal $P$ value < 0.05. $P$ values were calculated using PSI-Sigma bioinformatic analysis. Supplementary Table 2. Based on the gene names, the pathway enrichment analysis was conducted using the Reactome database[42]. Gene sets were limited to between 5 and 500 genes, and pathways were filtered for a statistical threshold of $P$ value < 0.05 using over-representation analysis (hypergeometric distribution) test. Reactome analysis is provided in Supplementary Table 4. For gene expression analysis, reads were mapped to the human genome (hg38) using STAR (version 2.5.3a)[41]. Ensembl gene annotation (version 87) was used as the reference transcriptome. DESeq2 (version 3.16)[43] was used to estimate the significance of differential expression between the sample groups. Overall, gene expression changes were considered to be significant if they passed the false-discovery rate (FDR) threshold of <5%.

### Sequence motif enrichment analysis

The XSTREME package[44] was used to identify motifs in the 5' splice site sequences of the cassette exons with significant splicing changes. The 5' splice site sequence is defined as the 500 bp sequence downstream of the alternative exon (+1 to +500 positions). RBFOX2 motif (GCWUG) was manually added to the motif collection of RNA-binding proteins in the XSTREME database (Ray2013 Homo sapiens). The sequence motif enrichment analysis was performed for the following comparisons: (1) primary tumour samples versus metastatic tumour samples, (2) events identified in the RHO pathways, and (3) the reciprocal splicing changes between RBFOX2 knockout in BxPC3 cells and RBFOX2 overexpression in X50 cells. By default, XSTREME reports 6- to 15-mer motifs whose $E$-value ≤ 0.05. The program uses Fisher's exact test or the binomial test to determine the significance of each motif found. Sequence motif enrichment analysis is provided in Supplementary Table 3.

### PDA PDX generation in nude mice

Xenograft models were established from tumour samples at Sheba Medical Center, as described previously[21,45]. X50 PDX cell line was generated from pleural effusion of a patient with stage IV pancreatic adenocarcinoma, X139 PDX cell line was derived from a liver metastasis of patient with stage IV pancreatic adenocarcinoma, and X252 PDX cell line was generated from the primary pancreatic tumour of a patient with stage II pancreatic adenocarcinoma. All samples were obtained with approval of the patients at the Sheba Medical Center. In brief, core needle biopsies and pleural effusion from PDAC tumours were collected and implanted subcutaneously into NOD-SCID mice. Xenografts were propagated and serially passaged into new recipient mice. Tumour chunks were bio-banked in 90% serum + 10% DMSO for future experiments and cryopreserved in liquid nitrogen for DNA, RNA and protein extraction. PDX-derived cells were generated by tissue dissociation and cultured in RPMI-1640 supplemented with 1% L-glutamine and 10% fetal bovine serum (FBS) (Biological Industries). PDA PDX generation in nude mice was performed in accordance with the guidelines of Sheba Medical Center Institutional Animal Care and Use Committee (IACUC) (5539/13).

### Cell lines and tissue culture

The BxPC3, HEK293T, Phoenix-AMPHO and HEK293 cells lines were originally obtained from the American Type Culture Collection (ATCC). BxPC3 cell line was grown in RPMI-1640 supplemented with 10% FBS, 1% L-glutamine, and 1% penicillin-streptomycin. HEK293T, Phoenix-AMPHO and HEK293 cell lines were grown in DMEM supplemented with 10% FBS and 1% penicillin-streptomycin.

### Immunoblot analysis

For immunoblotting, cells were lysed in Laemmli buffer for 5 min at 95 °C, lysates were separated on 10% or 12% SDS–PAGE gels and transferred to PVDF membranes (Invitrogen). The list of antibodies used in this study and the corresponding dilutions are provided in Supplementary Table 12.

### mRNA stability assay

For mRNA half-life measurements, BxPC3 and X50 cells were treated with 10 µg ml$^{-1}$ actinomycin D for 0, 2, 4, 6 and 8 h. RBFOX2 and 18S rRNA mRNA levels were measured by RT–qPCR. A list of RT–qPCR primers used in this study is provided in Supplementary Table 11.

### Protein stability assay

For RBFOX2 protein stability assay, BxPC3 and X50 cells were treated with 10 µg ml$^{-1}$ cycloheximide for 0, 2, 4, 6 and 8 h. RBFOX2 protein levels were detected by Immunoblot analysis. The list of antibodies used in this study are provided in Supplementary Table 12.

### CRISPR–Cas9-directed mutations for knockout and splicing modulation

For CRISPR–Cas9 directed knockout, sgRNAs were designed using CHOPCHOP (version 3), a web tool for selecting target sites for CRISPR–Cas9[46]. BxPC3 and X252 cells were transduced with LentiCRISPR v2 vector (Addgene #52961) containing sgRNAs targeting *RBFOX2*. sgRNA targeting the exon–intron junction (*RBFOX2* EIJ sgRNA) was designed manually and confirmed by the CHOPCHOP tool.

For CRISPR–Cas9-directed splicing modulation, sgRNAs were designed using the CHOPCHOP platform to target the 3' and 5' end splice sites of the target exon in order to induce skipping of *MPRIP*, *MYL6* and *CLSTN1*. *MPRIP* exon 23 inclusion sgRNA was designed to target the RBFOX2 motif downstream of exon 24, which was designed manually and confirmed by the CHOPCHOP tool. Modulation of all three targets simultaneously was achieved using dual sgRNAs, targeting the 3' splice site target exons of *MYL6* and *MPRIP*, which were cloned into

lentiCRISPR v2-Blast (Addgene #52961). This plasmid was then transduced into X50 metastatic cells with *CLSTN1* 3′ splice site sgRNA[47]. A list of sgRNAs used in this study is provided in Supplementary Table 10.

## Lentiviral infection

Lentiviruses were produced by co-transfection of HEK293T cells with psPax2 (Addgene #12260), pMD2.G (Addgene #12259) and either Flag SFFV2 ΔRRM RBFOX2-puro (Twist Bioscience) or specific sgRNAs LentiCRISPRV2, using FuGENE-HD (Promega E2312) and OptiMEM (Gibco 51985-026). One day after transfection, the medium was replaced, and 48 h after transfection, viruses were collected and filtered through a 0.45 µm membrane. BxPc3, X50, and X252 cells were infected with the viruses with the addition of polybrene (10 µg ml$^{-1}$, Sigma 107689). Selection with puromycin (2 µg ml$^{-1}$, Sigma P8833) was initiated 2 days after infection. Immunoblot analysis was performed to confirm overexpression using antibodies against either Flag or RBFOX2. Validation of the CRISPR-induced skipping event was performed by RT–PCR using specific primers for targeted splicing events.

## Retroviral infection

Retroviruses were produced by co-transfection of Phoenix-AMPHO cells with pCL-Eco (Addgene #12371), pMD2.G (Addgene #12259) and Flag pWZL-RBFOX2-hygromycin (received from A. Krainer) using FuGENE-HD (Promega E2312) and OptiMEM (Gibco 51985-026). One day after transfection, the medium was replaced, and 48 h after transfection, viruses were collected and filtered through a 0.45 µm membrane. X50 and X139 cells were infected with the viruses with the addition of polybrene (10 µg ml$^{-1}$) (Sigma 107689). Selection with hygromycin (50 µg ml$^{-1}$, Merck 400052) was initiated 2 days after infection. Immunoblotting was performed to confirm overexpression using antibodies against either Flag or RBFOX2.

## Anchorage-independent growth

Six-well plates were coated with a bottom layer containing 2 ml agar mixture (culture media, 20% FBS, 1% agar). After the bottom layer solidified, 1 ml of top agar mixture (culture media, 20% FBS, 0.3% agar) containing the cells (2 × 10$^4$ cells per well of BxPC3 or X50 cells in triplicate) was added. After this layer had solidified, 2 ml of media (culture media, 10% FBS) was added. Plates were incubated for 10–21 days, colonies from ten different fields were counted, and the average number of colonies per well was calculated.

## Growth curves

For proliferation quantification, 2,000 cells were seeded per well in 96-well plates in 4 replicates, fixed, and stained with methylene blue. Cell density was determined every 24 h (up to 96 h) by absorbance of the methylene blue dye at 655 nm, measured on a plate reader (Bio-Rad) according to the manufacturer's instructions. For MBQ167 treatment, cells were treated with either 0.05 µM MBQ167 or DMSO for 24 h after seeding.

## Quantitative wound healing assay

We used an automated Incucyte wound maker tool to create precise, uniform cell-free zones in cell monolayers enabling real-time, automated measurement of cell migration with a tool to measure the density of the wound region relative to the density of the cell region. Migration was assessed by performing scratch wound healing assays using a real-time cell imaging system (IncuCyte Live cell). One-hundred thousand cells per well were plated onto an IncuCyte 96-Well ImageLock Plate (IncuCyte 4379) (*n* = 3 independent experiments, each with 8 replicates). Twenty-four hours later, the confluent monolayer of cells was washed in PBS and scratched with the IncuCyte: 96-pin wound-making tool (IncuCyte WoundMaker). Starvation media (basic culture media supplemented with 1% FCS) was applied. Plates were transferred to the IncuCyte live-cell imaging system, and the 96-well wound assay protocol was run on the software. X50, BxPC3, X252, and X139 cells were imaged at 1-h intervals for 24 h to monitor cell migration. For MBQ167 treatment, cells were treated with either 0.05 µM MBQ167 or DMSO for 24 hours. Analysis was performed using the Cell Migration Analysis software (IncuCyte 4400).

## Trypan blue exclusion survival assay

BxPc3 cells (1 × 10$^6$ cells) were seeded on 6 wells plate in triplicates. The following day cells were treated with either 0.05 µM MBQ167 or DMSO as a vehicle. After 24 h, cells were collected, including the floating dead cell fraction, and resuspended in HBSS. The percentage of dead cells was determined on a Bio-Rad cell counter using 0.4% trypan blue.

## RNA-seq analysis of RBFOX2-manipulated cells

mRNA was isolated using oligo-dT purification from three independent biological replicates from the following cell conditions: BxPC3 CRISPR control, BxPC3 with two different *RBFOX2* sgRNAs, X50 empty vector and X50 with RBFOX2 overexpression.

RNA-seq experiments were performed using NextSeq 2000 system (Illumina). Libraries were prepared using TruSeq RNA Sample preparation kit. More than 100 × 10$^6$ reads of 150 bp from each side (paired-end) per sample were generated. Reads were mapped to the human genome (hg38) using STAR (version 2.5.3a) with two-pass mode[41]. Alternative splicing analysis was performed using PSI-Sigma (version 1.9c)[15]. Ensembl gene annotation (version 87) was used as the reference transcriptome.

To identify significant reciprocal splicing changes, we compared the overlap between two comparisons: (1) RBFOX2 OE (versus pWZL(−)) and RBFOX2 sgRNA-1 (versus CRISPR control), 217 significant events, and (2) RBFOX2 OE (versus pWZL(−)) and RBFOX2 sgRNA-2 (versus CRISPR control), 222 significant events. We compared ΔPSI and *P* values based on exon coordinates. An exon must fit three criteria to be considered as 'oppositely spliced': (1) the exon has |ΔPSI| > 10% and nominal *P* value < 0.05 in either KO or OE condition, (2) ΔPSI direction is the opposite between KO and OE conditions, and (3) |ΔPSI| > 5% in both KO and OE conditions. *P* values were calculated using PSI-Sigma bioinformatic analysis. A list of differentially spliced events in RBFOX2 manipulated cells is provided in Supplementary Table 6. Pathway enrichment analysis was conducted by using Reactome database[42] based on the gene names of opposite splicing changes between KO and OE conditions. Gene sets were limited to between 5 and 500 genes, and pathways were filtered for a statistical threshold of *P* < 0.05 using over-representation analysis (hypergeometric distribution) test. Reactome analysis is provided in Supplementary Table 4.

## RT–PCR

Total RNA was isolated using TRI Reagent (Sigma T9424). For cDNA synthesis, 1 µg of total RNA was reverse transcribed to cDNA with iScript cDNA Synthesis kit (Bio-Rad 1708891). RT–PCR was conducted on 1 µl of cDNA using PCRBIO HS Taq Mix Red kit (BIOSYSTEMS PB10.23.02) to confirm splicing and splicing modulation by CRISPR–Cas9. PCR conditions were as described in the manufacturer's protocol. PCR products were separated either on a 2% agarose gel or using the LabChip GX microfluidics platform. The list of primers used in this study are provided in Supplementary Table 11.

## RT–qPCR

Total RNA was extracted with TRI Reagent (Sigma), and 1 µg of total RNA was reverse transcribed using iScript cDNA Synthesis kit (Bio-Rad 1708891). qPCR was performed on the cDNA using SYBR green (Applied Biosystems) and the Step one plus real-time PCR system (Applied Biosystems). Normalization was performed using 18S rRNA primers. Primers are listed in Supplementary Table 11.

## Immunofluorescence

BxPC3 cells (1 × 10$^4$ cells) were seeded in µ-Slide 8 well-ibiTreat, tissue culture-treated (IBIDI 80826) and incubated overnight. Cells were

fixed with 2% paraformaldehyde in PBS for 20 min, then permeabilized with 0.1% Triton X-100 for 20 min and blocked with BSA and 1% FBS for 10 min. Cells were stained with Paxillin antibody for 2 h (1:200 BD Transduction Laboratories 612405), then incubated with a secondary antibody for 1 h (1:100 Goat anti-Mouse Alexa Fluor 488 A-11029). Antifade Mounting Medium with DAPI (Vector laboratory H-1200) was used to stain the nuclei and mount the samples. Images were acquired using a spinning disk confocal microscope (Nikon) with 60× and 100× objectives. The fields of view were randomly chosen. Image analysis was performed using NIS Elements version 4.13 imaging software.

## Animal studies
Metastatic and tumour formation in vivo experiments were performed in accordance with the guidelines of IACUC at the Hebrew University (MD-15-14634-5). The study is in compliance with all the relevant ethical regulations. NOD-SCID mice (Jackson Laboratories, 0001303) were ordered at 6 weeks of age. The mice were housed under standard laboratory conditions in specific-pathogen-free cages in an animal room at constant temperature (19–23 °C) and regulated humidity under a 12 h:12 h light-dark cycle and received standard laboratory chow and water ad libitum. All mice entered the experiments at 8–12 weeks of age. Both male and female mice were used for the experiments.

## In vivo metastasis model
X50 mCherry or GFP-labelled or BxPC3 GFP-labelled cells ($1 \times 10^6$ cells) in PBS were injected intravenously into NOD-SCID mice. One month after injection, the lungs were removed and analysed for metastases. For azathioprine and MBQ167 treatments, mice were treated with either 10 mg kg$^{-1}$ of azathioprine or 3 mg kg$^{-1}$ of MBQ167 starting 1 week after intravenous injection of cells. Azathioprine and MBQ167 were injected intraperitoneally every 3 days for a total of 15 doses. The mice were closely monitored on a daily basis for any signs of disease. Mice were killed at 30 days post-injection (endpoint), or earlier if they failed to thrive, experienced a weight loss of greater than 10% of their total body weight, or showed any signs of infection, as per our IACUC. These limits were not exceeded in any of our experiments. Images were obtained using a fluorescent binocular microscope. Quantification of lung metastases was performed using NIS Elements version 4.13 imaging software.

## Tumour formation in mice
BxPC3 and X50 cells ($1 \times 10^6$ cells) in PBS were injected subcutaneously into NOD-SCID mice. For azathioprine and MBQ167 treatments, mice were treated either with 10 mg kg$^{-1}$ of azathioprine or 3 mg kg$^{-1}$ of MBQ167 starting 1 week after intravenous injection of cells. Azathioprine and MBQ167 were injected intraperitoneally every 3 days for a total of 15 doses. The mice were monitored for tumour volume and weight. All mice were sacrificed approximately 6 weeks after injection and before their tumour volume reached the maximum allowed limit of 2,000 mm$^3$ as permitted by our IACUC guideline. No experiments exceeded this limit.

## Histological analysis
Tissues were fixed in 4% formaldehyde for 16 h. Tissues were embedded in paraffin, and 5 μm sections were cut and mounted on slides. Slides were rehydrated through a series of xylene and ethanol washes, and antigen retrieval was performed in a 10 mM citrate buffer in a pressure cooker. Tissues were blocked in 2.5% normal horse serum blocking solution (Vector Laboratories S-2012) and subjected to staining with anti-GFP antibody (abcam ab6673). ImmPRESS HRP anti-goat IgG polymer (Vector Laboratories MP-7401) was used as a secondary antibody. ImmPACT DAB peroxidase substrate (Vector Laboratories SK-4105) was used as a substrate. Hematoxylin (Vector Laboratories H-3404) was used as a counterstain. Slides were imaged using Aperio Digital Pathology Slide Scanners (Leica Biosystems).

## Serine/threonine kinase predictions
The kinase predictions were based on experimental biochemical data of their substrate motifs. Synthetic peptide libraries, containing 198 peptide mixtures, that explored amino acid preference up to 5 residues N-terminal and C-terminal to the phosphorylated Ser/Thr to determine the optimal substrate sequence specificity for recombinant Ser/Thr kinases were utilized. In total, 303 kinases were profiled. Their motifs were quantified into position-specific scoring matrices (PSSMs) and then applied computationally to score phosphorylation sites based on their surrounding amino acid sequences. These PSSMs were ranked against each site to identify the most favourable kinases[48]. Serine/threonine kinome analysis is provided in Supplementary Table 8.

## Structural analysis
Structural illustrations were generated with PYMOL, using predicted models of MPRIP on alphaFold: AF-Q6WCQ1-F1-model_v3_1 (Exon 23 included) and AF-B9EGI2-F1-model_v3 (Exon 23 excluded)[49].

## Immunoprecipitation
HEK293 cells were transfected with 24 μg plasmid DNA per 100 mm plate of either WPRE(−) empty plasmid or Flag-MPRIP exon 23-skipped isoform, or Flag-MPRIP exon 23-included isoform plasmids. Cells were lysed 48 h later in CHAPS buffer. Protein concentrations were normalized, and the samples were brought up to the same volume in CHAPS lysis buffer. A total of 50 μl of total lysate was set aside for immunoblotting. The remaining lysate was incubated overnight with 1 μg of anti-Flag M2 affinity gel (Sigma, A2220). The resin was washed 4 times with wash buffer, incubated with 50 μl of 2× Laemmli buffer, heated at 95 °C for 5 min, and separated by SDS–PAGE.

## Mass spectrometry analysis
**Sample preparation.** The immunoprecipitation beads were resuspended in 100 μl of 8 M urea, 10 mM DTT, 25 mM Tris-HCl pH 8.0 and incubated for 30 min, followed by addition of iodoacetamide to a concentration of 55 mm and incubated for 30 min in the dark. The urea was diluted by the addition of 7 volumes of 25 mM Tris-HCl pH 8.0. Trypsin (0.4 μg) was added (Promega), and the beads were incubated overnight at 37 °C with gentle agitation. The released peptides were desalted by loading the whole bead supernatant on C18 stage tips[50].

**LC–MS/MS analysis.** Mass spectrometry analysis was performed using a Q Exactive Plus mass spectrometer (Thermo Fisher Scientific) coupled online to a nanoflow UHPLC instrument (Ultimate 3000 Dionex, Thermo Fisher Scientific). Eluted peptides were separated over a 120-min gradient run at a flow rate of 0.15–0.3 μl min$^{-1}$ on a reverse phase 25-cm-long C18 column (75 μm internal diameter, 2 μm, 100 Å, Thermo PepMap RSLC, from Thermo Fisher Scientific). The survey scans (380–2,000 $m/z$, target value $3 \times 10^6$ charges, maximum ion injection times 50 ms) were acquired and followed by higher energy collisional dissociation-based fragmentation (normalized collision energy 25). A resolution of 70,000 was used for survey scans, and up to 15 dynamically chosen most abundant precursor ions were fragmented (isolation window 1.6 $m/z$). The MS/MS scans were acquired at a resolution of 17,500 (target value $10^5$ charges, maximum ion injection times 120 ms).

**Mass spectrometry data analysis.** Mass spectra data were processed using the MaxQuant computational platform, version 2.0.3.0[51]. Peak lists were searched against the *Homo sapiens* Uniprot FASTA sequence database UP000005640 appended with MPRIP isoforms. The search included cysteine carbamidomethylation as a fixed modification and oxidation of methionine as variable modifications. Match between runs option was selected. Peptides with a minimum of seven amino acid lengths were considered, and the required FDR was set to 1% at the

peptide and protein levels. Relative protein quantification in MaxQuant was performed using the label-free quantification (LFQ) algorithm[51]. MaxLFQ allows accurate proteome-wide label-free quantification by delayed normalization and maximal peptide ratio extraction[52]. Statistical analysis was performed using the Perseus statistical package, Perseus computational platform for comprehensive analysis of proteomics data[53]. Only those proteins for which at least two valid LFQ values were obtained in at least one sample group were accepted for statistical analysis. After application of this filter, a random value was substituted for proteins for which LFQ could not be determined ('imputation' function of Perseus) using default parameters. Volcano plot and $t$-tests were obtained using the following parameters in the volcano plot Perseus function: Randomization: 250, FDR: 0.05, S0: 0.1. Mass spectrometry analysis is provided in Supplementary Table 9.

## Statistical analysis

Tables and graphs for statistical analysis were created using GraphPad Prism 9 (GraphPad Software). $P$ values < 0.05 were considered significant. Statistical significance between two groups was determined by one- or two-tailed Student's $t$-test, and for experiments with more than two groups was determined by one- or two-way ANOVA, details of statistical analyses are indicated in figure legends. All the data in the graphs are shown as mean ± s.d. unless stated otherwise. All experiments were performed a minimum of three independent times with similar results, each containing at least three technical replicates (wells) for each condition.

## Reporting summary

Further information on research design is available in the Nature Portfolio Reporting Summary linked to this article.

## Data availability

RNA-seq data generated as part of this study were deposited to the BioProject under accession number PRJNA797585. Raw data of the RNA-seq patient samples are available at European Genome Phenome Archive (https://www.ebi.ac.uk/ega/home) (study ID EGAS00001002543), with databases IDs EGAD00001003584, EGAD00001004548, EGAD00001005799 and EGAD00001006081. Source data are provided with this paper.

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

**Acknowledgements** The authors thank S. Winograd-Katz and B. Geiger for fruitful discussions; Y. Cohen, A. Mogilevsky, A. Prabhu and A. Elhaj for assistance with in vitro and in vivo experiment validations. The authors also acknowledge support for A.J. from the Israel Ministry of Science, Zvi Yanai Ph.D. and post-doctoral fellowship programme for outstanding minority students. This research study was further supported by the Alex U. Soyka Pancreatic Cancer Research grant (CFHU), the Israel Cancer Association grant number 20220038, ISF grant 1510/17 (to R.K.) and Binational Science Foundation (BSF) grant number 2021108 (to R.K. and A.R.K.). K.-T.L. and A.R.K. acknowledge support provided by the NCI grant CA13106.

**Author contributions** The study was designed and all data were analysed by A.J. and R.K. A.J. conducted the majority of the experiments, aided by A.A.-S., H.S. and O.O. A.J. and A.A.-S. performed immunoprecipitation experiments. A.J. and H.S. conducted lentiviral and retroviral infections. A.J. and O.O. performed splicing validation for RNA-seq analysis and protein stability assay. E.E. and H.S. conducted the histological analysis. A.J., K.T.-L. and M.D.-G. performed computational analyses in consultation with A.R.K. and E.Y.L. The mouse experiments were carried out by A.J., R.K. and M.M. C.S., M.R.-G, and T.G. established PDXs and G.H.J. and S.G. provided human PDX RNA-seq data. J.L.J. and T.M.Y. performed kinome and AlphaFold analyses in consultation with L.C.C. The manuscript was prepared and edited by A.J., Z.S., E.E. and R.K., with contributions from all co-authors.

**Competing interests** The authors declare no competing interests.

**Additional information**
**Correspondence and requests for materials** should be addressed to Rotem Karni.

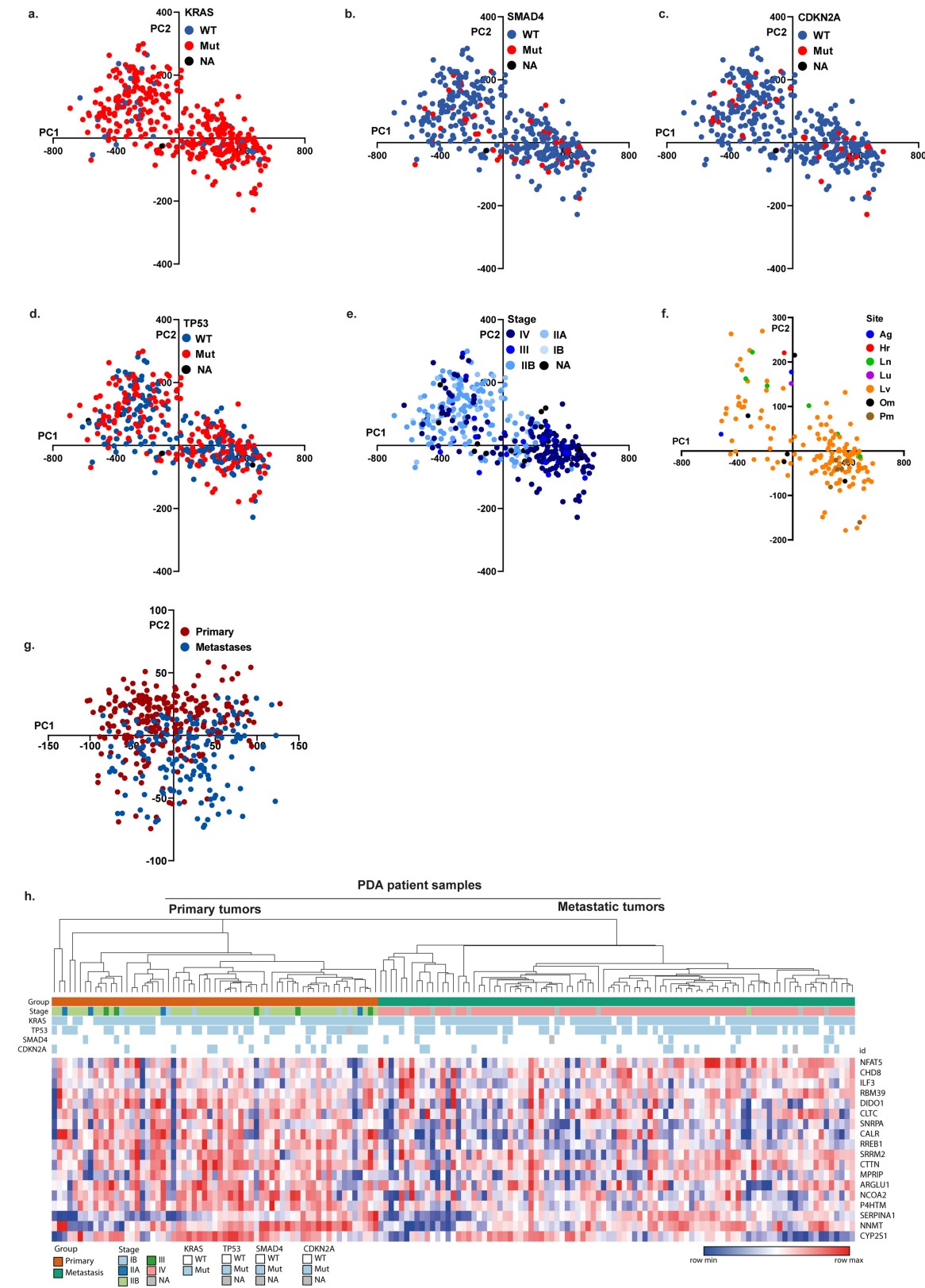

**Extended Data Fig. 1** | See next page for caption.

**Extended Data Fig. 1 | PCA analysis of splicing and gene expression changes in pancreatic cancer patient samples based on driver mutation status, stage, and tumor site. a**–**e**. PCA analysis of alternative splicing events in 395 pancreatic cancer patient samples computed by PSI scores. Each color-coded point represents a sample based on: *KRAS* mutation status (blue: wild-type, red: mutant, black: data not available) (a); *SMAD4* mutation status (blue: wild-type, red: mutant, black: data not available) (b); *CDKN2A* mutation status (blue: wild-type, red: mutant, black: data not available) (c); *TP53* mutation status (blue: wild-type, red: mutant, black: data not available) (d); and clinical tumor stage (e). **f**. PCA analysis of alternative splicing in 166 metastatic pancreatic cancer patient samples, computed by PSI scores. Each color-coded point represents a sample based on the location of the metastasis. Adrenal gland (Ag), heart (Hr), lymph node (Ln), lung (Lu), liver (Lv), omentum (Om), and peritoneum (Pm). **g**. PCA analysis of gene expression in 155 pancreatic cancer patient samples. Each color-coded point represents a sample according to its tumor origin; red: primary, blue: metastasis. **h**. Unsupervised clustering of gene expression in 155 PDA patient samples, based on the top 20 differentially spliced events (PSI) (18 genes) that were identified in Fig. 1b. Supplementary Tables 1-2 and 5.

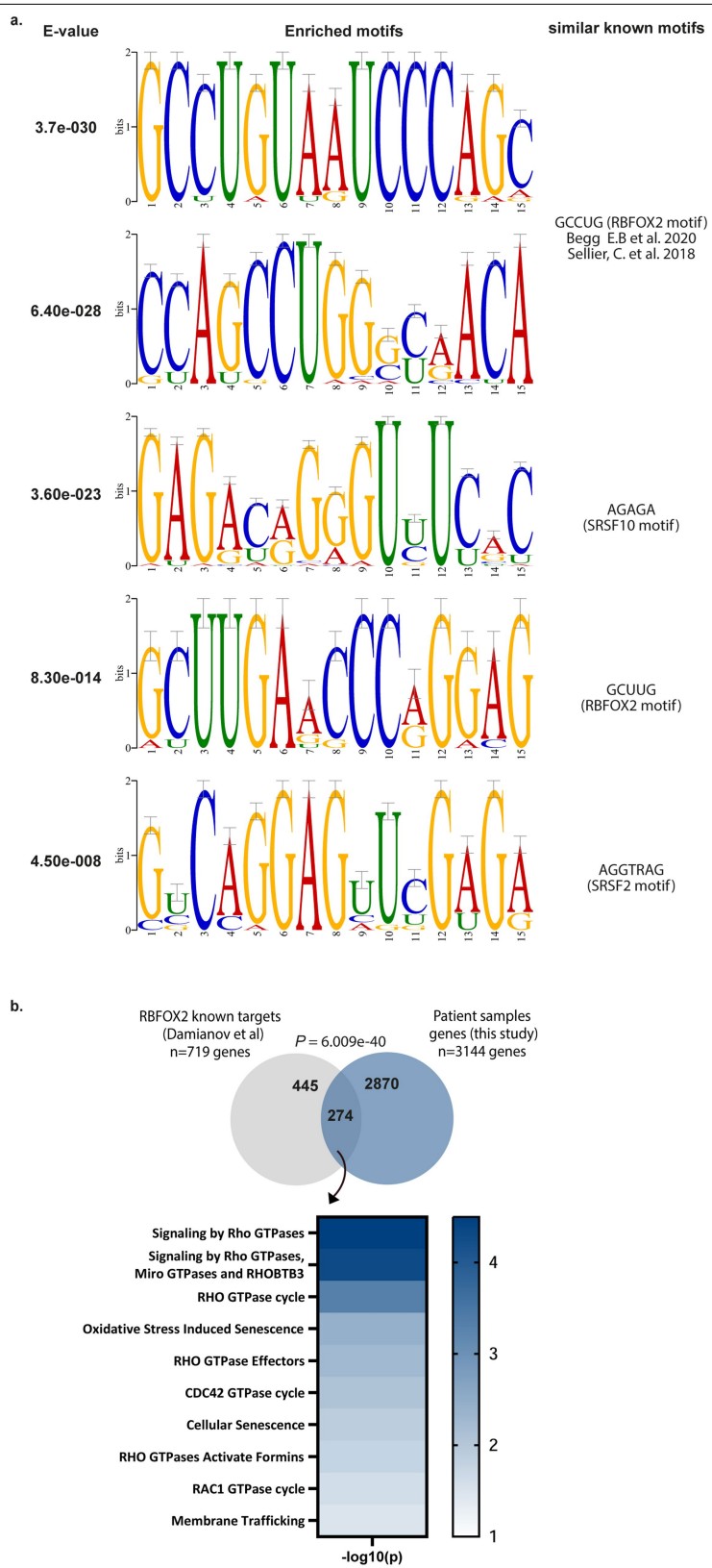

**Extended Data Fig. 2 | Motif enrichment and Reactome analysis of differentially spliced events identified in the primary versus metastatic PDA patient samples. a**. Top 5 enriched motifs identified in the 5′ ss of the differentially spliced events shown in Supplementary Table 3. **b**. The intersection of RBFOX2 known targets[18] and differentially spliced genes obtained from the comparisons in Supplementary Table 7. Statistical analysis was performed using Normal approximation test, and the exact *p*-values are shown. Reactome pathway analysis of the intersection of the differentially spliced genes with known RBFOX2 targets (bottom). Supplementary Table 4. Gene sets were limited to between 5 and 500 genes, and pathways were filtered for a statistical threshold of *P* < 0.05 using over-representation analysis (hypergeometric distribution) test.

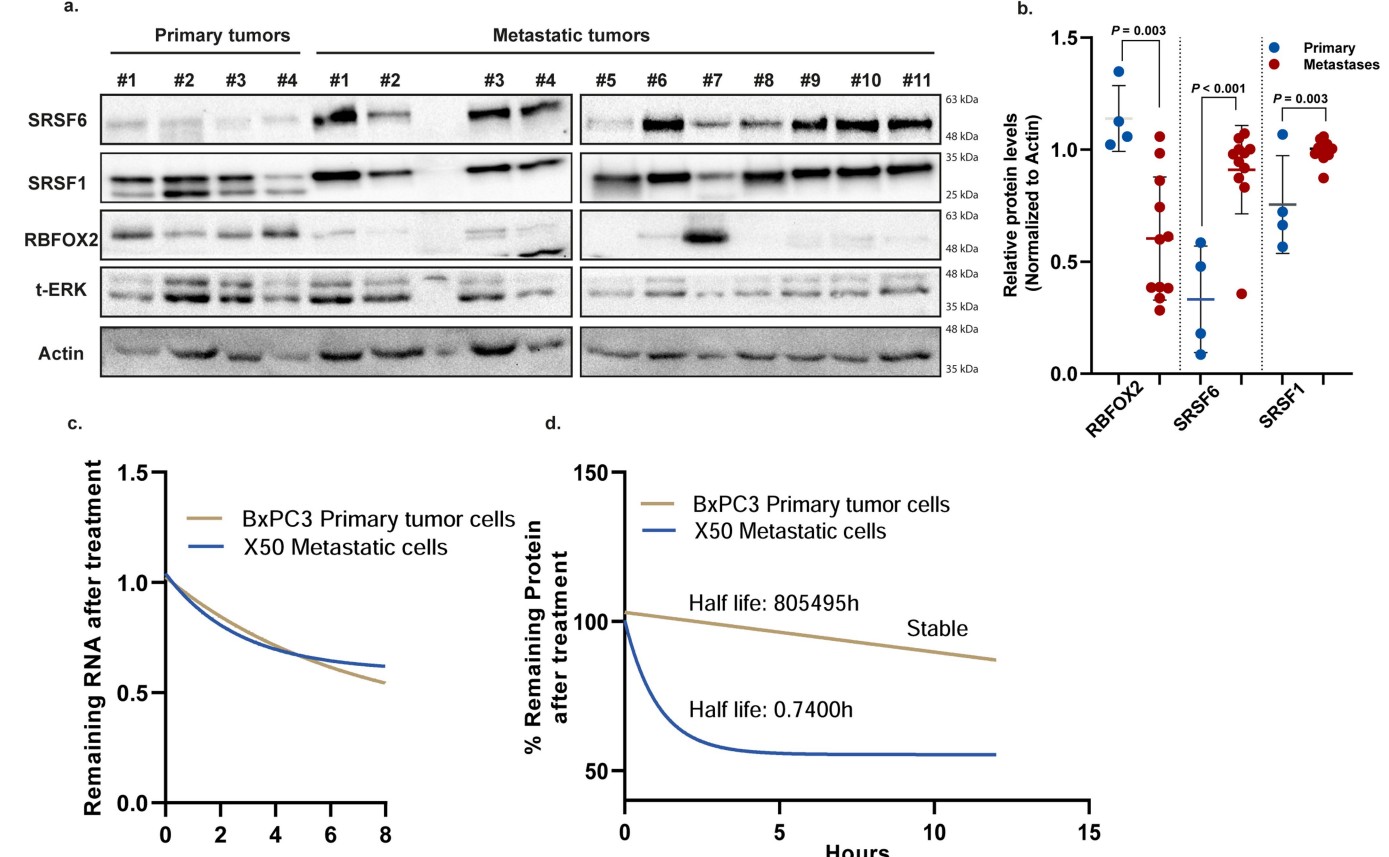

**Extended Data Fig. 3 | Decreased RBFOX2 protein levels in PDA patient metastatic samples. a**. Immunoblot analysis of representative primary and metastatic (PDX) pancreatic tumors. Total ERK (t-ERK) and actin serve as loading controls. Gel source data is provided in Supplementary Fig. 4. **b**. Quantification of immunoblot analysis of primary and metastatic (PDX) pancreatic tumors. n = 4 primary tumors, n = 11 metastases. Actin was used for normalization. Data are mean ± SD. Statistical analysis was performed using unpaired two-tailed Student's t-test; exact *p*-values are shown. **c**. RT-qPCR of RBFOX2 mRNA in BxPC3 primary tumor cells and X50 metastatic cells treated with actinomycin D (10 μg/ml) at different time points after treatment. **d**. Quantification of protein levels of RBFOX2 in BxPC3 primary tumor cells and X50 metastatic cells after treatment with cycloheximide (10 μg/ml) at different time points after treatment. n = 3 independent experiments (c, d). Statistical analysis was performed using non-linear regression (one phase decay) (c, d).

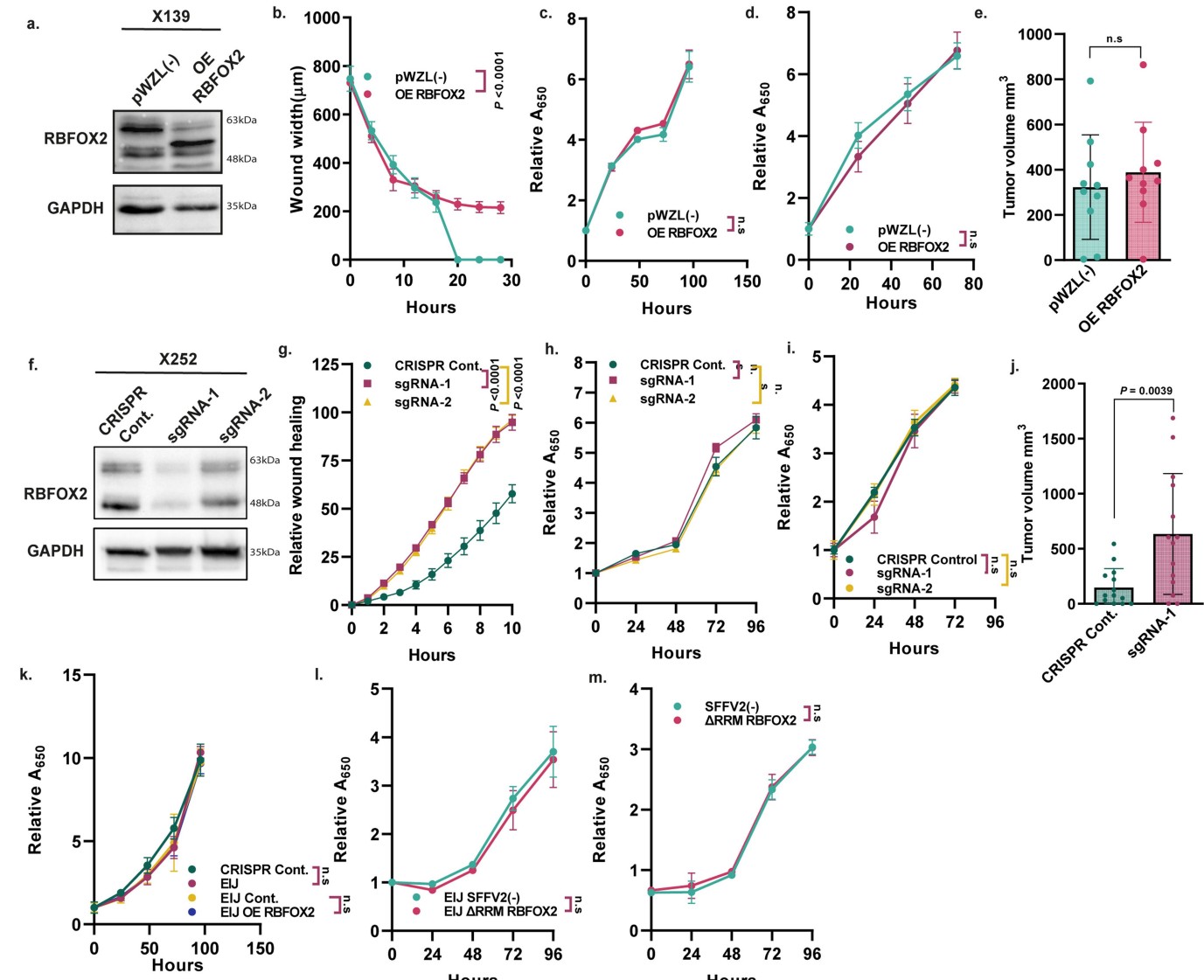

**Extended Data Fig. 4 | RBFOX2 acts as a metastatic tumor suppressor in pancreatic cancer progression. a.** Immunoblot analysis of metastatic cell line X139 (PDX-derived) transduced with either empty vector pWZL(−) or RBFOX2 (OE RBFOX2). **b.** Quantification of wound healing assay of cells described in (a). **c.** Proliferation assay of cells described in (a). **d.** Proliferation assay of metastatic cell line X50 (PDX-derived) transduced with retroviruses expressing either empty vector pWZL(−) or RBFOX2 cDNA (OE RBFOX2). **e.** Tumor volumes of tumors formed in NOD-SCID mice injected subcutaneously with X50 cells described in (a) (n = 5 mice per group, 2 tumors/mouse). **f.** Immunoblot analysis of primary pancreatic tumor cell line X252 transduced with lentivirus encoding either empty CRISPR vector (CRISPR Cont.) or two different RBFOX2 specific sgRNAs (sgRNA-1, sgRNA-2). **g.** Quantification of wound healing assay of cells described in (f). **h.** Proliferation assay of cells described in (f). **i.** Proliferation assay of primary tumor cell line BxPC3 transduced with lentiviruses encoding either empty CRISPR vector (CRISPR Cont.) or two different RBFOX2 specific sgRNAs (sgRNA-1, sgRNA-2). **j.** Tumor volumes of tumors formed in NOD-SCID

mice injected subcutaneously with BxPC3 cells expressing either empty vector (CRISPR Cont.) or RBFOX2 specific sgRNA (sgRNA-1) (n = 7 mice per group, 2 tumors/mouse). **k.** Proliferation assay of primary tumor cell line BxPC3 transduced with lentiviruses encoding either empty CRISPR vector (CRISPR Cont.) or RBFOX2 specific sgRNA targeting endogenous RBFOX2 exon-intron junction (EIJ sgRNA) and BxPC3 cells with EIJ sgRNA transduced with retroviruses encoding either empty vector pWZL(−) or OE RBFOX2. **l.** Proliferation assay of RBFOX2 EIJ sgRNA-expressing BxPC3 cells (BxPC3 EIJ sgRNA) transduced with lentiviruses encoding either SFFV2(−) empty vector or ΔRRM RBFOX2. **m.** Proliferation assay of X50 cells transduced with lentiviruses encoding either SFFV2(−) empty vector or ΔRRM RBFOX2. Gel source data (h, k) are provided in Supplementary Fig. 5. Data are mean ± SD. For all panels n≥3 independent experiments, $p$-values are shown, n.s. non-significant. Statistical analysis was performed using unpaired two-tailed Student's t-test (e, j) and two-way ANOVA test (b–d, g–i, k–m).

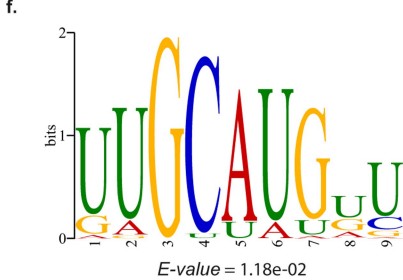

**Extended Data Fig. 5 | RBFOX2 regulates alternative splicing events in pancreatic cancer progression. a.** Volcano plot showing delta PSI of splicing changes in metastatic cell line X50 (PDX-derived) transduced with empty vector pWZL(−) compared to cells transduced with OE RBFOX2 **b.** Volcano plot showing delta PSI of splicing changes in primary tumor cell line BxPC3 transduced with lentivirus encoding empty CRISPR vector (CRISPR Cont.) compared to cells transduced with RBFOX2 sgRNA-1. **c.** Volcano plot showing delta PSI of splicing changes in primary tumor cell line BxPC3 transduced with lentivirus encoding empty CRISPR vector (CRISPR Cont.) compared to cells transduced with RBFOX2 sgRNA-2. *p*-values were calculated using PSI-Sigma bioinformatic analysis (nominal *p*-value < 0.05 and |ΔPSI| > 10%) (a-c) (more details described in the methods section). **d.** Intersection of the differentially spliced genes that were identified in the patient's samples from Supplementary Table 2 and the merged genes that were found in the comparison of X50 OE RBFOX2 versus BxPC3 sgRNAs (see Fig. 3a). Statistical analysis was performed

using Normal approximation test, and the exact *p*-values are shown (top). Reactome pathway analysis of the intersection of differentially spliced genes (bottom). Gene sets were limited to between 5 and 500 genes, and pathways were filtered for a statistical threshold of *p* < 0.05 using over-representation analysis (hypergeometric distribution) test. **e.** Intersection of RBFOX2 known targets[18] and the merged genes that were found in the comparison of X50 OE RBFOX2 versus BxPC3 sgRNAs (see Fig. 3a). Statistical analysis was performed using Normal approximation test, and the exact *p*-values are shown (top). Reactome pathway analysis of the intersection of differentially spliced genes (bottom). Gene sets were limited to between 5 and 500 genes, and pathways were filtered for a statistical threshold of *P* < 0.05 using over-representation analysis (hypergeometric distribution) test. **f.** Enriched motif identified in the 5′ ss of the differentially spliced events shown in Fig. 3b using XSTREME package (more details described in the methods section). Supplementary Tables 3-4 and 6-7.

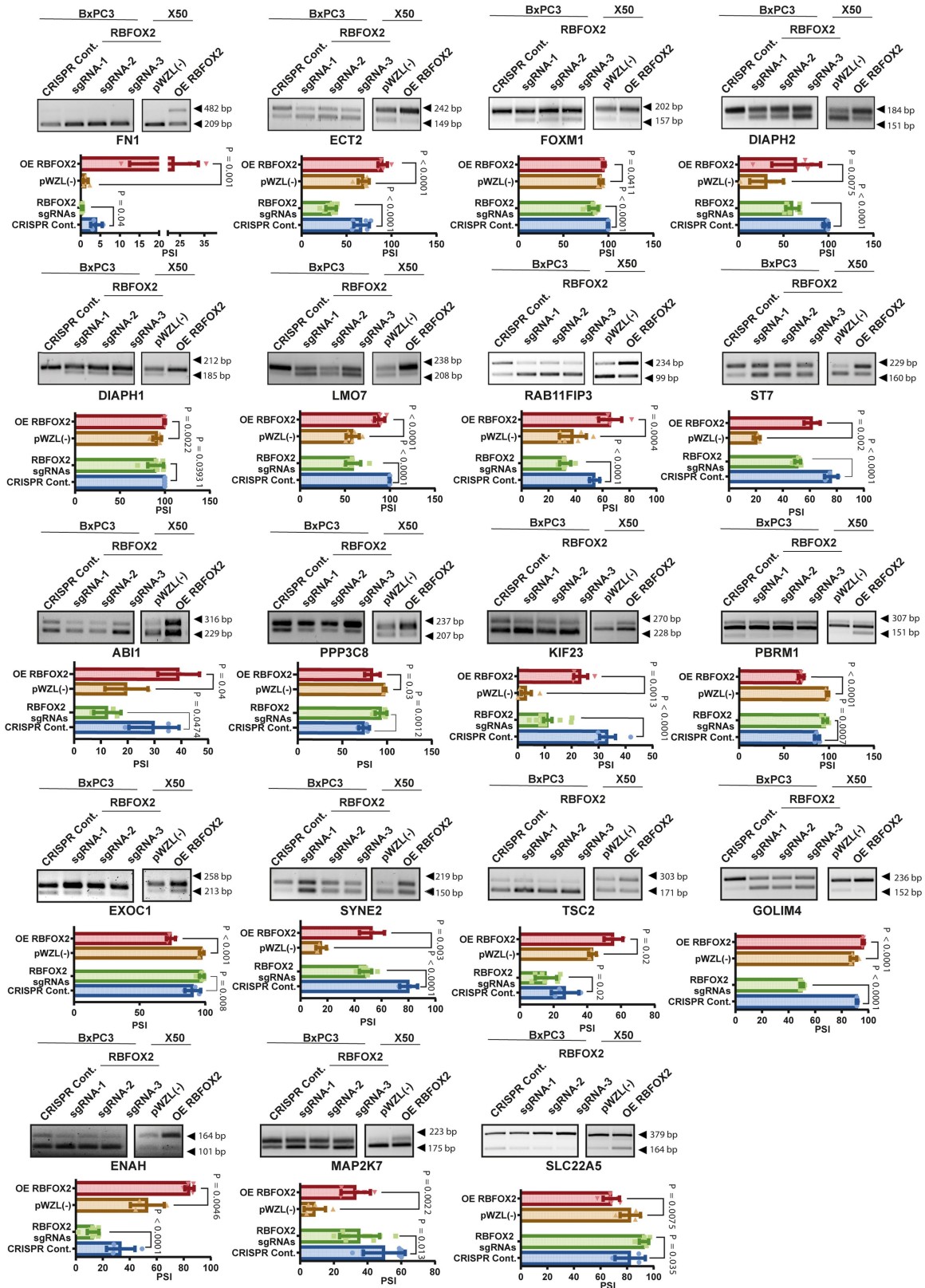

**Extended Data Fig. 6 | Validation of RBFOX2 splicing targets.** RT-PCR and quantitation of alternative splicing of RBFOX2 targets in BxPC3 cells expressing RBFOX2 sgRNAs or X50 cells expressing OE RBFOX2 and their respective controls. Primers are specific to regions upstream and downstream of the alternatively spliced exons (Supplementary Table 11). The percent spliced-in (PSI) was quantified using the Image Lab platform. Data are mean ± SD. For all panels n = 3 independent experiments, exact p-values are shown. Statistical analysis was performed using unpaired two-tailed Student's t-test. Gel source data is provided in Supplementary Fig. 6.

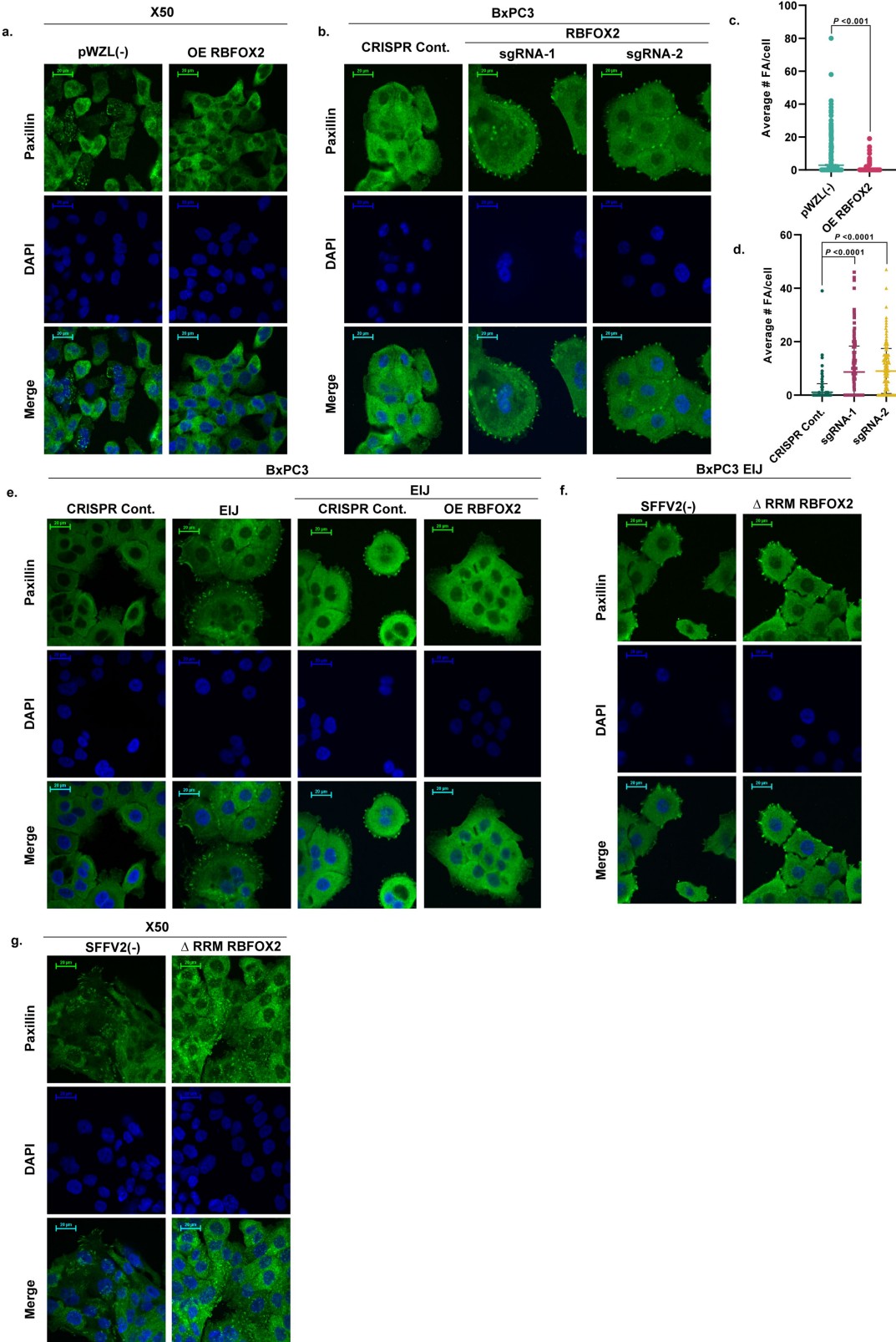

**Extended Data Fig. 7 | RBFOX2 modulates cytoskeleton organization in pancreatic tumor cells. a**. Paxillin immunofluorescence (IF) of X50 cells expressing either empty vector pWZL (−) or OE RBFOX2. **b**. Paxillin IF of BxPC3 cells transduced with either CRISPR Cont. or RBFOX2 sgRNAs (sgRNA-1, sgRNA-2). **c**, **d**. Quantification of the IF (a and b, respectively). Data are mean ± SD. The exact *p*-values are shown. Statistical analysis was performed using unpaired two-tailed Student's t-test. **e**. Paxillin IF of BxPC3 cells transduced with lentivirus encoding either empty CRISPR vector

(CRISPR Cont.) or RBFOX2 specific sgRNA targeting RBFOX2 exon-intron junction (EIJ, which silences endogenous RBFOX2 but not exogenous RBFOX2 cDNA), and same cells transduced again with either empty vector pWZL(−) (EIJ Cont.) or RBFOX2 cDNA (EIJ OE RBFOX2). **f**. Paxillin IF of BxPC3 EIJ sgRNA cells transduced with either empty vector (SFFV2(−)) or ΔRRM RBFOX2 cDNA. **g**. Paxillin IF of X50 cells transduced with either empty vector (SFFV2(−)) or ΔRRM RBFOX2 cDNA. For all panels, paxillin: green, DAPI: blue. Scale bars, 20 µm. For all panels, n = 3 independent experiments.

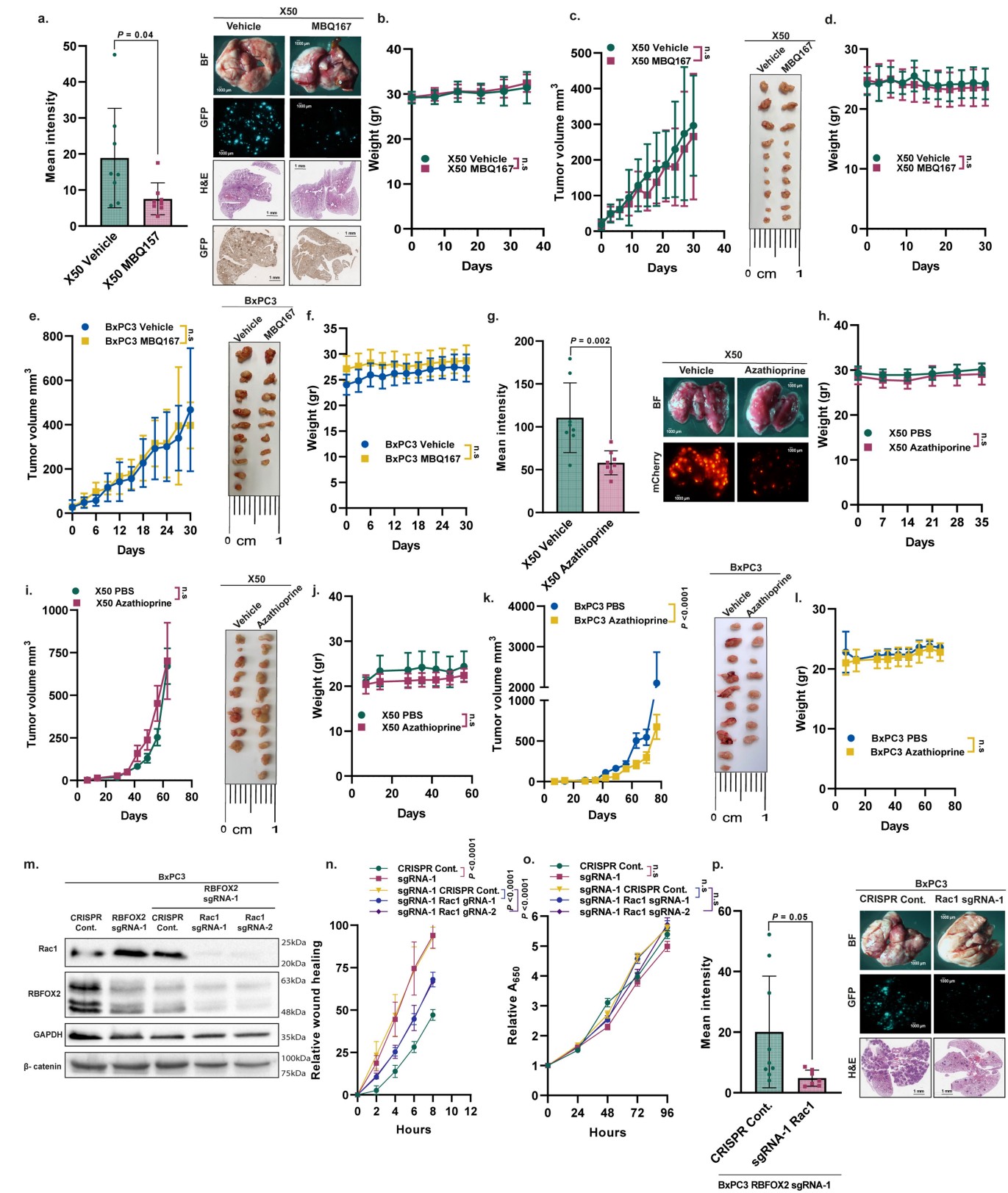

**Extended Data Fig. 8** | See next page for caption.

**Extended Data Fig. 8 | Modulation of Rho GTPase pathways reverse RBFOX2 knock-out effect on PDA cells. a**. Mean GFP intensity of lungs from NOD-SCID mice injected intravenously with GFP-labeled X50 cells treated with either vehicle or MBQ167 3 mg/Kg (n = 8 mice/group) (left). Representative pictures of the lungs visualized by fluorescent microscopy (scale bar 1000 μm) and H&E and GFP staining (scale bar 1 mm) (right). **b**. Weight of mice throughout the experiment described in (a). **c**. Tumor volumes of tumors formed in NOD-SCID mice injected subcutaneously with X50 cells and treated with either vehicle or MBQ167 3 mg/Kg (n = 5 mice/group) (left). Representative pictures of the tumors (right) (scale bar 1 cm). **d**. Weight of the mice throughout the experiment described in (c). **e**. Tumor volumes of tumors formed in NOD-SCID mice injected subcutaneously with BxPC3 cells and treated with either vehicle or MBQ167 3 mg/Kg (n = 5 mice/vehicle group, n = 4 mice/treated group) (left). Representative pictures of the tumors (right) (scale bar 1 cm). **f**. Weight of mice throughout the experiment described in (e). **g**. Quantification of the mean mCherry intensity of lungs from NOD-SCID mice injected intravenously with mCherry-labeled X50 cells treated with either vehicle (PBS) or Azathioprine 10 mg/Kg (n = 8 mice/group) (left). Representative pictures of the lungs were visualized by fluorescent microscopy (scale bar 1000 μm) (right). **h**. Weight of the mice throughout the experiment described in (g). **i**. Tumor volumes of tumors formed in NOD-SCID mice injected subcutaneously with X50 cells and treated with either vehicle (PBS) or Azathioprine 10 mg/Kg (n = 4 mice/vehicle group, n = 5 mice/treated group) (left). Representative pictures of the tumors (right) (scale bar 1 cm). **j**. Weight of the mice throughout the experiment described in (i). **k**. Tumor volumes of tumors formed in NOD-SCID mice injected subcutaneously with BxPC3 cells and treated with either vehicle (PBS) or Azathioprine 10 mg/Kg (n = 5 mice/vehicle group, n = 4 mice/treated group) (left). Representative pictures of the tumors (right) (scale bar 1 cm). **l**. Weight of mice throughout the experiment described in (k). **m**. Immunoblot analysis of BxPC3 cells with RBFOX2 sgRNA-1 transduced with either CRISPR Cont. or two different sgRNAs for Rac1. **n**. Quantification of wound healing assay of cells described in (m). **o**. Proliferation assay of cells described in (m). **p**. Mean GFP intensity of lungs from NOD-SCID mice injected intravenously with GFP-labeled BxPC3 cells with RBFOX2 sgRNA-1 transduced with either CRISPR Cont. or sgRNA-1 Rac1 (n = 9, n = 7 mice/group, respectively) (left). Representative pictures of the lungs, visualized by fluorescent microscopy and H&E staining (scale bar 1000 μm) (right). Data are mean ± SD. n≥3 independent experiments (n, o), exact *p*-values are shown, n.s. non-significant. Statistical analysis was performed using unpaired two-tailed Student's t-test (a, g, and p) and two-way ANOVA test (b-f, h-l, and n-o). Gel source data, pictures of the lungs, and H&E staining are provided in Supplementary Figs. 7, 15, and 16.

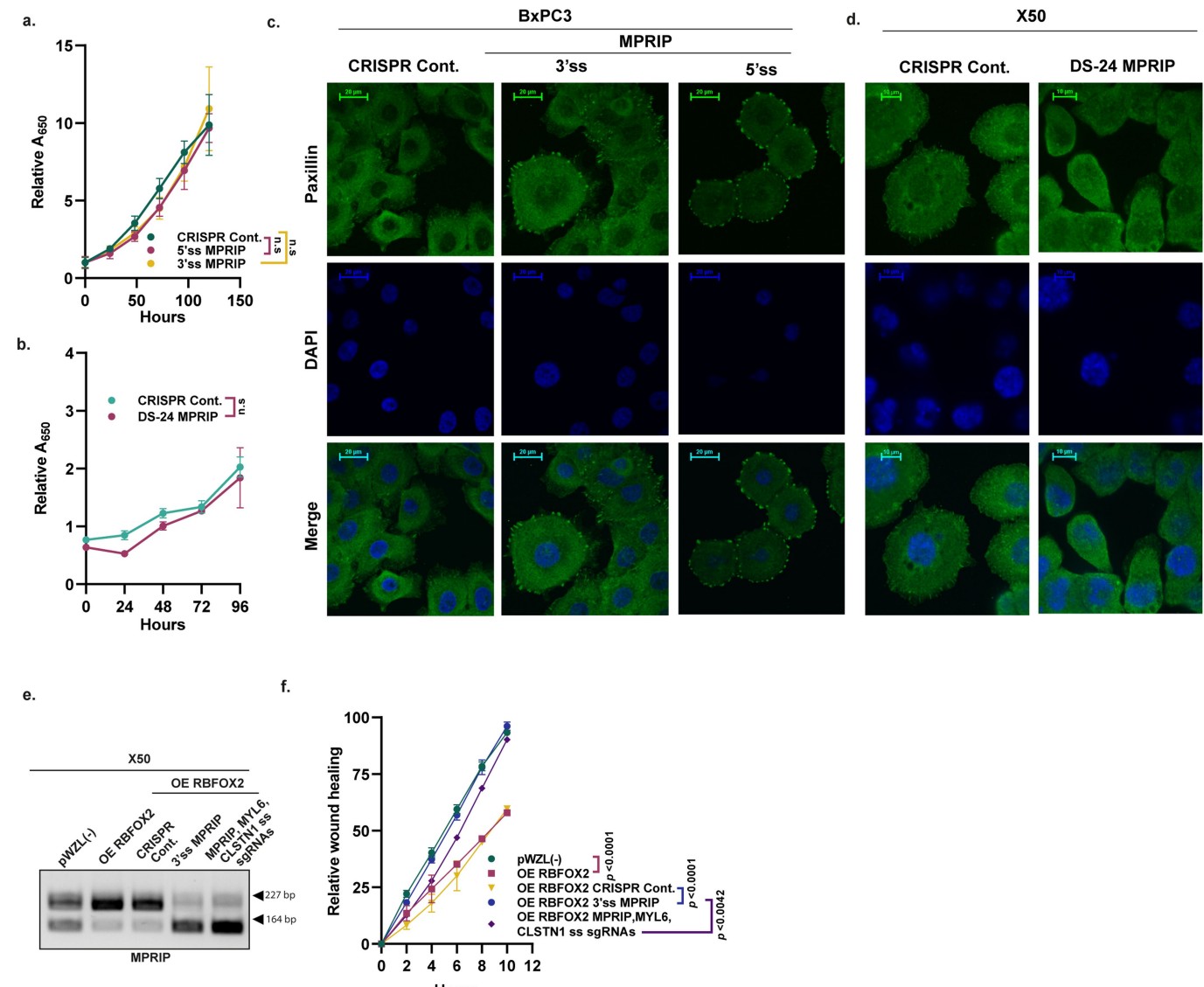

**Extended Data Fig. 9 | MPRIP alternative splicing isoforms alter the cytoskeleton organization of PDA cells. a.** Proliferation assay of MPRIP splicing changes in BxPC3 cells transduced with either CRISPR Cont. or 3' and 5' ss MPRIP sgRNAs. **b.** Proliferation assay of MPRIP splicing changes in X50 cells transduced with either CRISPR Cont. or RBFOX2 motif downstream MPRIP exon 24 sgRNA (DS-24 MPRIP sgRNA). **c.** IF of Paxillin in BxPC3 cells transduced with 3' ss and 5' ss MPRIP sgRNAs. **d.** IF of Paxillin in X50 cells transduced with RBFOX2 motif sgRNA (DS-24 MPRIP sgRNA). For (a, b) Paxillin: green, DAPI: blue. Scale bars, 20 μm. **e.** RT-PCR validation of MPRIP alternative splicing in X50 cells expressing either empty vector pWZL(−) or OE RBFOX2, and X50 cells with OE RBFOX2 transduced with lentiviruses encoding either CRISPR Cont. or 3' ss MPRIP sgRNA or MPRIP, MYL6, and CLSTN1 3' ss sgRNAs together. Gel source data is provided in Supplementary Fig. 8. **f.** Quantification of wound healing assay of cells described in (e). Data are mean ± SD. For all panels n≥3 independent experiments, exact *p*-values are shown, n.s. non-significant. Statistical analysis was performed using two-way ANOVA test (a-b and f).

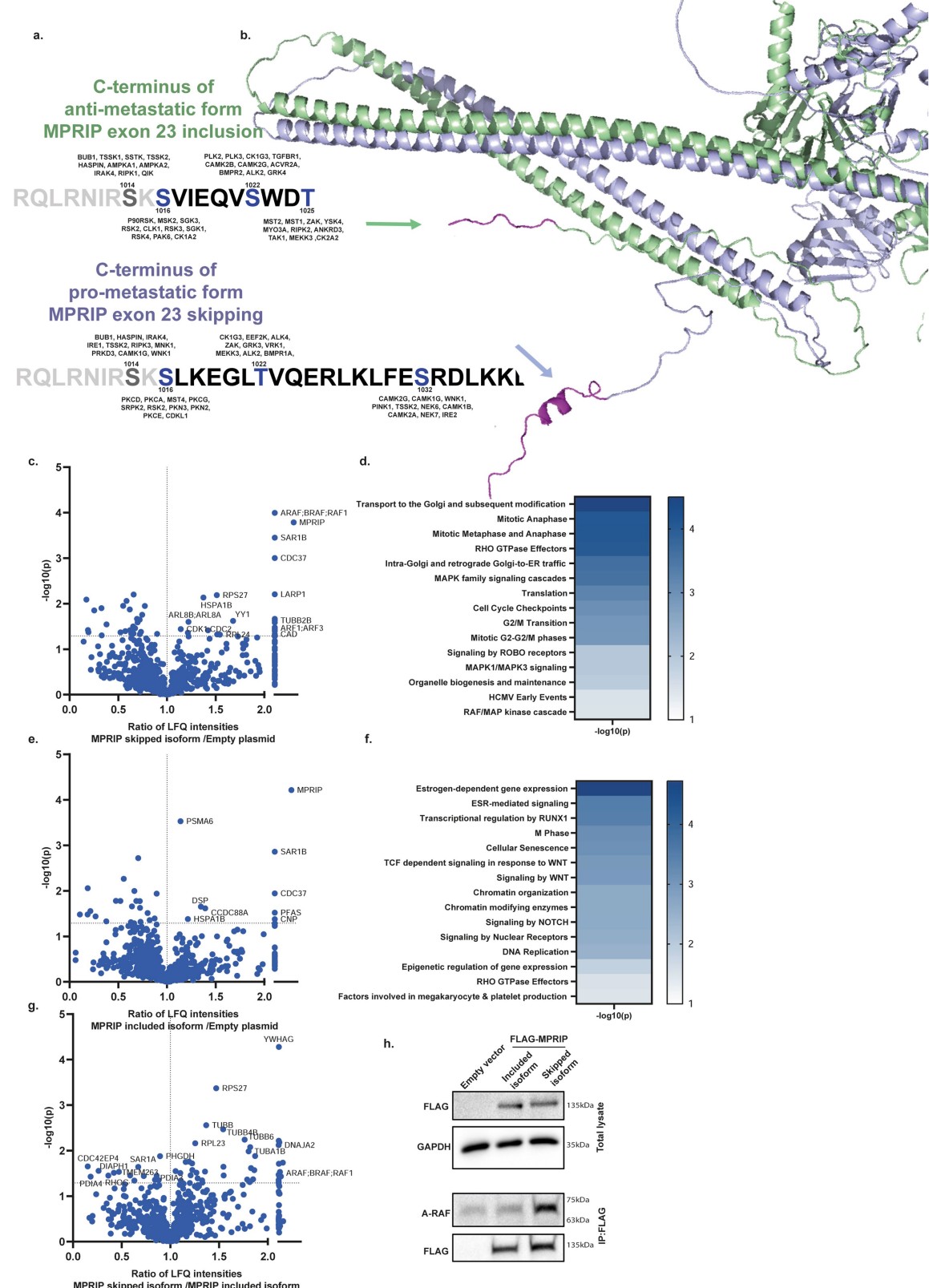

**Extended Data Fig. 10** | See next page for caption.

**Extended Data Fig. 10 | MPRIP exon 23 skipped isoform binds RAF/MAP kinase cascade proteins. a**. Summary of predicted kinases for different phosphorylation sites on each MPRIP isoform as predicted by serine-threonine kinome prediction tool. Supplementary Table 8. **b**. Structure analysis of the C-terminus of each MPRIP isoform as predicted by AlphaFold prediction tool. **c**. Volcano plot representation of the ratio of label-free quantitation (LFQ) intensities of the proteins pulled-down by MPRIP 23 skipped isoform compared to empty vector. **d**. Reactome pathway analysis of the proteins identified in (c). **e**. Volcano plot representation of the ratio of label-free quantitation (LFQ) intensities of the proteins pulled-down by MPRIP exon 23 included isoform compared to empty vector. **f**. Reactome pathway analysis of the proteins identified in (e). **g**. Volcano plot representation of the ratio of label-free quantitation (LFQ) intensities of the proteins pulled-down by MPRIP 23 skipped isoform compared to MPRIP exon 23 included isoform. *p*-value < 0.05 calculated using two-tailed Student's t-test (Perseus statistical package) (c, e, g) (more details described in the methods section). Genes entered into Reactome analysis were identified with imposed cutoffs *p*-value < 0.05. Gene sets were limited to between 5 and 500 genes, and pathways were filtered for a statistical threshold of *p* < 0.05 using over-representation analysis (hypergeometric distribution) test. (d, f). Supplementary Table 9. **h**. Immunoblot of total lysate (top) and immunoprecipitation (bottom) of HEK293 cells transfected with either empty vector, FLAG-MPRIP exon 23 included isoform, or FLAG-MPRIP exon 23 skipped isoform with antibodies against A-Raf and FLAG. n = 3 independent experiments. Gel source data is provided in Supplementary Fig. 9.

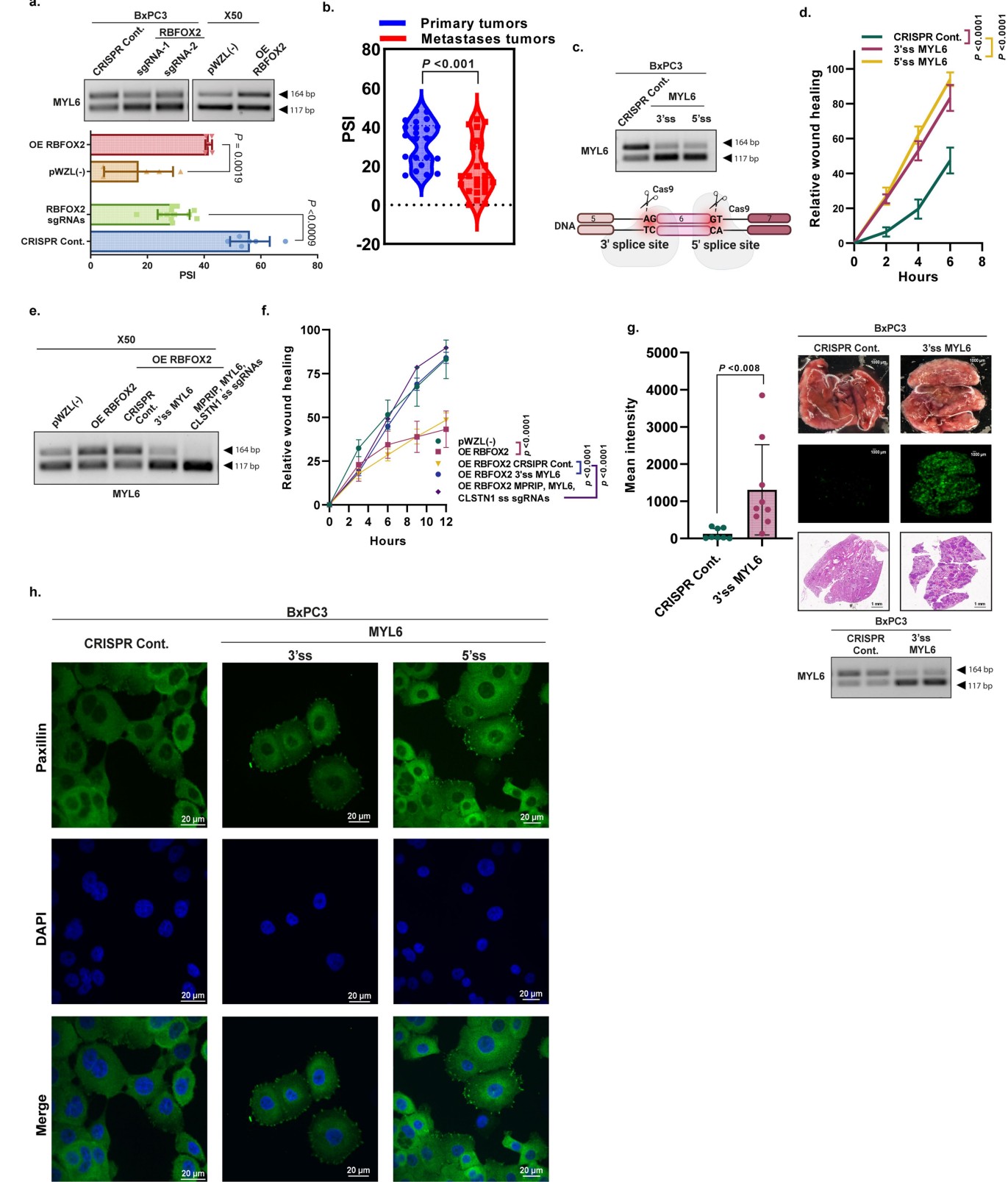

**Extended Data Fig. 11** | See next page for caption.

**Extended Data Fig. 11 | Manipulation of alternative splicing of MYL6 enhances the metastatic potential of primary tumor pancreatic cells.**
**a**. RT-PCR and quantification of MYL6 alternative splicing in BxPC3 cells expressing CRISPR Cont. or RBFOX2 sgRNA-1, 2 and X50 cells expressing pWZL (−) or OE RBFOX2. **b**. Violin plot of MYL6 PSI-values in PDA patient samples (n = 20 primary tumors, n = 24 metastases), analyzed using LabChip®GX microfluidics platform. **c**. Schematic representation of MYL6 splicing modulation (skipping) by CRISPR sgRNAs.(bottom). RT-PCR validation of MYL6 splicing changes in BxPC3 cells transduced with either CRISPR Cont. or 3′ and 5′ ss MYL6 sgRNAs (top). The diagram was created using BioRender.com. **d**. Quantification of wound healing assay of cells described in (c). **e**. RT-PCR validation of MYL6 alternative splicing in X50 cells expressing either empty vector pWZL(−) or OE RBFOX2, and X50 cells with OE RBFOX2 transduced with lentiviruses encoding either CRISPR Cont. or 3′ ss MYL6 sgRNA or MPRIP, MYL6

and CLSTN1 3′ ss sgRNAs together. **f**. Quantification of wound healing assay of cells described in (e). **g**. Mean GFP intensity of lungs from NOD-SCID mice injected intravenously with GFP-labeled BxPC3 cells expressing either CRISPR Cont. or 3′ ss MYL6 sgRNAs (n = 8 mice/group for CRISPR Cont. and 3′ ss MYL6 sgRNAs). Representative pictures of the lungs visualized by fluorescent microscopy (scale bar 1000 µm) and H&E staining (scale bar 1 mm) (right). RT-PCR of RNA from two representative lungs from each group (bottom). **h**. IF of Paxillin in BxPC3 cells transduced with 3′ ss and 5′ ss MYL6 sgRNAs. Paxillin: green, DAPI: blue. Scale bar 20 µm. Data are mean ± SD. For all panels n≥3 independent experiments, exact *p*-values are shown. Statistical analysis was performed using unpaired two-tailed Student's t-test (a-b, g) and two-way ANOVA test (d, f). Gel source data, pictures of the lung, and H&E staining are provided in Supplementary Figs. 10 and 17.

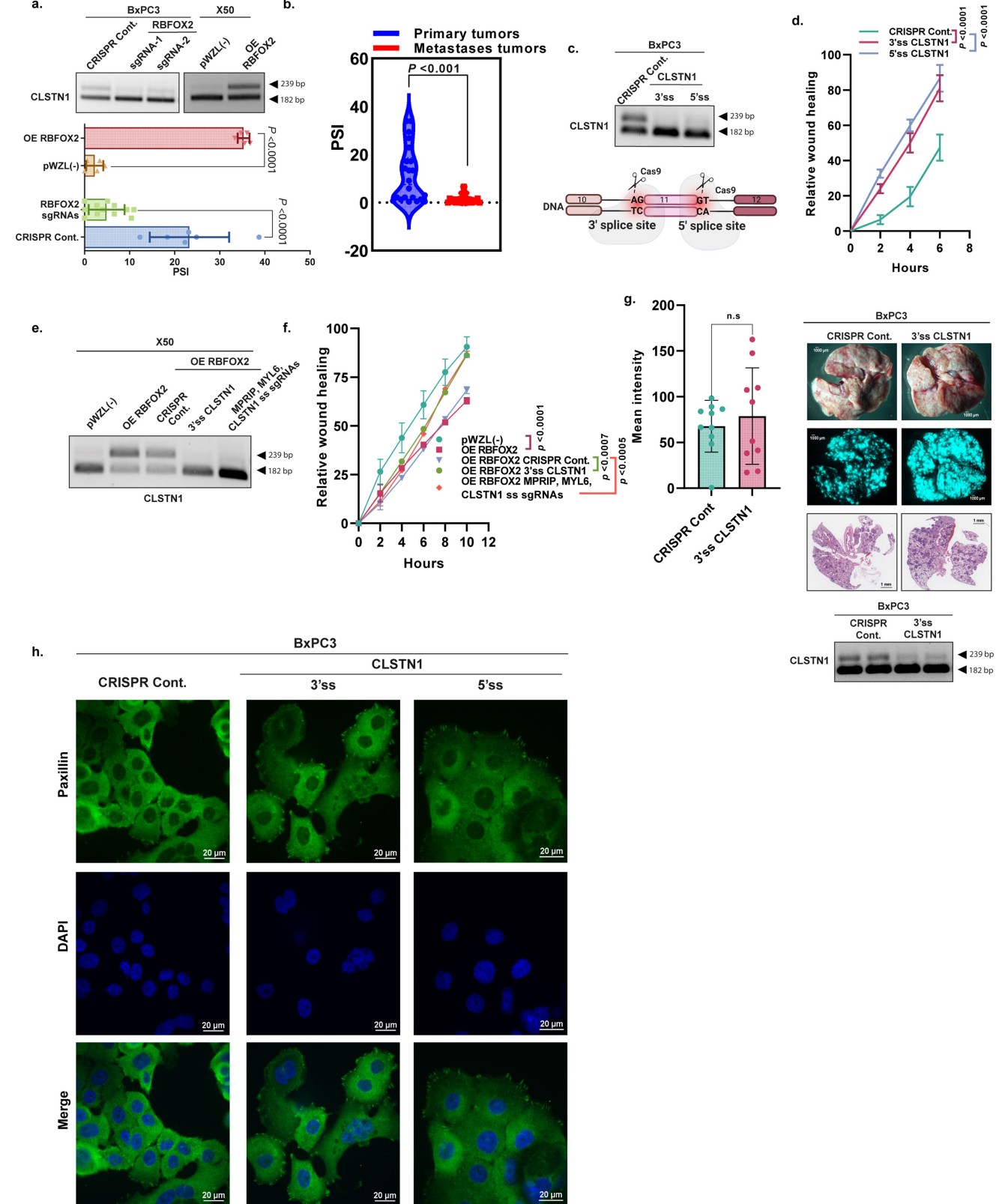

**Extended Data Fig. 12** | See next page for caption.

**Extended Data Fig. 12 | Manipulation of alternative splicing of CLSTN1 enhances the oncogenic potential of primary tumor pancreatic cells. a**. RT-PCR and quantification of CLSTN1 alternative splicing in BxPC3 cells expressing CRISPR Cont. or RBFOX2 sgRNA-1, 2 and X50 cells expressing pWZL (−) or OE RBFOX2. **b**. Violin plot of CLSTN1 PSI-values in PDA patient samples (n = 22 primary tumors, n = 21 metastases), analyzed using LabChip®GX microfluidics platform. **c**. Schematic representation of CLSTN1 splicing modulation (skipping) by CRISPR sgRNAs. (bottom). RT-PCR validation of CLSTN1 splicing changes in BxPC3 cells transduced with either CRISPR Cont. or 3′ and 5′ ss CLSTN1 sgRNAs(top). The diagram was created using BioRender.com. **d**. Quantification of wound healing assay of cells described in (c). **e**. RT-PCR validation of CLSTN1 alternative splicing in X50 cells expressing either empty vector pWZL(−) or OE RBFOX2, and X50 cells with OE RBFOX2 transduced with lentiviruses encoding either CRISPR Cont. or 3′ ss CLSTN1 sgRNA or MPRIP, MYL6 and CLSTN1 3′ ss sgRNAs together. **f**. Quantification of wound healing assay of cells described in (e). **g**. Mean GFP intensity of lungs from NOD-SCID mice injected intravenously with GFP-labeled BxPC3 cells expressing either CRISPR Cont. or 3′ ss CLSTN1 sgRNAs (n = 9 mice/group for CRISPR Cont. and 3′ ss CLSTN1 sgRNAs). Representative pictures of the lungs visualized by fluorescent microscopy (scale bar 1000 μm) and H&E staining (scale bar 1 mm) (right). RT-PCR of RNA from two representative lungs from each group (bottom). **h**. IF of Paxillin in BxPC3 cells transduced with 3′ ss and 5′ ss CLSTN1 sgRNAs. Paxillin: green, DAPI: blue. Scale bar 20 μm. Data are mean ± SD. For all panels n≥3 independent experiments, exact $p$-values are shown, n.s. non-significant. Statistical analysis was performed using unpaired two-tailed Student's t-test (a-b,g) and two-way ANOVA test (d, f). Gel source data, pictures of the lung, and H&E staining are provided in Supplementary Figs. 11 and 18.

# Reporting Summary

## Statistics

For all statistical analyses, confirm that the following items are present in the figure legend, table legend, main text, or Methods section.

| n/a | Confirmed | |
|---|---|---|
| ☐ | ☒ | The exact sample size (*n*) for each experimental group/condition, given as a discrete number and unit of measurement |
| ☐ | ☒ | A statement on whether measurements were taken from distinct samples or whether the same sample was measured repeatedly |
| ☐ | ☒ | The statistical test(s) used AND whether they are one- or two-sided<br>*Only common tests should be described solely by name; describe more complex techniques in the Methods section.* |
| ☐ | ☒ | A description of all covariates tested |
| ☐ | ☒ | A description of any assumptions or corrections, such as tests of normality and adjustment for multiple comparisons |
| ☐ | ☒ | A full description of the statistical parameters including central tendency (e.g. means) or other basic estimates (e.g. regression coefficient) AND variation (e.g. standard deviation) or associated estimates of uncertainty (e.g. confidence intervals) |
| ☐ | ☒ | For null hypothesis testing, the test statistic (e.g. *F*, *t*, *r*) with confidence intervals, effect sizes, degrees of freedom and *P* value noted<br>*Give P values as exact values whenever suitable.* |
| ☒ | ☐ | For Bayesian analysis, information on the choice of priors and Markov chain Monte Carlo settings |
| ☒ | ☐ | For hierarchical and complex designs, identification of the appropriate level for tests and full reporting of outcomes |
| ☒ | ☐ | Estimates of effect sizes (e.g. Cohen's *d*, Pearson's *r*), indicating how they were calculated |

*Our web collection on statistics for biologists contains articles on many of the points above.*

## Software and code

Policy information about availability of computer code

| Data collection | Nikon-TL (fluorescence microscopy images), BIORAD ChemiDoc XRS+ System (immunoblot blots and PCR gels), IncuCyte (wound healing), 2200 TapeStation, Illumina NextSeq 2000 system, LabChip®GX microfluidics platform, Step one plus real-time PCR system-Applied Biosystems, Spinning Disk Confocal Microscope - Nikon Instruments Inc., Aperio Digital Pathology Slide Scanners - Leica Biosystems, real-time cell imaging system - IncuCyte Live cell, Q Exactive Plus mass spectrometer - Thermo Fisher Scientific, nanoflow UHPLC instrument Ultimate 3000 Dionex - Thermo Fisher Scientific. |
|---|---|
| Data analysis | Bioconductor (v3.7) within the R (v3.5.1) programming environment, GraphPad Prism (version 9.5.0.730) for data analysis and plots, R's prcomp function for PCA analysis, PSI-Sigma (version 1.9c) for splicing analysis, Reactome database for pathway enrichment, XSTREME package (version 5.5) for motif analysis, NIS Elements (version 4.13) imaging software, Analysis software (IncuCyte Cat No 4400 version 2022A) for wound healing analysis, MaxQuant computational platform (version 2.0.3.0) for mass spec analysis, LabChip GX Reviewer software (version 5.3.2115.0) for splicing validations and quantifications, STAR v 2.5.3a. CHOP CHOP for sgRNAs design., AlphaFold for structure analysis. DESeq2 for gene expression analysis. |

For manuscripts utilizing custom algorithms or software that are central to the research but not yet described in published literature, software must be made available to editors and reviewers. We strongly encourage code deposition in a community repository (e.g. GitHub). See the Nature Portfolio guidelines for submitting code & software for further information.

## Data

Policy information about <u>availability of data</u>

All manuscripts must include a <u>data availability statement</u>. This statement should provide the following information, where applicable:
- Accession codes, unique identifiers, or web links for publicly available datasets
- A description of any restrictions on data availability
- For clinical datasets or third party data, please ensure that the statement adheres to our <u>policy</u>

RNA-seq data generated as part of this study is deposited ito the BioProject under accession number PRJNA797585.
RNA-seq data of PDA patients are available at European Genome Phenome Archive https://www.ebi.ac.uk/ega/home (Study ID EGAS00001002543) databases IDs:
EGAD00001003584, EGAD00001004548, EGAD00001005799, EGAD00001006081.
XSTREME database (Ray2013 Homo sapiens), Reactome database (Homo sapiens), Ensembl gene annotation Human (GRCh38.p13, Homo sapiens).

# Field-specific reporting

Please select the one below that is the best fit for your research. If you are not sure, read the appropriate sections before making your selection.

☒ Life sciences  ☐ Behavioural & social sciences  ☐ Ecological, evolutionary & environmental sciences

For a reference copy of the document with all sections, see <u>nature.com/documents/nr-reporting-summary-flat.pdf</u>

# Life sciences study design

All studies must disclose on these points even when the disclosure is negative.

| | |
|---|---|
| Sample size | For experiments in mice, group sizes were determined by power analysis on prior data collected using the same experimental procedures (Tavazoie et.al 2008, Golan, T. et al. 2017) and calibration experiments we performed. Experiments were designed to detect differences greater than 20% at a significance of $p<0.05$.  For all mouse experiments (including tumor growth and metastatic model) minimum n=4 mice was used. All the exact sample sizes are stated in the figure legends. |
| Data exclusions | No data were excluded. |
| Replication | The experiments were repeated multiple times (2-4), with 2-4 biological replicates each time, and the results were consistent across all the trials. The majority of the assays were conducted three times, with three biological replicates each time, which demonstrates the reproducibility of the data.  All attempts at replication in this study were successful. |
| Randomization | In the in vivo experiments the animals were randomly divided into different groups. For immunofluorescence and immunohistochemistry staining, the fields of view were randomly chosen. For other experiments, where the samples were not randomly assigned, there was no need for group allocation or randomization because all samples were consistently and independently measured in a controlled manner. |
| Blinding | In the mouse experiments, the experimenters were not blinded to the treatment groups, as they administered the treatments and measured the tumors themselves.<br>In vitro cell culture experiments, blinding is not applicable because the researchers need to verify samples and controls for each experiment. However, whenever feasible, a second  researcher confirmed the results. |

# Reporting for specific materials, systems and methods

We require information from authors about some types of materials, experimental systems and methods used in many studies. Here, indicate whether each material, system or method listed is relevant to your study. If you are not sure if a list item applies to your research, read the appropriate section before selecting a response.

## Materials & experimental systems

| n/a | Involved in the study |
|---|---|
| ☐ | ☒ Antibodies |
| ☐ | ☒ Eukaryotic cell lines |
| ☒ | ☐ Palaeontology and archaeology |
| ☐ | ☒ Animals and other organisms |
| ☒ | ☐ Human research participants |
| ☒ | ☐ Clinical data |
| ☒ | ☐ Dual use research of concern |

## Methods

| n/a | Involved in the study |
|---|---|
| ☒ | ☐ ChIP-seq |
| ☒ | ☐ Flow cytometry |
| ☒ | ☐ MRI-based neuroimaging |

# Antibodies

| Antibodies used | RBFOX2, Sigma, #006240, Prestige Antibodies® Immunoblotting: 0.4 µg/mL<br>SRSF1, mAb AK96 culture supernatant (Cáceres et al. 1997), Immunoblotting: 1:1000<br>SRSF6, mAb 8-1-28 culture supernatant (Fu and Maniatis 1990), Immunoblotting: 1:1000<br>Tubulin, Abcam, #ab6160, [YL1/2], Immunoblotting: 1:10000<br>Flag, Sigma, # F3165, clone M2, Immunoblotting: 10 µg/mL<br>GAPDH, Sigma, # G9545, polyclonal, Immunoblotting: 0.2 µg/mL<br>β-catenin, Abcam, #ab6302, polyclonal, Immunoblotting: 1:4000<br>β-Actin, Santa Cruz, #sc-1616, I-19, Immunoblotting: 1:1000<br>total-MEK 1/2, Cell Signaling, #8727, D1A5, Immunoblotting: 1:1000<br>Paxillin, BD Biosciences, BD612405, Clone 349/Paxillin (RUO), Immunofluorescence: 1:1000<br>Rac1, Cytoskeleton, Inc., ARC03, Immunoblotting: 1:500<br>A-Raf Santa Cruz, sc-408, C-20, Immunoblotting: 1:500<br>GFP abcam #ab6673 Immunohistochemistry 1:1000<br>Anti-FLAG Sigma #A2220 M2 Affinity gel Immunoprecipitation 30 µg/µL<br>ImmPRESS HRP anti-Goat IgG polymer) Vector Laboratories #MP-7401<br>Peroxidase-conjugated AffiniPure Goat Anti-Mouse IgG (H+L), Jackson ImmunoResearch Inc.,# 115-035-003, Immunoblotting: 1:10000<br>Peroxidase-conjugated AffiniPure Goat Anti-Rabbit IgG (H+L), Jackson ImmunoResearch Inc.,# 111-035-003, Immunoblotting: 1:10000<br>Peroxidase-conjugated AffiniPure donkey Anti- goat IgG (H+L), Jackson ImmunoResearch Inc., # 705-035-003, Immunoblotting: 1:10000<br>Alexa Fluor® 488 AffiniPure Goat Anti-Mouse IgG (H+L), Jackson ImmunoResearch Inc., 115-545-003, Immunofluorescence: 1:800<br>Donkey Anti-Rat IgG H&L (HRP) preadsorbed, Abcam, ab102265, Immunoblotting: 1:10000 |
|---|---|
| Validation | All antibodies were validated by the supplier on human samples. All antibodies were checked in the lab by immunoblotting using cell lysates and compared to the supplier's. |

# Eukaryotic cell lines

Policy information about cell lines

| Cell line source(s) | X50, X139, X252, were provided by Dr. Talia Golan (PMID: 28489577) (PMID: 29396858). BxPC3 (CRL-1687), HEK293T (CRL-3216), Phoenix-AMPHO (CRL-3213) and HEK293 (CRL-1573) cells lines were originally obtained from the American Type Culture Collection (ATCC). |
|---|---|
| Authentication | Cell line authentication test was performed at the Technion Genomics Center. The test was performed using the Promega GenePrint 24 System in order to determine short tandem repeat (STR) profile of 23 loci plus Amelogenin for gender determination (X or XY). In addition, the male-specific DYS391 locus is included to identify null Y allele results for Amelogenin. |
| Mycoplasma contamination | All cell lines are frequently tested for mycoplasma contamination. Cell lines used in this study were verified to be mycoplasma negative before undertaking any experiments . |
| Commonly misidentified lines (See ICLAC register) | None |

# Animals and other organisms

Policy information about studies involving animals; ARRIVE guidelines recommended for reporting animal research

| Laboratory animals | NOD SCID mice (Jackson Lab, 0001303) were ordered at 6 weeks of age. The mice were housed under standard laboratory conditions in specific-pathogen-free cages in an animal room at constant temperature (19–23°C) and regulated humidity under a 12h/12h light–dark cycle and received standard laboratory chow and water ad libitum. All mice entered the experiments at 8–12 weeks of age. Both male and female mice were used for the experiments. |
|---|---|
| Wild animals | The study did not involve wild animals. |
| Field-collected samples | The study did not involve samples collected from the field. |
| Ethics oversight | PDA patient-derived xenograft (PDX) generation in nude mice were performed in accordance with the guidelines of Sheba Medical Center Institutional Animal Care and Use Committee (IACUC) (5539/13).<br>Metastatic and tumor formation in vivo experiments were performed in accordance with the guidelines of IACUC at the Hebrew University (MD-15-14634-5). |

Note that full information on the approval of the study protocol must also be provided in the manuscript.

