## [Peer Review File · Nature]

Manuscript Title: RBFOX2 modulates a metastatic signature of alternative splicing in pancreatic cancer

Reviewer Comments & Author Rebuttals

Reviewer Reports on the Initial Version:

Referees' comments:

Referee #1 (Remarks to the Author):

Jbara et al report results consistent with a model in which the splicing regulator RBFOX2 controls a program of alternative splicing decisions relevant for metastasis of pancreatic cancer. To build this model, the authors start by analyzing transcriptome data from cohorts of primary and metastatic pancreatic tumors and identify a signature of alternative splicing changes that distinguishes between these groups. Sequence motif analyses identify potential binding sites for RBFOX2 enriched in the differentially spliced regions and protein analyses document strongly decreased levels of RBFOX2 in metastatic tumors. Modulation of RBFOX2 levels in primary or metastatic pancreatic tumor cell lines affects wound healing and colony formation but not proliferation of these cell lines, as well as their capacity to form lung metastasis when injected into nude mice, consistent with RBFOX2 acting as a tumor suppressor in pancreatic cancer progression. A detailed analyses of alternative RBFOX2-dependent alternative splicing changes in these cell lines reveals an enrichment in genes involved in Rho GTPase pathways, consistent with phenotypic changes in focal adhesion and with the effects of Rho GTPase inhibitors in cellular phenotypes and metastatic formation. Finally, the authors focus on an alternative splicing event in the MPRIP gene controlled by RBFOX2 and show that modulation of the switch between isoforms affects cellular phenotypes such as wound healing, colony formation or focal adhesion as well as the capacity of pancreatic cell lines to form lung metastasis.

The insights of Karni and co-workers are very relevant because they identify a post-transcriptional regulatory circuit important for pancreatic cancer metastasis (the most frequent complication of these deadly tumors) and may pave the way for novel therapeutic approaches by repurposing Rho GTPase inhibitors and/or modulating alternative splicing of key RBFOX2 targets. The data supporting the model, spanning transcriptomics, cellular phenotypes and tumor xenografts, are convincing.

The following revisions could help to improve the manuscript:

1. In contrast with the clear results of Figure 1g, the western blot analysis of RBFOX2 protein levels in cell lines shown in Figure 2a is less compelling. While overexpression results are clear, the detection of endogenous protein in the non-metastatic cell line BxPC3 is not and the result is further complicated by additional bands of various relative intensities in the different lanes. Knockdown results should be better documented.
2. The authors argue that the results of Figures 3g-m nominate Rho GTPase inhibitors as potential therapeutic agents for metastatic pancreatic cancer. The authors were prompted to carry out these experiments because of the enrichment of genes involved in this pathway among the RBFOX2

splicing targets. However the pathway had been implicated before in pancreatic cancer metastasis and inhibitors of the pathway shown to reduce this process (e.g. Park et al, 2017; Li et al 2009; Vennin et al, 2020; Ungefroren et al, 2018; Arnst et al, 2017; Whatcott et al, 2017). Azathiopine, which shows clear inhibitory effects in Figures 3g-m, had been reported before to decrease metastasis in both xenograft and genetic mouse models of pancreatic cancer (Razidlo et al, 2015). Therefore it is not clear how the results of Figures 3g-m advance our knowledge of the involvement of Rho GTPases in pancreatic cancer metastasis and how they are linked to the specific alterations of this pathway achieved through the RBFOX2-mediated splicing alterations reported in this manuscript.

3. The results of Figure 4 assessing the contribution of MPRIP alternative splicing are very interesting. While the differences observed in some assays (e.g. Fig 3f) are less compelling, it remains remarkable that modulation of a single alternative splicing event, out of the many regulated by RBFOX2, shows detectable effects in whole phenotypes. It may be relevant to analyze whether the level of alterations in this alternative splicing event (or more generally those events found to correlate with metastasis at transcriptome level) correlate with worse prognosis (e.g. using psichomics: <https://www.bioconductor.org/packages/release/bioc/html/psichomics.html>). Also related to this, the results of Fig 3d show that a number of primary tumors also show reduced PSI values. Is this the case for other RBFOX2 targets in the same samples?

4. There is an apparent discrepancy between the results of Figs 3b and 3e: the results of knockdown/overexpression of RBFOX2 (Fig 3b) consistently argue that the protein promotes accumulation of the longer MPRIP isoform. One would therefore expect that a mutation that prevents RBFOX2 binding would induce the shorter isoform, but the opposite is observed in Fig 4e. How do the authors explain this discrepancy?

5. The results of Fig 4n are compelling and critically important for the main argument of the paper, and therefore it would be important to verify that the levels of MPRIP isoforms remain altered by the sgRNA treatments in the metastatic tumor samples.

6. It would be good to at least discuss the expectations/difficulties of the converse experiment: would promotion of the longer isoform inhibit metastatic potential of X50 cells?

7. It would be also good to discuss whether evidence for downregulation of RBFOX2 protein levels and alterations in splicing of Rho GTPase pathway genes is observed in metastasis in other cancer types.

8. Another interesting discussion point is the potential mechanism by which RBFOX2 protein (but not mRNA) levels are downregulated in metastatic tissue.

Referee #2 (Remarks to the Author):

The manuscript by Jbara and colleagues unveils an interesting regulatory pathway in the progression to metastasis of pancreatic adenocarcinoma (PDA), led by loss of RBFOX2, which modulates alternative splicing events. The authors first examined the differentially spliced events in metastatic vs. primary human PDA samples, and identified an enrichment of RBFOX2 binding motifs in the differentially spliced genes. Then they assessed the role of RBFOX2, using a variety of in-vitro and in-vivo assays. Deep RNA-seq data from RBFOX2 OE/KD experiments revealed changes in alternative splicing of genes associated with Rho GTPase pathways, and the authors confirmed this observation

by showing altered focal adhesion formation upon RBFOX2 activity. Furthermore, Jbara et al. identified and investigated three candidate RBFOX2-regulated splicing events (in MPRIP, MYL6, and CLSTN1), and demonstrated their roles in altering the phenotype of PDA cancer cells.

Finding RBFOX2 as a master regulator of RNA splicing events that limit PCA metastasis, thereby rationalizing its observed downregulation in metastases, is new and interesting. There are, however, a number of questions raised that warrant attention to justify publication in Nature.

1) The authors report a list of genes that are distinguished by alternative splicing events in human PCA primary and metastatic tumors mediated by the presence or absence of RBFOX2, and assert that these changes are not associated with changes in gene expression. The authors should show an mRNA heat map with an identical gene list to that shown in Figure 1d for splicing events, to substantiate their conclusion that changes in gene expression of these candidates is not changed. Other figure panels could be moved to an ED-Figure to make space

2) Also, ILF3 is listed three times in Fig 1d, and SRRM2. Are these different splicing events in the same transcript, and if so is there a need to keep them separate?

3) In Figures 1.h and 1.i, the authors relay that RBFOX2 is under translational control. Since the data indicate that none of the signature mutations in PCA correlate with the differential splicing events in which RBFOX2 is implicated, the authors should provide some clues into how this downregulation of RBFOX2 protein is achieved in metastases? E.g., upregulating microRNAs or RNA-binding translational repressor proteins?

4). The pharmacological inhibitors of implicated downstream pro-metastatic effectors of the loss of RBFOX2 seem unlikely to be pure metastasis suppressors. The authors should also treat s.c. transplant tumors and assess tumor growth of primary tumors, to address their specificity for impairing metastasis. There is no indication of differential proliferation or survival of primary vs metastatic human or mouse PDA cancer cells in vivo, and yet data is presented indicating that these inhibitors affect cell survival.

5) The scratch migration (“wound healing”) assay used throughout the manuscript is an imprecise metric of bona fide invasiveness. Have Boyden chamber assays involving invasion through matrigel or other ECM been performed? If not, the limitations of the migration assay should be described.

5. Three candidate alternatively spliced targets of RBFOX2 are introduced and functionally assessed in analogous assays to RBFOX2 itself: MPRIP, MYL6 and CLSTN1. All three have similar effects in the migration assay, noting its limitations. Tail vein metastasis assays are not included for MYL6 and CLSTN1 and should be. Moreover, the similarity of results with the four genes - RBFOX2, MPRIP, MYL6 and CLSTN1 – begs the question: Are all four in one pathway? Epistasis assays, e.g. co-modulating RBFOX2 and each of the three downstream effectors in the metastasis assay (and maybe also the migration assay, where effects are quite modest in general) would shed light on this question, as would co-modulating all three candidate effectors together. More sophisticated analytical procedures might also be informative, for example performing immunohistochemistry on the metastases, rather than just showing and quantitating fluorescent protein signal.

Referee #3 (Remarks to the Author):

The manuscript by Jbara and colleagues describe an interesting approach to identify global changes in protein expression during malignant transformation in pancreatic cancers. The results underscore modifications in the alternative splicing machinery as an underappreciated mechanism to affect the expression of various proteins and thus have a high impact in tumour progression. Such mechanism has been the topic of much interest, with reports of its relevance in different types of cancers breast cancer, lung cancer, melanomas or myelofibrosis.

The novelty of the manuscript is the identification of a novel splicing factor, RBFOX2, as a potent metastasis suppressor in pancreatic primary tumour cells. Metastatic samples of patients showed a remarkable enrichment of alternatively spliced mRNAs and proteins belonging to the Rho GTPase signalling pathways. The experimental design to identify the alternatively spliced mRNAs are complementary and robustly support their findings using different pancreatic tumour cells and patient samples. The text is well written, and the rationale of the experimental approach clear to follow. The figures and data presentation are of high quality.

The screening for affected mRNAs is complemented with the demonstration of the protein level of the respective translated isoforms: alternative spliced protein appears when RBFOX2 is modified in the different cellular models. This reviewer finds that the validation of the potential impact of specific isoforms is not well elaborated, and thus the mechanisms underpinning the distinct phenotypes are not shown. Please see specific criticisms below.

Specific comments:

1. While validating MPRIP isoforms, it would be important to show what is the consequence of skipping or inclusion of exon 23 for example. Does inclusion of exon 23 provides phosphorylation sites that could regulate MPRIP function? Or affect its interactions with RhoA, thereby altering its signalling?
2. In figure 4, skipping exon 23 of MPRIP leads to a striking phenotype of more efficient wound healing (Fig.4i-j) and higher lung colonization in vivo (Fig.4n). However, immunofluorescence staining shows that these cells have much larger and thicker focal adhesions that would be predicted to bind better to ECM and have slower motility (Fig.3m). How can these results be conciliated?
3. It is also remarkable that MPRIP exon skipping by itself (Fig.3m) can mimic the focal adhesion phenotype of RBFOX sgRNA (Fig.3d). Would alternative splicing of other RhoA targets also phenocopy this? Is a reversion to a more epithelial-type morphology also observed with MPRIP splicing?
4. Are Rac1b levels affected by RBFOX2? This is a well-characterized isoform of Rac1 that is upregulated in different tumours and with well mapped distinct signalling from Rac1. It will be important to report either way.
5. The rationale for investigating Rac1 and Cdc42 signalling is not clear, in line with the above experiments focusing on RhoA pathway. The choice of the inhibitor used (Azathioprine) is not appropriate. This drug is not specific for Rac1 or Cdc42 as it broadly inhibits DNA, RNA and protein synthesis. Thus, the approach is not specific and does not determine causality. There are other specific inhibitors for these GTPases that could have been used to support their claim.

Author Rebuttals to Initial Comments:

Reviewer 1

The insights of Karni and co-workers are very relevant because they identify a post-transcriptional regulatory circuit important for pancreatic cancer metastasis (the most frequent complication of these deadly tumors) and may pave the way for novel therapeutic approaches by repurposing Rho GTPase inhibitors and/or modulating alternative splicing of key RBFOX2 targets. The data supporting the model, spanning transcriptomics, cellular phenotypes and tumor xenografts, are convincing.

We thank the reviewer for appreciating the significance of our work.

1. In contrast with the clear results of Figure 1g, the western blot analysis of RBFOX2 protein levels in cell lines shown in Figure 2a is less compelling. While overexpression results are clear, the detection of endogenous protein in the non-metastatic cell line BxPC3 is not and the result is further complicated by additional bands of various relative intensities in the different lanes. Knockdown results should be better documented.

We apologize for the unclear blot of RBFOX2 protein levels. We have kept the overexpression results from the metastatic cells (which the reviewer wrote are clear). We have now substituted a different blot for the non-metastatic cell line BxPC3 that clearly detects endogenous RBFOX2 protein, showing the same intensity and pattern of RBFOX2 protein bands in the different lanes. We believe that this revised Fig. 2a better documents the knockdown of RBFOX2 by gRNAs in the non-metastatic BxPC3 cell line.

2. The authors argue that the results of Figures 3g-m nominate Rho GTPase inhibitors as potential therapeutic agents for metastatic pancreatic cancer. The authors were prompted to carry out these experiments because of the enrichment of genes involved in this pathway among the RBFOX2 splicing targets. However, the pathway had been implicated before in pancreatic cancer metastasis and inhibitors of the pathway shown to reduce this process (e.g. Park et al, 2017; Li et al 2009; Vennin et al, 2020; Ungefroren et al, 2018; Arnst et al, 2017; Whatcott et al, 2017). Azathioprine, which shows clear inhibitory effects in Figures 3g-m, had been reported before to decrease metastasis in both xenograft and genetic mouse models of pancreatic cancer (Razidlo et al, 2015). Therefore, it is not clear how the results of Figures 3g-m advance our knowledge of the involvement of Rho GTPases in pancreatic cancer metastasis and how they are linked to the specific alterations of this pathway achieved through the RBFOX2-mediated splicing alterations reported in this manuscript.

We agree with the reviewer that the notion that the Rac/Rho/CDC42 pathways are important for pancreatic cancer progression has been argued previously. However, the mechanisms by which these pathways are regulated and their direct role in the metastatic process of pancreatic cancer are still not fully clear. We also agree that the use of Azathioprine may not be the best choice. We have now added several experiments that tie RBFOX2 as an upstream regulator of the Rho GTPase pathway, as well as, an additional Rac1 inhibitor which is more specific than Azathioprine. It is important to note that knockout of Rac1 (which mimics treatment with either Azathioprine or MBQ-167), or use of a more specific inhibitor MBQ-167 (Rac and Cdc42 inhibitor), does not affect the proliferation or survival of the cells (new Fig. 3g-k, new Extended Data Fig. 8m-p). This is in contrast to Azathioprine and most of the inhibitors tested in the cited papers, Fasudil (Vennin et al. 2020 and Whatcott et al. 2017) and

the Rac1 inhibitor (Arnst et al. 2017), that were tested only in cell culture, that do affect proliferation/survival. The use of anti-oxidant metabolite 8-OHdG, which is a pluripotent inhibitor (Park et al. 2017), does not contribute specifically to our understanding of pancreatic cancer progression.

We now replace the experiments using azathioprine (Fig. 3g-k) with experiments using MBQ-167 on BxPC3 primary tumor cells with RBFox2 knockout, which become metastatic only due to this single gene knockout (**new Fig. 3g-k**). In addition, in vivo administration of MBQ-167 to mice injected intravenously with GFP-labeled PDX-derived metastatic X50 cells inhibited lung metastases (**Extended Data Figs. 8a-b**), without an effect on primary tumor growth subcutaneously (**Extended Data Figs. 8i-j**). We removed Extended Data Fig. 8 due to use of high MBQ-167 dosage that causes cell death and inhibition of proliferation. We have calibrated the dose and now use a lower dose (**new Fig. 3g-k**).

new Fig.3 g. Quantification of wound healing assay of BxPC3 RBFox2 sgRNA-1 cells treated with either MBQ-167 (0.05 μ M) or DMSO. **h.** Quantification of proliferation assay of BxPC3 sgRNA-1 cells treated with either MBQ-167 (0.05 μ M) or DMSO. **i.** Trypan blue-based cell count of BxPC3 sgRNA-1 cells treated with either MBQ167 (0.05 μ M) or DMSO for 24 hours. **j.** Quantification of the mean GFP intensity of lungs from NOD-SCID mice injected intravenously with GFP-labeled BxPC3 RBFox2 sgRNA-1 cells treated either with vehicle or MBQ167 3mg/Kg (n= 7 mice for each group) (**left**). Representative pictures of the lungs, visualized by fluorescent microscopy (**right**). **k.** Weight measure of the mice throughout the experiment described in (**j**).

new Extended Data Fig. 8. a. Quantification of representative lungs from NOD-SCID mice injected intravenously with GFP-labeled X50 cells and treated with either Vehicle or MBQ-167 starting 5 days after cell injection (3 mg/kg, 3 times a week for 5 weeks) (n=8 mice per group). Representative pictures are shown on the right. **b.** weight measure of the mice throughout the experiment described in (a).

new Extended Data Fig. 8. i. Tumor volumes of tumors formed in NOD-SCID mice injected subcutaneously with X50 cells and treated with either Vehicle or MBQ-167 starting 5 days after cell injection (3 mg/kg, 3 times a week for 5 weeks) (n=5 mice per group, two tumors per mouse) **(left)**. Representative pictures of the tumors **(right)**. **j.** weight measure of the mice throughout the experiment described in **(i)**.

In order to mimic the effect, we genetically knocked-out Rac1 in the BxPC3 primary tumor cells with RBFOX2 knockout (**new Extended Data Fig. 8m-p**). We show that treatment with MBQ-167 Rac1/Cdc42 inhibitor, as well as Rac1 knockout, reverses the enhanced migration effect of RBFOX2 knockout, without an effect on proliferation or survival, suggesting that indeed RBFOX2 acts upstream to Rac1 in modulating cellular motility/invasiveness.

new Extended Data Fig. 8. m. Western blot analysis of BxPC3 cells with RBFOX2 sgRNA-1 transduced with either CRISPR Cont. or two different sgRNAs for Rac1. **n.** Quantification of wound healing assay in BxPC3 cells with RBFOX2 sgRNA-1 transduced with either CRISPR Cont. or two different sgRNAs for Rac1. **o.** Proliferation assay in BxPC3 cells with RBFOX2 sgRNA-1 transduced with either CRISPR Cont. or two different sgRNAs for Rac1. **p.** Quantification of the mean GFP intensity of lungs metastases in NOD-SCID mice injected intravenously with GFP-labeled BxPC3 RBFOX2 sgRNA-1 cells which transduced with either CRISPR Cont. or sgRNA-1 for Rac1 (n=9, n=7 mice for each group respectively) **(left)**. Representative pictures of the lungs, visualized by fluorescent microscopy **(right)**.

We now added to the text (Page 8 lines 225 -256):

"To examine if the regulation of the Rho GTPase pathways by RBFOX2 is critical for pancreatic cancer progression, we took advantage of two inhibitors of the Rho GTPase pathways: MBQ-167, a dual Rac/Cdc42 inhibitor^{38,39}, and Azathioprine, a blocker of Rac1 activity, which generates 6-thioguanine triphosphate (6-Thio-GTP) that competes with GTP, on binding to Rac1. The pharmacological interference of Rho GTPase pathways may affect RBFOX2-regulated pathways. MBQ-167 administration to RBFOX2-depleted BxPC3 cells inhibited the migration rates, without a change in survival or proliferation rates in vitro (**Figs. 3g-i**). In vivo administration of MBQ-167 to mice injected intravenously with either GFP-labeled RBFOX2-depleted BxPC3 cells or GFP-labeled PDX-derived metastatic X50 cells inhibited lung metastases (**Figs. 3j-k, Extended Data Figs. 8a-b**), without an effect on primary tumor growth subcutaneously (**Extended Data Figs. 8e-f and 8i-j**). Similar results were obtained using Azathioprine (**Extended Figs. 8c-d, 8k-l**). However, Azathioprine, unlike MBQ-167 inhibited primary tumor growth of the primary tumor cell line BxPC3 (**Extended Data Fig. 8g-h**). To mimic the effect of these inhibitors we genetically knocked-out Rac1 in RBFOX2-depleted BxPC3 cells (**Extended Figs. 8m**). Comparable to treatment with the Rac1 inhibitors, these cells had slower migration rates in vitro, without any change in proliferation, and formed a significantly reduced number of metastases in lungs in vivo (**Extended Fig. 8n-p**).

Thus, Azathioprine and MBQ-167 are good candidates for potential therapeutic intervention in metastatic PDA. Moreover, these results corroborate the direct role of RBFOX2 in PDA metastatic progression; RBFOX2 - depleted BxPC3 cells are more sensitive to the effect of either Rac1 inhibitors or Rac1 knock-out."

We now also cite the (Ungefroren, Witte and Lehnert, 2018) review summarizing previous literature on the roles of Rac/Rho/CDC42 in cancer progression (page 8 line 203).

3. The results of Figure 4 assessing the contribution of MPRIP alternative splicing are very interesting. While the differences observed in some assays (e.g. Fig 3f) are less compelling, it remains remarkable that modulation of a single alternative splicing event, out of the many regulated by RBFOX2, shows detectable effects in whole phenotypes.

We assume that the reviewer is referring to **Fig. 4f** here. We repeated this experiment (three times) and now present a representative picture (**Figure 1 for reviewer**) and quantification (**replaced Fig. 4g**).

Figure 1 for reviewer.

Representative pictures of wound healing assay of X50 metastatic cells transduced with either CRISPR control or DS-24 MPRIP sgRNA.

replaced Fig. 4g. Quantification of wound healing assay.

We now also include evidence of the effect of MPRIP inclusion in vivo (**new Fig. 4j**). We believe that the combination of both the in vitro and the new in vivo experiments present compelling evidence that the phenotype of metastatic cells is affected by inclusion of MPRIP exon 23.

new Fig. 4j. Quantification of the mean GFP intensity of lungs from NOD-SCID mice injected intravenously with GFP-labeled X50 metastatic cells expressing either CRISPR Cont. or DS-24 MPRIP sgRNA (n= 10 mice for each group). Lung metastases were visualized using a fluorescent microscope. Representative pictures and RT-PCR of RNA from two representative lungs from each group are shown on the right.

It may be relevant to analyze whether the level of alterations in this alternative splicing event (or more generally those events found to correlate with metastasis at transcriptome level) correlate with worse prognosis (e.g. using psichomics).

We appreciate the suggestions from this reviewer. We obtained overall survival information of PanCurX dataset and generated survival plots to show that patients in the MPRIP-low group, based on the median PSI value, have worse survival outcome (10.965 months shorter) (**new Fig. 4e**). We have also tried psichomics to obtain survival information from the TCGA's PAAD cohort. However, psichomics was only able to quantify 274 splicing events in 74 genes from the PAAD cohort. Our genes of interest were not in the 74 genes. We include our R scripts for psichomics process below.

We now added to the text (Page 10 lines 268 -270): "Survival analysis of patients in the MPRIP-low group, based on the median PSI value, have worse survival outcome (10.965 months shorter) (**Fig. 4e**)."

e.

new Fig. 4e. Kaplan–Meier curves of patients from PanCurX dataset used in this study separated into two groups, Low and High, based on MPRIP PSI-values. Median survival (Higher PSI) = 19.04 months, median survival (Lower PSI) =8.075 months, Hazard ratio (MPRIP-low) =9.0775. p<2.2e-16.

R script example:

```
library(psichomics)
folder <- getDownloadsFolder()
data <- loadFirebrowseData(folder=folder,
                           cohort="PAAD",
                           data=c("clinical", "junction_quantification",
                                   "RSEM_genes"),
                           date="2016-01-28")
```

```

clinical <- data[[1]]$`Clinical data`
sampleInfo <- data[[1]]$`Sample metadata`
junctionQuant <- data[[1]]$`Junction quantification (Illumina HiSeq)`
geneExpr <- data[[1]]$`Gene expression`
annotList <- listSplicingAnnotations()
hg38 <- listSplicingAnnotations(assembly="hg38")[[1]]
annotation <- loadAnnotation(hg38)
minReads <- 10
psi <- quantifySplicing(annotation, junctionQuant, minReads=minReads)
psi <- readRDS("PAAD_psi.RDS")
sampleInfo <- parseTCGAsampleInfo(colnames(psi))
events <- rownames(psi)
length(events)
[1] 274

```

Also related to this, the results of Fig 3d show that a number of primary tumors also show reduced PSI values. Is this the case for other RBFOX2 targets in the same samples?

We assume that the reviewer is referring to Fig. 4d here. We thank the reviewer for this question. We have now performed this analysis and the results are shown in **Figure 2 for reviewer**. One primary tumor had reduced PSI values for all three target genes (MPRIP, MYL6 and CLSTN1). Five primary tumors had reduced PSI values for two of the three target genes.

Figure 2 for reviewer.

Quantification of RT-PCR results of MPRIP, MYL6 and CLSTN1 alternative splicing events in pancreatic tumor samples. Analyzed using LabChipGX microfluidics platform.

4. There is an apparent discrepancy between the results of Figs 3b and 3e: the results of knockdown/overexpression of RBFOX2 (Fig 3b) consistently argue that the protein promotes accumulation of the longer MPRIP isoform. One would therefore expect that a mutation that prevents RBFOX2 binding would induce the shorter isoform, but the opposite is observed in Fig 4e. How do the authors explain this discrepancy?

We thank the reviewer for raising this question and giving us an opportunity to clarify. **Fig. 4b** shows the splicing of MPRIP in cells knocked-out for RBFOX2, resulting in skipping of MPRIP exon 23, producing the shorter isoform. RBFOX2, when it binds to its motif downstream to an exon, induces inclusion of that exon. The DS-24 gRNA MPRIP (used in **Fig. 4f**) targets the RBFOX2 motif located downstream of MPRIP exon 24. In this case, binding of RBFOX2 will cause inclusion of exon 24 which has a stop codon (skipping of exon 23), producing the shorter isoform. Here we used a sgRNA that destroyed this motif. In this case, we assume that RBFOX2 will now bind other RBFOX2 motifs, such as the one downstream to exon 23, thus inducing inclusion of exon 23. This will result in more of the longer isoform.

5. The results of Fig 4n are compelling and critically important for the main argument of the paper, and therefore it would be important to verify that the levels of MPRIP isoforms remain altered by the sgRNA treatments in the metastatic tumor samples.

We thank the reviewer for raising this issue. We have now repeated the *in vivo* experiment and verified the levels of MPRIP isoforms in the lung metastases. We performed RT-PCR on RNA extracted from lungs of mice injected with BxPC3 cells containing either CRISPR control or 5'ss sgRNA MPRIP (RT-PCR panel in Fig. 4o. We also include below (Figure 3 for reviewer) a panel of pictures of lung tumors (top) and RT-PCR (bottom) from mice injected with BxPC3 primary tumor cells containing either CRISPR control or 5'ss MPRIP sgRNA. Because the alteration occurs on the genomic DNA (CRISPR sgRNA), we can detect the altered human MPRIP isoform in the mouse lung tumor samples.

Figure 3 for reviewer. Representative pictures of lungs from mice injected with BxPC3 cells containing either CRISPR control or 5'ss sgRNA MPRIP (top) and RT-PCR of RNA extracted from the lungs of the injected mice (bottom).

6. It would be good to at least discuss the expectations/difficulties of the converse experiment: would promotion of the longer isoform inhibit metastatic potential of X50 cells?

We definitely agree with the reviewer and therefore have performed this experiment. The inclusion of exon 23 MPRIP in metastatic X50 cells inhibits metastatic potential *in vivo*. The results are now shown in new Fig. 4j.

We now added to the text (Page 10 lines 289 -292): "To examine the effects of MPRIP splicing modulation *in vivo*, we injected GFP- labeled X50 metastatic cells expressing either CRISPR control or DS-24 MPRIP sgRNA intravenously into NOD-SCID mice. Remarkably the inclusion of MPRIP exon 23 in metastatic X50 cells inhibits metastatic potential *in vivo* (Figs. 4j)."

new Fig. 4j. Quantification of the mean GFP intensity of lungs from NOD-SCID mice injected intravenously with GFP-labeled X50 metastatic cells expressing either CRISPR Cont. or DS-24 MPRIP sgRNA (n= 10 mice for each group). Lung metastases were visualized using a fluorescent microscope. Representative pictures and RT-PCR of RNA from two representative lungs from each group are shown on the right.

7. It would be also good to discuss whether evidence for downregulation of RBFOX2 protein levels and alterations in splicing of Rho GTPase pathway genes is observed in metastasis in other cancer types. We agree that it is important to discuss the complexity of RBFOX2 roles in other cancers. There are actually more papers claiming tumor/invasiveness-promoting roles for RBFOX2 in some cancers. For example, in breast and ovarian cancers, several papers have shown that RBFOX2 is up-regulated (due to EMT/TGF β signaling), induces a mesenchymal splicing signature and promotes oncogenic splice-switching that drives an invasive phenotype (Venables *et al.*, 2013; Braeutigam *et al.*, 2014; Tripathi *et al.*, 2019a; Ahuja *et al.*, 2020). Other papers have shown decreased expression of RBFOX2 in certain cancers (Venables *et al.*, 2009). We previously also reported a predicted lower expression in some cancers (Danan-Gotthold *et al.*, 2015), however until now these observations were mostly correlative. Thus, the findings we present here are suggestive that, at least in pancreatic cancer progression, RBFOX2 plays a tumor suppressive role by orchestrating an anti-invasive alternative splicing program.

We now added to the text (Page 5 lines 123 -130): "Several studies have reported that RBFOX2 is up-regulated (due to EMT/TGF β signaling), induces a mesenchymal splicing signature and promotes oncogenic splice-switching that drives an invasive phenotype in certain cancers (Ahuja *et al.*, 2020; Braeutigam *et al.*, 2014; Tripathi *et al.*, 2019; Venables *et al.*, 2013). Other studies have reported decreased expression of RBFOX2 in these cancers (Venables *et al.*, 2009). However, these observations were mostly correlative until now. Thus, the findings we present here suggest that, at least in pancreatic cancer progression, RBFOX2 plays a tumor suppressive role by orchestrating an anti-invasive alternative splicing program."

8. Another interesting discussion point is the potential mechanism by which RBFOX2 protein (but not mRNA) levels are downregulated in metastatic tissue.

We agree with the reviewer that this is an important point that might be complex and require further investigation on several levels. We have now addressed the initial layer of this question by examining the mRNA and protein half-lives of RBFOX2 in the primary (BxPC3) and metastatic (X50) pancreatic cancer cells.

We found that while the mRNA half-life of RBFOX2 is similar, RBFOX2 protein half-life is shorter in the metastatic X50 cells suggesting enhanced protein degradation (**new Extended Data Figure 3f-g**). Further experiments are required to understand if additional mechanisms, such as translational regulation or RNA processing, play a role in RBFOX2 down-regulation as well. A recent report shows that in some conditions RBFOX2 protein is down-regulated by proteasomal degradation upon binding to a p53-induced lncRNA in gastric cancer (Ou *et al.*, 2022). However, in this case, RBFOX2 down-regulation inhibits, and RBFOX2 overexpression promotes, oncogenesis.

We now added to the text (Page 5 lines 109-112): " Actinomycin D treatment showed that RBFOX2 mRNA half-life is similar in the metastatic and the primary tumor cell lines. RBFOX2 protein half-life is shorter in the metastatic X50 cells suggesting enhanced protein degradations (**Extended Data Figs. 3f-g**)."

f.

g.

new Extended Data Figure 3f. BxPC3 primary tumor cells and X50 metastatic cells were treated with actinomycin D (10 μ g/ml), RNA was extracted at different time points and qRT-PCR of RBFOX2 mRNA was performed. **g.** quantification of protein levels of RBFOX2 in BxPC3 primary tumor cells and X50 metastatic cells were treated with cyclohexamide (10 μ g/ml), at different time points.

Reviewer 2

Finding RBFOX2 as a master regulator of RNA splicing events that limit PCA metastasis, thereby rationalizing its observed downregulation in metastases, is new and interesting.

We thank the reviewer for recognizing the importance of our work.

1. The authors report a list of genes that are distinguished by alternative splicing events in human PCA primary and metastatic tumors mediated by the presence or absence of RBFOX2, and assert that these changes are not associated with changes in gene expression. The authors should show an mRNA heat map with an identical gene list to that shown in Figure 1d for splicing events, to substantiate their conclusion that changes in gene expression of these candidates is not changed. Other figure panels could be moved to an ED-Figure to make space.

We thank the reviewer for their helpful suggestion. We now include a heatmap of gene expression with a gene list based on the top 20 differentially spliced events (18 genes) to that shown in Fig. 1d (new Extended Data Figure 1h).

We now added to the text (Page 4 lines 77 -79): " However, such clustering based on gene expression of these genes did not separate to any groups (Extended Data Fig. 1h). "

new Extended Data Figure 1h. Unsupervised clustering of PDA patient samples, based on the top 20 differentially spliced events (PSI) (18 genes) that were identified in Fig. 1b.

2. Also, ILF3 is listed three times in Fig 1d, and SRRM2. Are these different splicing events in the same transcript, and if so is there a need to keep them separate?

The reviewer is correct; these are different splicing events for the same gene. This is now clarified in Fig. 1b (addition of asterisks) and figure legend.

3. In Figures 1.h and 1.i, the authors relay that RBFOX2 is under translational control. Since the data indicate that none of the signature mutations in PCA correlate with the differential splicing events in which RBFOX2 is implicated, the authors should provide some clues into how this downregulation of RBFOX2 protein is achieved in metastases? E.g., upregulating microRNAs or RNA-binding translational repressor proteins?

We agree with the reviewer that this is an important point that might be complex and requires further investigation on several levels. Since we found that both RBFOX2 mRNA levels and its mRNA half-life (see below) are the same, we did not examine potential miRNA-mediated degradation. As for miRNA/RBP-mediated translational control, this is a plausible mechanism that requires further studies.

We addressed the initial layer of this question by examining the mRNA and protein half-lives of RBFOX2 in the primary (BxPC3) and metastatic (X50) pancreatic cancer cells. We found that while the mRNA half-life of RBFOX2 is similar, RBFOX2 protein half-life is shorter in the metastatic X50 cells, suggesting enhanced protein degradation (**see response to Reviewer 1 comment 8, new Extended Data Figure 3f-g**). Further experiments are required to understand if additional mechanisms, such as translational regulation or RNA processing, play a role in RBFOX2 down-regulation as well. A recent article reports that in some conditions RBFOX2 protein is down regulated by proteasomal degradation upon binding to a p53-induced lncRNA in gastric cancer (Ou *et al.*, 2022). However, in this case, RBFOX2 down-regulation inhibits, and RBFOX2 overexpression promotes, oncogenesis.

We now added to the text (Page 5 lines 109 -112): " Actinomycin D treatment showed that RBFOX2 mRNA half-life is similar in the metastatic and the primary tumor cell lines. RBFOX2 protein half-life is shorter in the metastatic X50 cells suggesting enhanced protein degradations (**Extended Data Figs. 3f-g**)."

new Extended Data Figure 3f. BxPC3 primary tumor cells and X50 metastatic cells were treated with actinomycin D (10µg/ml), RNA was extracted at different time points and qRT-PCR of RBFOX2 mRNA was performed. **g.** quantification of protein levels of RBFOX2 in BxPC3 primary tumor cells and X50 metastatic cells were treated with cycloheximide (10µg/ml), at different time points.

4. The pharmacological inhibitors of implicated downstream pro-metastatic effectors of the loss of RBFOX2 seem unlikely to be pure metastasis suppressors. The authors should also treat s.c. transplant tumors and assess tumor growth of primary tumors, to address their specificity for impairing metastasis. There is no indication of differential proliferation or survival of primary vs metastatic human or mouse PDA cancer cells in vivo, and yet data is presented indicating that these inhibitors affect cell survival.

We agree with the reviewer and therefore performed several experiments to address this question. Surprisingly, while Azathioprine inhibited the primary tumor growth of the non-metastatic cell line

BXPC3, it did not inhibit primary tumor growth of the metastatic cell line X50. This result suggests that in vivo, Azathioprine inhibits metastasis of X50 but not its primary growth (**new Extended Data Figure 8g-h and 8k-l**). This result suggests that in X50 cells, the targets of Azathioprine are important for metastasis but not for tumor growth.

We agree with the reviewer that the effects on proliferation of Azathioprine are different than the effects of the genetic manipulation of RBFOX2 and its splicing targets (none of which inhibited proliferation). For this reason, we added experiments using new Rac1/CDC42 inhibitor (MBQ-167) which is more specific and do not affect proliferation/survival but do inhibit migration of RBFOX2 knockout cells. In vivo administration of MBQ-167 to mice injected intravenously with either GFP-labeled RBFOX2-depleted BxPC3 cells or GFP-labeled PDX-derived metastatic X50 cells inhibited lung metastases (**Figs. 3j-k, Extended Data Figs. 8a-b**), without an effect on primary tumor growth subcutaneously (**Extended Data Figs. 8e-f and 8i-j**). (see the results in response to reviewer 1 point 2, new Fig. 3g-k and new Extended Data Fig. 8a-l). Moreover, we generated knockout of Rac1 in BxPC3 cells with RBFOX2 knockout. Both the Rac1 inhibitors as well as Rac1 knockouts reverse the enhanced migration effects of RBFOX2 knockout, without an effect on their proliferation, suggesting that indeed RBFOX2 acts also upstream to Rac1 in modulating cellular motility/invasiveness (see the results in response to reviewer 1 point 2, new Extended Data Fig. 8m-p).

new Extended Data Figure 8e-l. **e.** Tumor volumes of tumors formed in NOD-SCID mice injected subcutaneously with BxPC3 cells and treated with either Vehicle or MBQ-167 starting 5 days after cell injection (3 mg/kg, 3 times a week for 5 weeks) (n=5 mice for the vehicle group and n=4 mice for the treated group, two tumors per mouse) **(left)**. Representative pictures of the tumors **(right)**. **f.** weight measure of the mice throughout the experiment described in **(e)**. **g.** Tumor volumes of tumors formed in NOD-SCID mice injected subcutaneously with BxPC3 cells and treated with either PBS or Azathioprine starting 5 days after cell injection (10 mg/kg, 3 times a week for 5 weeks) (n=5 mice for the vehicle group and n=4 mice for the treated group, two tumors per mouse) **(left)**. Representative pictures of the tumors **(right)**. **h.** weight measure of the mice throughout the experiment described in **(g)**. **i.** Tumor volumes of tumors formed in NOD-SCID mice injected subcutaneously with X50 cells and treated with either Vehicle or MBQ-167 starting 5 days after cell injection (3 mg/kg, 3 times a week for 5 weeks) (n=5 mice per group, two tumors per mouse) **(left)**. Representative pictures of the tumors **(right)**. **j.** weight measure of the mice throughout the experiment described in **(i)**. **k.** Tumor volumes of tumors formed in NOD-SCID mice injected subcutaneously with X50 cells and treated with either PBS or Azathioprine starting 5 days after cell injection (10 mg/kg, 3 times a week for 5 weeks) (n=4 mice for the vehicle group and n=5 mice for the treated group, two tumors per mouse) **(left)** Representative pictures of the tumors **(right)**. **l.** weight measure of the mice throughout the experiment described in **(k)**.

5. The scratch migration (“wound healing”) assay used throughout the manuscript is an imprecise metric of bona fide invasiveness. Have Boyden chamber assays involving invasion through matrigel or other ECM been performed? If not, the limitations of the migration assay should be described.

We did not use Boyden chamber assays in this work. We agree with the reviewer that the migration assay we used has limitations. Given the known limitations of the wound healing assay to measure migration ability (no extracellular matrix, variation of wound width, influence on cell proliferation, cell damage), we tried to overcome some of these limitations.

We now describe this in the Materials and methods section under “Quantitative wound healing assay”. We write: “We used an automated Incucyte woundmaker tool to create precise, uniform cell-free zones in cell monolayers enabling real time, automated measurement of cell migration with a tool to measure the density of the wound region relative to the density of the cell region.” It should be noted that the results of the wound healing assays are supported by both in vitro proliferation assays, that exclude any proliferation effect, and in vivo experiments, that support the biological significance.

6. Three candidate alternatively spliced targets of RBFOX2 are introduced and functionally assessed in analogous assays to RBFOX2 itself: MPRIP, MYL6 and CLSTN1. All three have similar effects in the migration assay, noting its limitations. Tail vein metastasis assays are not included for MYL6 and CLSTN1 and should be.

We thank the reviewer for this suggestion. We now include tail vein metastasis assays for the other two RBFOX2 targets (MYL6 and CLSTN1) **(Extended Data Figures 10e and 10k)**.

We now added to the text (Page 12 lines 329 -331): “GFP- labeled BxPC3 primary tumor cells with 3' ss for either MYL6 or CLSTN1 or control sgRNAs injected intravenously into NOD-SCID mice revealed a significant increase in the number of lung metastatic foci **(Extended Data Fig. 10e and 10k)**.”

e.

New Extended Data Figure 10e.

Quantification of the mean GFP intensity of lungs from NOD-SCID mice injected intravenously with GFP-labeled BxPC3 primary tumor cells expressing either CRISPR Cont. or 3'ss MYL6 sgRNA (n= 8 mice for each group). Lung metastases were visualized using a fluorescent microscope. Representative pictures and RT-PCR of RNA from two representative lungs from each group are shown on the right.

k.

New Extended Data Figure 10k.

Quantification of the mean GFP intensity of lungs from NOD-SCID mice injected intravenously with GFP-labeled BxPC3 primary tumor cells expressing either CRISPR Cont. or 3'ss CLSTN1 sgRNA (n= 10 mice for each group). Lung metastases were visualized using a fluorescent microscope. Representative pictures and RT-PCR of RNA from two representative lungs from each group are shown on the right.

Moreover, the similarity of results with the four genes - RBFOX2, MPRIP, MYL6 and CLSTN1 – begs the question: Are all four in one pathway? Epistasis assays, e.g. co-modulating RBFOX2 and each of the three downstream effectors in the metastasis assay (and maybe also the migration assay, where effects are quite modest in general) would shed light on this question as would co-modulating all three candidate effectors together.

We have now performed wound healing assay with metastatic cells overexpressing RBFOX2 modulated for targets either individually or in combination. From these experiments we can conclude that modulation of MPRIP had the greatest effect, increasing wound healing potential in the cells. Modulation of each of the targets alone had similar effect, reversing the phenotype to the phenotype of control cells (without RBFOX2 overexpression). We did not observe any synergistic effect with co-modulation of all three targets. This suggests that the targets either have similar functions or are part of the same pathway (**Figure 4 for reviewer**).

Figure 4 for reviewer.

X50 cells overexpressing RBFOX2 were modulated for RBFOX2 targets either individually (3'ss MYL6, 5'ss MPRIP, 3'ss CLSTN1) or in combination (5'ss MPRIP, 3'ss MYL6, 3'ss CLSTN1). RT-PCR splicing validation of the modulated cells (left). Quantification of wound healing assay using the above cells (right).

More sophisticated analytical procedures might also be informative, for example performing immunohistochemistry on the metastases, rather than just showing and quantitating fluorescent protein signal.

We agree with the reviewer that further analytical procedures might be informative. However, due to financial and ethical burden of repeating all the mouse experiments we are not able to perform all the in vivo experiments again for this analysis. We therefore performed H&E staining on a representative experiment, lung metastases from mice injected with BxPC3 primary cells with 3'ss MYL6 sgRNA. The H&E staining recapitulates the fluorescence staining, showing increased lung metastases in mice injected with BxPC3 primary cells with 3'ss MYL6 sgRNA compared to control BxPC3 cells (**Figure 5 for reviewer**).

Figure 5 for reviewer. a. Pictures of lungs from NOD-SCID mice injected intravenously with GFP-labeled BxPC3 primary tumor cells expressing either CRISPR Cont. or 3'ss MYL6 sgRNA (n= 8 mice for each group). Lung metastases were visualized using a fluorescent microscope. b. H&E staining of tissue slices from the above lungs. c. RT-PCR of RNA from representative lungs from each group.

Reviewer 3

The novelty of the manuscript is the identification of a novel splicing factor, RBFOX2, as a potent metastasis suppressor in pancreatic primary tumour cells.

The experimental design to identify the alternatively spliced mRNAs are complementary and robustly support their findings using different pancreatic tumour cells and patient samples. The text is well written, and the rationale of the experimental approach clear to follow. The figures and data presentation are of high quality.

We thank the reviewer for the compliments on our work.

Specific comments:

1. While validating MPRIP isoforms, it would be important to show what is the consequence of skipping or inclusion of exon 23 for example. Does inclusion of exon 23 provides phosphorylation sites that could regulate MPRIP function?

In an attempt to uncover the mechanism of action of the spliced isoform of MPRIP we analyzed the potential kinase repertoire predicted to phosphorylate each serine/threonine in the unique sequence of each MPRIP isoform using a serine-threonine kinome analysis a prediction tool (Johnson *et al.*, 2022). This analysis identified that the MPRIP splicing isoforms have differing phosphorylation sites which are predicted to be phosphorylated by various kinases (**Extended Data Fig 9a**). One example is phosphorylation of Ser1016 (which is present in both isoforms but has different kinase predictions due to unique downstream amino acid sequences of each MPRIP isoform). The skipped isoform received a high score rank for PKCs kinases. The PKC family functions downstream of focal adhesion complex formation and regulates focal adhesion assembly and dynamics (Fogh, Multhaupt and Couchman, 2014). Phosphorylation on this site may function as a feedback mechanism. This could explain the increase in focal adhesion formation we observed in cells that expressed mainly the MPRIP skipped isoform.

Moreover, using AlphaFold prediction tool (Fogh, Multhaupt and Couchman, 2014) we established that the C-terminus of the MPRIP skipped exon 23 isoform has an alpha-helical structure. This region is predicted to project outwards from the protein, in contrast to the shorter C-terminus from exon 23 included isoform (**Extended Data Figs 9b**). This alpha-helical structure might allow interaction of the oncogenic MPRIP skipped isoform with different binding partners.

This is now added to the text (page 11 lines 296 -305) " In an attempt to uncover the mechanism of action of the spliced isoform of MPRIP we analyzed the potential kinase repertoire predicted to phosphorylate each serine/threonine in the unique sequence of each MPRIP isoform using a serine-threonine kinome analysis a prediction tool⁴⁸. This analysis identified that the MPRIP splicing isoforms have differing phosphorylation sites which are predicted to be phosphorylated by various kinases (Extended Data Fig. 9a). Moreover, using AlphaFold prediction tool ⁴⁹ we established that the C-terminus of the MPRIP skipped exon 23 isoform has an alpha-helical structure. This region is predicted to project outwards from the protein, in contrast to the shorter C-terminus from exon 23 included isoform (Extended Data Fig. 9b). This alpha-helical structure might allow interaction of the oncogenic MPRIP skipped isoform with different binding partners."

Extended Data Figure 9 new a. Summary of predicted kinases for different phosphorylation sites for each MPRIP isoforms as predicted by serine-threonine kinome prediction tool. b. Structure analysis of the C-terminus of each MPRIP isoforms as predicted by AlphaFold prediction tool.

Or affect its interactions with RhoA, thereby altering its signalling?

We have now performed immunoprecipitation on 293T cells transfected with either FLAG-MPRIP exon 23 included isoform or FLAG-MPRIP exon 23 skipped isoform. The pulled down proteins were analyzed by mass spectrometry. Comparison of the proteins pulled down by each isoform reveals enrichment for proteins in different pathways. The MPRIP exon 23 skipped isoform pulled down proteins enriched in MAPK family signaling cascades, MAPK1/MAPK signaling and RAF/MAP kinase cascade and cell cycle pathways. The MPRIP exon 23 included isoform pulled down proteins enriched in a variety of different pathways. Both isoforms pulled-down proteins in the RHO GTPase Effectors, as expected, with the skipped isoform binding more proteins of this pathway (**new Extended Data Figure 10**). These results are very interesting and we plan to pursue further research in this direction.

This is now added to the text (page 11 lines 306 -313) " In addition, we performed immunoprecipitation with either isoform followed by mass spectrometry (Fig. 4q). These experiments revealed enrichment for proteins in different pathways. The MPRIP exon 23 skipped isoform pulled down proteins enriched in MAPK family signaling cascades, RAF/MAP kinase cascade and cell cycle pathways (Extended Data Figs. 9c-f). Co-immunoprecipitation experiments recapitulated the binding of MPRIP exon 23 skipped isoform to A-Raf (Extended Data Fig. 9g). Both isoforms pulled-down proteins in the Rho GTPase Effector pathways, as expected, with the skipped isoform binding more proteins of this pathway (Extended Data Figs. 9c-f)."

Figure 4. q. Volcano plot representation the label-free quantitation (LFQ) intensities ratio of MPRIP exon 23 skipped isoform to MPRIP exon 23 included isoform.

Extended Data Figure 9 new c. Volcano plot representation the ratio of label-free quantitation (LFQ) intensities of the proteins pulled-down by MPRIP exon 23 included isoform and empty vector. **d.** Reactome pathway analysis of the proteins identified in (a). **e.** Volcano plot representation the ratio of label-free quantitation (LFQ) intensities of the proteins pulled-down by MPRIP 23 skipped isoform, and empty vector. **f.** Reactome pathway analysis of the proteins identified in (b). **g.** western blot of total lysate from HEK293 cells co- transfected with either empty vector, FLAG-MPRIP exon 23 included isoform or FLAG-MPRIP exon 23 skipped isoform (upper panel). Western blot of immunoprecipitation of lysates of these cells with anti-A-Raf and anti-FLAG.

2. In figure 4, skipping exon 23 of MPRIP leads to a striking phenotype of more efficient wound healing (Fig.4i-j) and higher lung colonization in vivo (Fig.4n). However, immunofluorescence staining shows that these cells have much larger and thicker focal adhesions that would be predicted to bind better to ECM and have slower motility (Fig.3m). How can these results be conciliated?

We agree with the reviewer that the presence of larger and thicker focal adhesions seems not in line with enhanced migration and invasion. However, previous studies have shown that focal complexes and nascent focal adhesions are formed during migration, apparently acting as mechanical anchor points that promote the actin polymerization-based protrusion at the leading edge. Moreover, although sometimes counterintuitive, enhanced focal adhesion formation assembly (and specifically invadopodia and podosome formation) can contribute to migration and invasion (Seetharaman and Etienne-Manneville, 2019; Zhong *et al.*, 2021; Xu *et al.*, 2022). We have not fully characterized the focal adhesions shown in the figures as invadopodia, but the functional experiments suggest that this is indeed the case.

We have now added an explanation to the main text (page 8 lines 205-212):" Previous studies have shown that focal complexes and nascent focal adhesions are formed during migration, apparently acting as mechanical anchor points that promote the actin polymerization-based protrusion at the leading edge. The formation of these complexes are dynamic. Although sometimes counterintuitive, enhanced focal adhesion formation assembly (and specifically invadopodia and podosome formation) can contribute to migration and invasion."

3. It is also remarkable that MPRIP exon skipping by itself (Fig.3m) can mimic the focal adhesion phenotype of RBFOX sgRNA (Fig.3d). Would alternative splicing of other RhoA targets also phenocopy this?

We thank the reviewer for this suggestion. We have now included immunofluorescence staining of paxillin to detect focal adhesion formation in BxPC3 primary tumor cells with either CRISPR control of 3' or 5' ss of either MYL6 or CLSTN1 (**Extended Data Figure 10 f and I**). Similar as to what was observed for MPRIP exon skipping (**Fig. 4p**) and RBFOX sgRNA (**Fig. 3d**), splicing of these targets increased focal adhesion formation.

This is now added to the text (page 12 lines 313 -333) " Moreover, more focal adhesions were observed in BxPC3 cells expressing either 3'ss MYL6 or CLSTN1 sgRNAs compared to control cells (**Extended Data Fig. 10f and 10I**)."

Extended Data Figure 10 new panel I. IF of Paxillin in BxPC3 cells transduced with 3'ss and 5'ss CLSTN1 sgRNAs. Paxillin: green, DAPI: blue. Scale bar 20 μ m.

Extended Data Figure 10 new panel f. IF of Paxillin in BxPC3 cells transduced with 3'ss and 5'ss MYL6 sgRNAs. Paxillin: green, DAPI: blue. Scale bar 20 μ m.

Is a reversion to a more epithelial-type morphology also observed with MPRIP splicing?

Yes, this is shown in **Fig. 4p** (left panels).

4. Are Rac1b levels affected by RBFOX2? This is a well-characterized isoform of Rac1 that is upregulated in different tumours and with well mapped distinct signalling from Rac1. It will be important to report either way.

We thank the reviewer for suggesting this experiment. We do not detect any difference in Rac1 splicing in either BxPC3 primary tumor cells knocked-down for RBFOX2 or X50 metastatic cells overexpressing RBFOX2. We also do not detect any significant difference in Rac1 splicing in primary and metastatic pancreatic patient samples using RT-PCR (**Figure 6 for reviewer**).

Figure 6 for reviewer. RT-PCR analysis was performed on RNA from primary BxPC3 cells with two different RBFOX2 sgRNAs and X50 metastatic cells RBFOX2 overexpression for Rac1 splicing (left). Quantification of RT-PCR of Rac1 PSI from pancreatic tumor samples (right).

5. The rationale for investigating Rac1 and Cdc42 signalling is not clear, in line with the above experiments focusing on RhoA pathway. The choice of the inhibitor used (Azathioprine) is not appropriate. This drug is not specific for Rac1 or Cdc42 as it broadly inhibits DNA, RNA and protein synthesis. Thus, the approach is not specific and does not determine causality. There are other specific inhibitors for these GTPases that could have been used to support their claim.

The rationale for investigating Rac1 and Cdc42 signaling is based on the reactome analysis performed on oppositely spliced events identified in our RNA-seq splicing analysis of RBFOX2 manipulated pancreatic tumor cells (**Fig. 3c**). We also agree that the use of Azathioprine may not be the best choice.

We have now added several experiments that tie RBOX2 as an upstream regulator of Rho GTPase pathways effect on migration, as well as, additional Rac1 inhibitor which is more specific than Azathioprine. It is important to note that knockout of Rac1 or use more specific inhibitor, MBQ-167 (Rac1 and Cdc42 inhibitor), do not affect the proliferation or survival of the cells (see below).

We now replace the experiments using azathioprine (**Fig. 3g-k**) with experiments using MBQ-167 on BxPC3 primary tumor cells with RBOX2 knockout, which become metastatic only due to this single gene knockout and with PDX-derived metastatic X50 cells (**new Fig. 3g-k**). We removed Extended Data Fig. 8 due to use of high MBQ-167 dosage that causes cell death and inhibition of proliferation. We have calibrated the dose and now use a lower dose (**new Fig. 3g-k**).

new Fig.3 g. Quantification of wound healing assay of BxPC3 RBOX2 sgRNA-1 cells treated with either MBQ-167 (0.05 μ M) or DMSO. **h.** Quantification of proliferation assay of BxPC3 sgRNA-1 cells treated with either MBQ-167 (0.05 μ M) or DMSO. **i.** Trypan blue-based cell count of BxPC3 sgRNA-1 cells treated with either MBQ167 (0.05 μ M) or DMSO for 24 hours. **j.** Quantification of the mean GFP intensity of lungs from NOD-SCID mice injected intravenously with GFP-labeled BxPC3 RBOX2 sgRNA-1 cells treated either with vehicle or MBQ167 3mg/Kg (n= 7 mice for each group) (**left**). Representative pictures of the lungs, visualized by fluorescent microscopy (**right**). **k.** Weight measure of the mice throughout the experiment described in (**j**).

new Extended Data Fig. 8. a. Quantification of representative lungs from NOD-SCID mice injected intravenously with GFP-labeled X50 cells and treated with either Vehicle or MBQ-167 starting 5 days after cell injection (3 mg/kg, 3 times a week for 5 weeks) (n=8 mice per group). Representative pictures are shown on the right. **b.** weight measure of the mice throughout the experiment described in (a).

Reviewer Reports on the First Revision:

Referees' comments:

Referee #1 (Remarks to the Author):

I appreciate the efforts that the authors have made to address the issues raised in my previous report. In my opinion the paper has been strengthened considerably and is acceptable for publication in Nature. It will have a significant impact on both the cancer and RNA biology communities.

Referee #2 (Remarks to the Author):

The manuscript is significantly improved, with important new data added, addressing most of the critiques of the three reviewers, and is judged to be appropriate albeit not compelling for publication in Nature.

Several points.

It remains perplexing that the authors did not collect tumor samples for histological analysis. While the claim is a lengthy veterinary approval process did not allow a repeat of this lapse in experimental protocol to collect samples, the authors were able to perform new in vivo experiments, and still did not collect samples for histology? Definitely not state-of-the-art for studying mouse models of cancer.

The scratch assay is also not state-of-the-art for invasiveness, but the in vivo metastasis assays alleviate the necessity to perform more convincing assays, e.g., invasion in Boyden chambers through ECM analogs.

TV metastasis for MYL6 and CLSTN1: If you look at the data regarding CLSTN1 (ED Fig. 10K), it is not significant at all, despite a marginal low p-value. It is fine that this particular one didn't work similarly to the other two candidates, but this should be mentioned in the text.

The epistasis analysis is important, but only shown in Figure 4 for the reviewers, which is understandable since only the unconvincing scratch assay was used. If TV metastasis assays of the epistasis analysis were available, then the data should be added to the manuscript.

Referee #3 (Remarks to the Author):

The authors did a very good job in addressing the reviewers' comments and generated further experimental data to strengthen their data, generating interesting results. The authors provided initial characterization of the binding profile, protein modulation and putative phosphorylation of the different isoforms. The dissection of the molecular mechanisms of the different oncogenic mechanisms will be fascinating in future experiments. This reviewer finds the overall data more convincing and better supported.

Rebuttal Reviewer 3 point 5 – perhaps my comments about specificity of Azathioprine were not very clear. There is solid biochemical evidence that Azathioprine may interfere with Rac1 activation in two papers (doi.org/10.1172/JCI16432 and [PMC1965586/](https://pubmed.ncbi.nlm.nih.gov/1965586/)). These papers should be cited instead of the commentary paper Ref 45. However, the point is the interpretation of placing all the Azathioprine effects on Rac1 signalling alone: (i) in principle the metabolite 6-thio-GTP could bind to any other small GTPase of the Ras superfamily, (ii) only a couple of small GTPases were tested in those papers. However in various parts of the text there are strong statements that Azathioprine works via Rac1 – it is best to tone down such sentences in page 8 and others. This is not in any way to undermine the conclusion of Rac1 involvement, but rather to be more precise and balanced on attributions to Azathioprine and exon skipping mechanisms. For example, as added to the revised manuscript MPRIP isoforms would primarily interfere with RhoA pathways, not Rac1, and Fig.4q suggests that RhoG protein levels are reduced in MPRIP exon 23 skipped form.

Author Rebuttals to First Revision:

Point-by-point response:

Referee #1 (Remarks to the Author):

I appreciate the efforts that the authors have made to address the issues raised in my previous report. In my opinion the paper has been strengthened considerably and is acceptable for publication in Nature. It will have a significant impact on both the cancer and RNA biology communities.

We thank the reviewer for the positive remarks.

Referee #2 (Remarks to the Author):

The manuscript is significantly improved, with important new data added, addressing most of the critiques of the three reviewers, and is judged to be appropriate albeit not compelling for publication in Nature.

Several points.

It remains perplexing that the authors did not collect tumor samples for histological analysis. While the claim is a lengthy veterinary approval process did not allow a repeat of this lapse in experimental protocol to collect samples, the authors were able to perform new in vivo experiments, and still did not collect samples for histology? Definitely not state-of-the-art for studying mouse models of cancer.

We have collected all the lung tissues from the new in vivo experiments included in the revised manuscript (over 100 mice) and formalin-fixed them. We have now included representative sections with H&E staining (Fig. 3j, Figs. 4i and 4n, Extended Data Figs. 8a and 8p, Extended Data Figs. 10g and 10o), and for some of them, also GFP immunohistochemistry for the new in vivo experiments (Figs. 4i and Extended Data Figs. 8a).

The scratch assay is also not state-of-the-art for invasiveness, but the in vivo metastasis assays alleviate the necessity to perform more convincing assays, e.g., invasion in Boyden chambers through ECM analogs.

The scratch assay we performed (Incucyte) is fully automated with multiple biological and technical repeats. This manuscript includes experiments with over 200 mice and shows a positive correlation between this in vitro assay (on these cells and this mouse model) and the metastasis assay in vivo. We agree with the reviewer that this correlation alleviates the necessity to perform additional assays.

TV metastasis for MYL6 and CLSTN1: If you look at the data regarding CLSTN1 (ED Fig. 10K), it is not significant at all, despite a marginal low p-value. It is fine that this particular one didn't work similarly to the other two candidates, but this should be mentioned in the text.

We fully agree with the reviewer and added a sentence to mention that this target is a weaker effector of metastasis (page 9, line 248).

The epistasis analysis is important, but only shown in Figure 4 for the reviewers, which is understandable since only the unconvincing scratch assay was used. If TV metastasis assays of the epistasis analysis were available, then the data should be added to the manuscript.

The in vitro epistasis analysis shows that the combinatory effect of the three targets is not synergistic or additive and did not have a stronger effect than modulation of the individual

splicing events. We have now included this experiment in the manuscript (Fig. 4p, Extended Data Figs. 10e-f and 10 m-n).

Referee #3 (Remarks to the Author):

The authors did a very good job in addressing the reviewers' comments and generated further experimental data to strengthen their data, generating interesting results. The authors provided initial characterization of the binding profile, protein modulation and putative phosphorylation of the different isoforms. The dissection of the molecular mechanisms of the different oncogenic mechanisms will be fascinating in future experiments. This reviewer finds the overall data more convincing and better supported.

We thank the reviewer for the positive comments.

Rebuttal Reviewer 3 point 5 – perhaps my comments about specificity of Azathioprine were not very clear. There is solid biochemical evidence that Azathioprine may interfere with Rac1 activation in two papers (doi.org/10.1172/JCI16432 and [PMC1965586/](https://pubmed.ncbi.nlm.nih.gov/1965586/)). These papers should be cited instead of the commentary paper Ref 45.

We have now cited these references.

However, the point is the interpretation of placing all the Azathioprine effects on Rac1 signalling alone: (i) in principle the metabolite 6-thio-GTP could bind to any other small GTPase of the Ras superfamily, (ii) only a couple of small GTPases were tested in those papers. However in various parts of the text there are strong statements that Azathioprine works via Rac1 – it is best to tone down such sentences in page 8 and others. This is not in any way to undermine the conclusion of Rac1 involvement, but rather to be more precise and balanced on attributions to Azathioprine and exon skipping mechanisms. For example, as added to the revised manuscript MPRIP isoforms would primarily interfere with RhoA pathways, not Rac1, and Fig.4q suggests that RhoG protein levels are reduced in MPRIP exon 23 skipped form.

We have now toned down our statements about Azathioprine (page 6, lines 162-163).

Reviewer Reports on the Second Revision:

Referees' comments:

Referee #2 (Remarks to the Author):

The authors have appropriately addressed the critiques of this and the other reviewers and the manuscript is now judged to be meritorious for publication in Nature.

One final suggestion (or two), however: The summary would be strengthened by describing the identification, functional genetic studies and implications of the RBFOX2 target MPRIP, in regard to the effects of inclusion vs skipping of exon 23 of MPRIP. Many of the sentences in the abstract are verbose and could easily be streamlined to make space for a sentence or two on MPRIP. (It is much more compelling than the other two implicated targets.)

There are also grammatical errors here and there – try Grammarly.com, as I insist for my lab members.

Author Rebuttals to Second Revision:

Revision 3:

Referee #2 (Remarks to the Author):

The authors have appropriately addressed the critiques of this and the other reviewers and the manuscript is now judged to be meritorious for publication in Nature.

We thank the reviewer for the positive remarks.

One final suggestion (or two), however: The summary would be strengthened by describing the identification, functional genetic studies and implications of the RBFOX2 target MPRIP, in regard to the effects of inclusion vs skipping of exon 23 of MPRIP. Many of the sentences in the abstract are verbose and could easily be streamlined to make space for a sentence or two on MPRIP. (It is much more compelling than the other two implicated targets.)

Thank you for your suggestion. We now included MPRIP in the summary.

There are also grammatical errors here and there – try Grammarly.com, as I insist for my lab members.

Thank you for bringing attention to the grammatical errors.